# One for Two: A Unified Framework for Imbalanced Graph Classification via Dynamic Balanced Prototype

**Guanjun Wang**[1], **Binwu Wang**[1,2,*], **Jiaming Ma**[1],
**Zhengyang Zhou**[1,2], **Pengkun Wang**[1,2], **Xu Wang**[1,2], **Yang Wang**[1,2,*]
[1] University of Science and Technology of China (USTC), Hefei, Anhui, China
[2] Suzhou Institute for Advanced Research, USTC, Suzhou, Jiangsu, China
`{always, JiamingMa}@mail.ustc.edu.cn,`
`{wbw2024, zzy0929, pengkun, wx309, angyan}@ustc.edu.cn`

## Abstract

Graph Neural Networks (GNNs) have advanced graph classification, yet they remain vulnerable to graph-level imbalance, encompassing class imbalance and topological imbalance. To address both types of imbalance in a unified manner, we propose **UniImb**, a **Uni**fied framework for **Imb**alanced graph classification. Specifically, UniImb first captures multi-scale topological features and enhances data diversity via learnable personalized graph perturbations. It then employs a dynamic balanced prototype module to learn representative prototypes from graph instances, improving the quality of graph representations. Concurrently, a prototype load-balancing optimization term mitigates dominance by majority samples to equalize sample influence during training. We justify these design choices theoretically using the Information Bottleneck principle. Extensive experiments on **19** datasets-including a large-scale imbalanced air pollution graph dataset AirGraph released by us and **23** baselines demonstrate that UniImb has achieved dominant performance across various imbalanced scenarios. Our code is available at GitHub.

## 1 Introduction

Graph Neural Networks (GNNs) have achieved remarkable performance in graph classification tasks by iteratively aggregating local neighborhood information (Wei et al., 2023; Nguyen et al., 2022; Wu et al., 2020; Fan et al., 2023). However, the success of most architectures relies heavily on balanced graph datasets (Ma et al., 2025). In real-world scenarios, graph data often exhibits imbalanced distributions, which can be mainly categorized into two aspects:

❶ **Class Imbalance**. A few classes contain most of the samples while many classes are severely underrepresented, leading GNNs to favor head classes during training (Tang & Liang, 2023). (Tang & Liang, 2023). A promising approach to mitigate class imbalance is the graph-of-graphs framework (Wang et al., 2022b; Tang & Liang, 2023), which treats each input graph as a node in a higher-order meta-graph. Edges in the meta-graph are formed based on graph-level feature similarity, enabling information propagation among similar graphs and thereby alleviating data sparsity in tail classes.

❷ **Topological Imbalance**. Graph datasets frequently show a highly skewed distribution of node counts: a small number of large graphs contain most nodes, while the majority of graphs are substantially smaller. This imbalance often causes GNNs to focus disproportionately on larger graphs during training, leading to diminished performance on small-scale ones (Qin et al., 2025). To address this challenge, topology imbalance methods identify underrepresented small-scale graphs during training and enhance their contributions through reweighting or data augmentation, yielding more consistent performance (Liu et al., 2022c; Xu et al., 2024).

Despite remarkable progress, existing imbalanced graph learning methods focus unilaterally on either class imbalance or topological imbalance in isolation. Class-imbalance graph learning methods

---

*Binwu Wang and Yang Wang are the corresponding authors.

enhance the expressiveness of underrepresented classes but often overlook the structural heterogeneity within graphs. Conversely, methods focused on topology imbalance adapt to small graphs but typically ignore imbalanced class distributions. This narrow focus limits their ability to address the complex, intertwined data imbalances commonly found in real-world graph datasets. To remedy this, we propose UniImb, a **Uni**fied **Imb**alance framework for imbalanced graph learning. The core idea of UniImb is to extract semantically shared prototype features via a balanced extraction that equalizes the influence of tail graphs—both minority-class instances and small-scale graphs. These prototypes enrich graph representations and improve robustness to complex, intertwined imbalances.

Specifically, UniImb first encodes multi granularity topological features and applies a personalized graph perturbation module, which injects controlled randomness into graph features and connections with graph dependent intensity. Graphs from smaller or minority classes are augmented with different strengths than larger, frequent ones. We then design a local prior based GNN to encode topological information and obtain graph level representations. Next, we introduce the Dynamic Balanced Prototype (DBP) module, consisting of a set of learnable prototype embeddings that compactly represent shared semantics. Graphs are softly assigned to prototypes based on similarity in a shared embedding space, and prototypes are jointly perceived in a collaborative manner. To regulate this process, we derive a prototype regularizer based on the information bottleneck principle, which encourages prototype activation probabilities to approach a uniform distribution. This allows tail graphs, such as minority class or small scale graphs, to exert comparable influence during prototype learning. Finally, the learned prototypes are used to enhance and regularize underrepresented graph representations, thereby improving discriminative performance on imbalanced datasets.

**Contributions**. ❶ *Unified Insight.* We propose UniImb, a unified framework for imbalanced graph learning that effectively addresses both class imbalance and topological imbalance in graph classification tasks. ❷ *Practical Solution.* UniImb integrates a series of advanced strategies with a Dynamic Balanced Prototype mechanism, effectively enhancing the representational capacity for tail graphs. ❸ *Theoretical Explanation.* Our design incorporates an information bottleneck–driven regularization, which ensures balanced influence from tail classes during training. ❹ *Empirical Study.* Beyond the standard graph classification datasets commonly used in mainstream benchmarks (Qin et al., 2025), we further introduce 3D conformer graph datasets and AirGraph, a large-scale real-world air-pollution graph dataset with naturally occurring long-tailed imbalance, to spur further research and practical progress in this area. We conduct a systematic evaluation on 19 datasets, and the results show that UniImb consistently achieves stable and highly competitive performance in challenging imbalanced graph classification scenarios. Moreover, UniImb is well compatible with a wide range of graph representation backbones and remains efficient while maintaining strong performance.

## 2 RELATED WORK

Graph classification is a foundational task in graph data analysis (Zhang et al., 2018), with GNNs emerging as the dominant architecture for this task (Velickovic et al., 2017; Xu et al., 2018). More recently, inspired by the success of Transformer architectures in natural language processing, researchers have extended these approaches to graph representation learning (Yun et al., 2019; Rampášek et al., 2022; Wang et al., 2024b). However, these methods encounter challenges with graph imbalance distribution (Liu et al., 2025). Imbalanced graph learning has emerged as a dedicated field to address this issue, comprising two main branches: class imbalance and topology imbalance (Qin et al., 2024). Class-imbalanced graph learning methods aim to correct skewed class distributions (Mao et al., 2025). A pioneering work is G$^2$GNN (Wang et al., 2022b), which models each graph instance as a node in a "graph of graphs" and connects instances for data augmentation, thereby improving learning under imbalanced conditions. ImGKB (Tang & Liang, 2023) develops the restricted random walk kernel with the global graph information bottleneck to effectively capture inter-graph supervision for minority-class graphs. Conversely, topological imbalance methods aim to improve performance on small-scale graphs (Qin et al., 2025). For example, SOLT-GNN (Liu et al., 2022c) identifies recurring subgraph patterns to balance learning across graphs of varying sizes. TopoImb (Zhao et al., 2022) models topological groups and employs a modulator to assign greater importance to tail-topology graphs. However, existing methods often struggle in scenarios with complex imbalances. The development of a unified imbalanced graph learning framework remains an unexplored challenge.

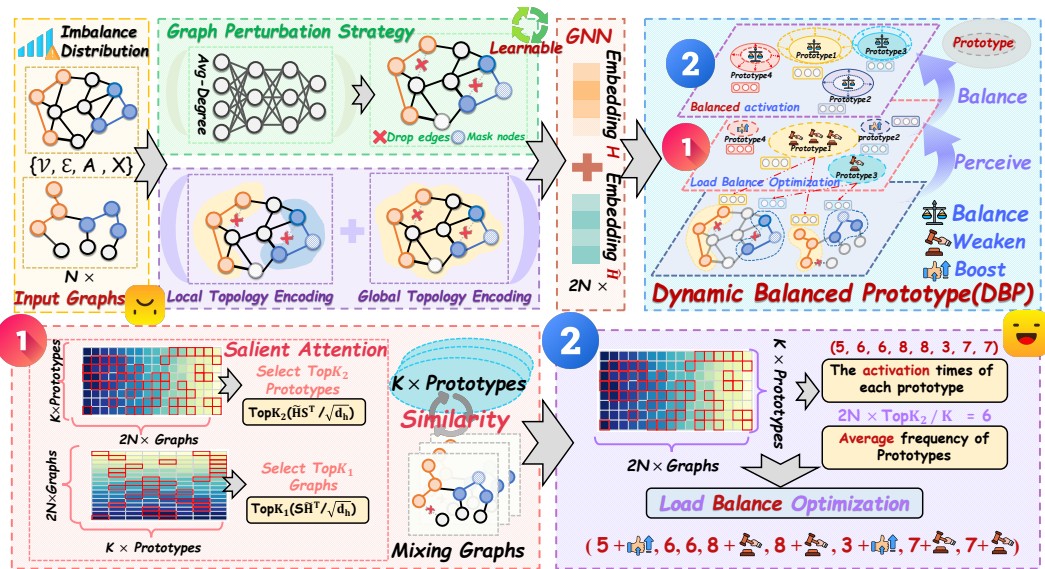

Figure 1: The overall architecture of UniImb which enhances graph representations by extracting prototypical features in an uniform manner.

The detailed discussions of related work on graph representation learning, node-level imbalance learning on graph, and edge-level imbalance learning on graph are given in **Appendix B.**

## 3 PRELIMINARIES

Let $\mathcal{G} = \{\mathcal{V}, \mathcal{E}, \mathbf{A}, \mathbf{X}\}$ be a graph, where $\mathcal{V}$ is the node set with $|\mathcal{V}|$ nodes, $\mathcal{E}$ is the edge set, $\mathbf{A} \in \mathbb{R}^{|\mathcal{V}| \times |\mathcal{V}|}$ is the adjacency matrix, $\mathbf{X} \in \mathbb{R}^{|\mathcal{V}| \times d}$ is the node feature matrix with $d$-dimension.

Given a graph set $G = \{\mathcal{G}_1, \mathcal{G}_2, \ldots, \mathcal{G}_N\}$ with N graph instances where the graph instance $\mathcal{G}_i = \{\mathcal{V}^{\mathcal{G}_i}, \mathcal{E}^{\mathcal{G}_i}, \mathbf{A}^{\mathcal{G}_i}, \mathbf{X}^{\mathcal{G}_i}\}$ and their corresponding label set $\mathbf{Y} = \{y_1, y_2, \ldots, y_N\}$, the goal of the graph-level classification task is to learn a mapping function $\mathcal{F} : \mathcal{G} \to \mathbb{R}^f$ to map any graph to a low-dimensional vector $h \in \mathbb{R}^f$. This representation is subsequently passed through a classifier to produce the predicted label distribution, yielding the final output prediction for the graph instance.

## 4 METHODOLOGY

As illustrated in Figure 1, UniImb frist adopts a personalized perturbation strategy with multi-scale graph topology encoding to enhance individual graph instances. It then employs GNNs to learn graph-level representations. Finally, we propose a dynamic balanced prototype mechanism to extract discriminative prototype features, thereby enhancing the model's capability to represent tail graphs and promoting unbiased learning.

### 4.1 GRAPH REPRESENTATION LEARNING

#### 4.1.1 GRAPH TOPOLOGY ENCODING

**Local Topology Encoding**   We adopt a learnable node position for learning local information (Dwivedi et al., 2021), which leverages random walks to perceive local structural information around each node. Specifically, for a graph $\mathcal{G}_i$, we first calculate its random walk operator $\mathbf{M}^{\mathcal{G}_i} = \mathcal{D}^{-1}\mathbf{A}^{\mathcal{G}_i}$, where $\mathbf{A}^{\mathcal{G}_i}$ is the adjacency matrix of $\mathcal{G}_i$, $\mathcal{D}$ is the corresponding degree matrix and $\mathcal{D}_{i,i} = \sum_j \mathbf{A}^{\mathcal{G}_i}_{i,j}$. With z-steps of random walk, for the node $v_j$ of graph $\mathcal{G}_i$, we can get its local topology encodings:

$$\mathbf{LE}^{\mathcal{G}_i}_j = [(\mathbf{M}^{\mathcal{G}_i})_{j,j}, (\mathbf{M}^{\mathcal{G}_i})^2_{j,j}, (\mathbf{M}^{\mathcal{G}_i})^3_{j,j}, \ldots, (\mathbf{M}^{\mathcal{G}_i})^z_{j,j}] \in \mathbb{R}^z \tag{1}$$

here we adopt a low-complexity usage of the random walk matrix by considering only the landing probability of the node $v_j$ to itself, i.e., $(\mathbf{M}^{\mathcal{G}_i})_{j,j}$. And we can get the initial node position encodings

$\mathbf{LE}^{\mathcal{G}_i} \in \mathbb{R}^{|\mathcal{V}^{\mathcal{G}_i}| \times z}$. Subsequently, The position encodings are integrated with GIN for end-to-end updates, enabling the capture of accurate graph topology information.

**Global Topology Encoding.** The Laplacian matrix of a graph contains rich information about the graph's global structural properties, such as subgraph frequency and connectivity (Dong et al., 2016). To this end, we introduce a global topology encoding strategy to learn this information. Specifically, for a graph $\mathcal{G}_i$, we first extract the eigenvalues and eigenvectors of its laplacian matrix $\mathbf{L}^{\mathcal{G}_i} = \mathcal{D}^{\mathcal{G}_i} - \mathbf{A}^{\mathcal{G}_i}$, which is denoted as $[\lambda_1, \lambda_2, \cdots, \lambda_z]$ and $[h_1, h_2, \cdots, h_z] \in \mathbb{R}^{|\mathcal{V}^{\mathcal{G}_i}| \times z}$. Then we can get the global topology encoding as follows,

$$\mathbf{GE}^{\mathcal{G}_i} = \varphi\left([\ell(h_i, \lambda_i) + \ell(-h_i, \lambda_i)]_{i=1}^z\right) \in \mathbb{R}^{|\mathcal{V}^{\mathcal{G}_i}| \times d_h} \tag{2}$$

where $\varphi(\cdot)$ and $\ell(\cdot)$ are permutation-invariant neural networks (e.g., MLP layer) used to map the input to a higher-dimensional hidden space.

### 4.1.2 Personalized Graph Perturbation Strategy

Graph perturbation strategies, such as edge drop and feature mask (Wang et al., 2022a), can effectively improve the generalization performance of GNNs. However, traditional methods apply uniform perturbations to all graph samples, which can increase the difficulty of learning tail graphs in imbalanced scenarios. To address this issue, we propose a personalized graph perturbation strategy that customizes learnable perturbations based on the unique characteristics of each graph.

**Edge Drop.** For the graph $\mathcal{G}_i$ with $|\mathcal{V}^{\mathcal{G}_i}|$ nodes and $|\mathcal{E}^{\mathcal{G}_i}|$ edges, we first calculate the average degree of the graph: $d^{\mathcal{G}_i} = \frac{2|\mathcal{E}^{\mathcal{G}_i}|}{|\mathcal{V}^{\mathcal{G}_i}|} \in \mathbb{R}$. Then, we create personalized edge dropout distribution $\mathcal{B}(a_e^{\mathcal{G}_i})$ where $a_e^{\mathcal{G}_i} = \sigma(\mathrm{MLP}(d^{\mathcal{G}_i})) \in \mathbb{R}$ is learned by one MLP layer. And we sample the edge drop matrix $\mathbf{M}_e^{\mathcal{G}_i}$, where any value $m$ in $\mathbf{M}_e^{\mathcal{G}_i}$ follows Bernoulli distribution $m_e \sim \mathcal{B}(a_e^{\mathcal{G}_i})$, and $m_e = 0$ means that the corresponding edge is dropped. Finally, we apply the learned perturbations to the adjacency matrix: $\widetilde{\mathbf{A}}^{\mathcal{G}_i} = \mathbf{A}^{\mathcal{G}_i} \odot \mathbf{M}_e^{\mathcal{G}_i}$, where $\odot$ means Hadamard product.

**Feature Mask.** We use a similar operation to personalized feature mask distribution $\mathcal{B}(\beta_n^{\mathcal{G}_i})$; for every node $v_j$ in graph $\mathcal{G}_i$, we also sample a feature mask $m_n^j \in \{0, 1\} \sim \mathcal{B}(\beta_n^{\mathcal{G}_i})$, if $m_n^j$ is equal to 0, we set the corresponding node feature value in $\mathbf{X}_j^{\mathcal{G}_i}$ to 0. And we can get personalized feature mask matrix $\mathbf{M}_f^{\mathcal{G}_i}$ and apply it to node feature matrix: $\widetilde{\mathbf{X}}^{\mathcal{G}_i} = \mathbf{X}^{\mathcal{G}_i} \odot \mathbf{M}_f^{\mathcal{G}_i}$.

### 4.1.3 Graph Representation Learning via GNN

We leverage the powerful graph representation capabilities of GNNs to generate graph-level representations (Xu et al., 2018; Khoshraftar & An, 2024). Specifically, we use an MLP layer to map each graph $\mathcal{G}_i$ into a high-dimensional space, and the output is then integrated with the global topology encoding as the initial output of the GNN, denoted as $\widetilde{\mathbf{X}}^{(0),\mathcal{G}_i} \in \mathbb{R}^{|\mathcal{V}^{\mathcal{G}_i}| \times d_h}$.

For local node position encodings $\mathbf{LE}^{\mathcal{G}_i} \in \mathbb{R}^{|\mathcal{V}^{\mathcal{G}_i}| \times z}$, we also take it into one MLP layer, and the output is denoted as $\mathbf{LE}^{0,\mathcal{G}_i} \in \mathbb{R}^{|\mathcal{V}^{\mathcal{G}_i}| \times d_h}$. This transformed encoding, together with $\widetilde{\mathbf{X}}^{0,\mathcal{G}_i}$, is fed into the GNN layer for graph representation learning. In this work, we use GIN (Xu et al.) as the primary backbone to exemplify our approach and validate the method across several other backbones, as shown in Table 4. For node $v_j$, we can aggregate features from the neighborhood as follows,

$$\widetilde{\mathbf{X}}_j^{(\ell+1),\mathcal{G}_i} = \mathrm{MLP}^{(\ell)}\left((1+\varepsilon)\begin{bmatrix}\widetilde{\mathbf{X}}_j^{(\ell),\mathcal{G}_i} \\ \mathbf{LE}_j^{(\ell),\mathcal{G}_i}\end{bmatrix} + \sum_{u \in \mathcal{N}(v_j)}\begin{bmatrix}\widetilde{\mathbf{X}}_u^{(\ell),\mathcal{G}_i} \\ \mathbf{LE}_u^{(\ell),\mathcal{G}_i}\end{bmatrix}\right), \forall \ell \in \{1, 2, \ldots, \mathrm{L}\} \tag{3}$$

where $[\cdot]$ means the concatenation of two matrices in the column dimension. $\varepsilon$ is a harmonic coefficient, $\mathcal{N}(v_j)$ represents the neighbors of node $v_j$ in $\widetilde{\mathbf{A}}^{\mathcal{G}_i}$ of the graph $\mathcal{G}_i$. And we also employ an independent GNN layer to update the local topology encoding, thereby capturing deeper graph topology information, which is denoted as follows,

$$\mathbf{LE}_j^{(\ell+1),\mathcal{G}_i} = \mathrm{MLP}^{(\ell)}\left((1+\varepsilon)\mathbf{LE}_j^{(\ell),\mathcal{G}_i} + \sum_{u \in \mathcal{N}_{(v_j)}} \mathbf{LE}_u^{(\ell),\mathcal{G}_i}\right) \tag{4}$$

After $L$ graph convolution layers, each node aggregates information from its neighborhoods within a radius of up to $L$ hops. Subsequently, a readout function is applied to aggregate the resulting node representations into a unified graph representation $\mathbf{H}_{\mathcal{G}_i}$ for the graph $\mathcal{G}_i$ as follows,

$$\mathbf{H}_{\mathcal{G}_i} = \text{READOUT}(\{\widetilde{\mathbf{X}}_j^{(\ell+1),\mathcal{G}_i} \mid v_j \in \mathcal{V}^{\mathcal{G}_i}\}) \in \mathbb{R}^{1 \times d_h} \tag{5}$$

Finally, we can obtain the graph representations of N graphs, denoted as $\mathbf{H} \in \mathbb{R}^{\text{N} \times d_h}$. To enhance the feature diversity of tail graphs, we employ a Feature Mixup strategy (Ju et al., 2025). This approach involves randomly rearranging the order of the representation vectors of graph instances, thereby generating a new matrix, denoted as $\widehat{\mathbf{H}} \in \mathbb{R}^{2\text{N} \times d_h}$. The details are provided in **Appendix C.0.1**.

## 4.2 Dynamic Balanced Prototype Based on Information Bottleneck

We further propose a **Dynamic Balanced Prototype (DBP)** strategy, which unbiasedly extracts representative prototype features from graph data. These prototypes are universally applicable, enabling them to effectively enhance the representational capacity of graphs—particularly for under-represented tail graph. Specifically, prototypes are defined as a set of learnable embeddings, denoted by $\mathbf{S} = [s_1, s_2, \ldots, s_{\text{K}}] \in \mathbb{R}^{\text{K} \times d_h}$, where the hyperparameter K represents the number of prototypes and satisfies $\text{K} \ll \text{N}$.

❶ **Prototype Perception.** Prototypes are perceived from the graph data based on the attention coefficients computed between the prototypes and graph representations, as follows:

$$\widetilde{\mathbf{H}}_\mathbf{S} = \text{Softmax}\left(\text{TopK}_1\left(\mathbf{S}\widetilde{\mathbf{H}}^\top / \sqrt{d_h}\right)\right)\widetilde{\mathbf{H}}\mathbf{W}_v \in \mathbb{R}^{\text{K} \times d_h} \tag{6}$$

where $\mathbf{W}_v \in \mathbb{R}^{d_h \times d_h}$ is the learnable parameters. Next, we utilize prototypes to enhance the representation learning of the graph.

❷ **Prototype Balance.** We employ an attention mechanism to compute the affinity between each graph and the prototypes, thereby selecting the most relevant prototype to enhance its representational capacity. Specifically, this process is formulated as follows:

$$\widehat{\mathbf{H}} = \text{Sigmoid}\left(\text{TopK}_2\left(\widetilde{\mathbf{H}}\mathbf{S}^\top / \sqrt{d_h}\right) + \boldsymbol{\gamma}\right)\widetilde{\mathbf{H}}_\mathbf{S}, \tag{7}$$

where $\widehat{\mathbf{H}} \in \mathbb{R}^{2\text{N} \times d_h}$ represents the enhanced representation. Here, we employ a sigmoid function together with a learnable vector $\boldsymbol{\gamma} \in \mathbb{R}^{1 \times \text{K}}$ to generate discriminative similarity scores, rather than producing values concentrated around the mid-range.

During the iterative training process involving two alternating stages, the model learns to extract prototype features. However, these learned prototypes may become biased due to imbalanced graph data distributions, which can negatively affect their generalization capability. Ideally, we should reduce the dependency of the prototypes on the input data in order to enhance both robustness and generalization performance. To this end, we introduce the information bottleneck theory to guide the learning of more generalized prototype representation.

**Theory 1. Information Bottleneck.** For the training graphs G, prototype features $\mathbf{S}$, and labels $\mathbf{Y}$, the objective of the information bottleneck theory is expressed as minimizing the following objective function (Tishby et al., 2000):

$$\min \text{I}(\mathbf{S}; \text{G}) - \beta \text{I}(\mathbf{S}; \mathbf{Y}) \tag{8}$$

where $\beta$ is the Lagrange multiplier. We can optimize the mutual information between the hidden feature and the labels $\text{I}(\mathbf{S}; \mathbf{Y})$ through supervised learning. Thus, the above objective is simplified to minimizing the mutual information between the training graph set G and the prototype feature $\mathbf{S}$, i.e., reducing the redundancy of imbalanced input information. For more details, please refer to **Appendix D.1**.

**Proposition 1. Prototype Load Balancing.** Let $\mathbf{P} = \{p_1, p_2, \cdots, p_{\text{K}}\}$ denote the activation [1] probability distribution of the K prototypes over the training graph set G and $\mathbf{U} = \{u_1, u_2, \cdots, u_{\text{K}}\}$

---

[1]A prototype is considered activated once if its attention coefficient on any graph is non-zero

Table 1: Macro-F1 and Micro-F1 scores on *class imbalance* datasets with *extreme imbalance degree* under *different* **U** *distributions*. The best results are marked and the runner-ups are underlined .

| | PROTEINS | | D&D | | NCI1 | | COLLAB | |
|---|---|---|---|---|---|---|---|---|
| **Distribution** | Macro-F1 | Micro-F1 | Macro-F1 | Micro-F1 | Macro-F1 | Micro-F1 | Macro-F1 | Micro-F1 |
| Zipf (Axtell, 2001) | 67.80 ± 3.71 | 73.29 ± 4.00 | 44.05 ± 2.29 | 79.02 ± 7.38 | 65.69 ± 3.11 | 79.78 ± 1.58 | 69.25 ± 10.26 | 69.79 ± 10.08 |
| Exponential (Marshall & Olkin, 1967) | 66.45 ± 3.34 | 71.99 ± 4.94 | 44.99 ± 2.95 | 82.31 ± 9.36 | 65.03 ± 2.41 | 79.85 ± 2.09 | 73.51 ± 6.73 | 74.47 ± 6.09 |
| Poisson (Consul & Jain, 1973) | 68.49 ± 1.70 | 73.94 ± 3.40 | 41.62 ± 3.72 | 71.98 ± 10.64 | 67.67 ± 5.50 | 80.60 ± 5.09 | 75.56 ± 2.03 | 75.85 ± 2.18 |
| **Uniform** (Kuipers & Niederreiter, 2012) | **70.44** ± 4.72 | **74.50** ± 4.99 | **46.63** ± 3.42 | **83.60** ± 6.50 | **68.30** ± 5.19 | **80.68** ± 4.22 | **75.73** ± 2.52 | **76.34** ± 2.60 |

represent an empirical distribution. Minimizing the mutual information between **S** and G is equivalent to the following objective:

$$\min I(\mathbf{S}; \mathbf{G}) = \sum_{\mathcal{G},s} p(\mathcal{G})p(s|\mathcal{G}) \log \frac{p(s|\mathcal{G})}{p(s)} \Rightarrow \min KL(\mathbf{P} \| \mathbf{U}) \approx \min \frac{1}{2} \sum_{k=1}^{K} (p_k - u_k)^2 \quad (9)$$

where $KL(\cdot)$ means Kullback-Leibler Divergence. The detailed proof is provided in **Appendix D.2**.

**Proposition 2. Prototype-balancing Optimization.** For 2N graphs, each graph hits $\text{TopK}_2$ prototypes, and we have a total hit count of $\mathcal{K} = 2N \times \text{TopK}_2$. The actual number of hits for each prototype is $\mathbb{N} = \{n_1, n_2, \cdots, n_K\}$, where $n_k = p_k \mathcal{K}$, $k \in \{1, \cdots, K\}$. Subsequently, the regularization term minimizes the discrepancy between the actual attention activation $n_k$ and the prior distribution $u_k$. To achieve a smoother and more effective optimization of the attention generation process, we introduce an intermediate modulation term $\eta$:

$$\widehat{\mathbf{H}} = \text{Sigmoid}\left(\text{TopK}_2\left(\widetilde{\mathbf{H}}\mathbf{S}^\top / \sqrt{d_h} + \eta\right) + \boldsymbol{\gamma}\right) \widetilde{\mathbf{H}}_{\mathbf{S}}, \quad (10)$$

where $\eta \in \mathbb{R}^K$ directly affects the attention generation process. Nevertheless, the TopK operation is non-differentiable in practice. To address this, we introduce a specific constraint loss $\mathcal{L}_\mathcal{M}$ that iteratively models the prototype activation process during training.

$$\mathcal{L}_\mathcal{M} = \frac{1}{2} \sum_{k=1}^{K} \left| n_k - \frac{2*N*\text{TopK}_2}{\frac{1}{u_k}} \right|^2 = \frac{1}{2} \sum_{k=1}^{K} \left| \eta + \text{StopGrad}(n_k - \eta) - \frac{2*N*\text{TopK}_2}{\frac{1}{u_k}} \right|^2 \quad (11)$$

where $\text{StopGrad}(\cdot)$ denotes the stop-gradient operator (Chen & He, 2021), which *behaves as identity in the forward pass but yields zero in the backward pass*, and this is a common approach used for non-differentiable operations. Building on Eq. 11, we further derive the optimization objective for $\eta$ with the learning rate $\varphi$ as follows:

$$\eta \leftarrow \eta - \varphi \nabla \mathcal{L}_\mathcal{M} = \eta - \varphi \, \text{sgn}\left(n_k - 2*N*\text{TopK}_2 *u_k\right), \quad (12)$$

To identify an appropriate prior distribution, we conduct experiments to evaluate model performance under four different distributions of **U**: Zipf $\left(u_k = \frac{\frac{1}{k^s}}{\sum_{j=1}^{K} \frac{1}{j^s}}\right)$, Exponential $\left(u_k = \frac{e^{-\lambda k}}{\sum_{j=1}^{K} e^{-\lambda j}}\right)$, Poisson $\left(u_k = \frac{\lambda^k e^{-\lambda}}{k! \sum_{j=1}^{K} \frac{\lambda^j e^{-\lambda}}{j!}}\right)$, and Uniform $\left(u_k = \frac{1}{K}\right)$. The complete analysis and more details are provided in **Appendix D.3**. The experimental setup is described in **Section 5.1**. As shown in Table 1, we can observe that when the empirical distribution **U** follows a uniform distribution, the model achieves the highest accuracy. In other words, whenever the activation count of any prototype exceeds the average level, $\eta$ imposes a penalty to reduce that prototype's activation priority in the subsequent optimization step, thereby enforcing load balancing. This mechanism effectively ensures balanced influence from tail graphs during learning.

Finally, we input the representations of the N input graph instances (i.e., the first N rows of $\widehat{\mathbf{H}}$) into a decoder composed of two layers of MLP for generating the predicted class labels $\widehat{\mathbf{Y}}$.

## 5 EXPERIMENTS

### 5.1 EXPERIMENTAL SETUP

**Datasets.** We evaluate the model's performance on **19** datasets under class imbalance and topological imbalance scenarios. Following existing protocols (Qin et al., 2024), the datasets were processed

into three levels of imbalance (low, medium, extreme). Due to space limitations, we primarily report results on six highly imbalanced datasets in the main text: D&D (Shervashidze et al., 2011), NCI1 (Wale et al., 2008), COLLAB (Leskovec et al., 2005), PROTEINS, REDDIT-B, and IMDB-MULTI (Yanardag & Vishwanathan, 2015). We also introduce a large-scale dataset, Airgraph (described below), and three 3D conformational graph datasets MoleculeNet (Wu et al., 2018): BBBP for blood-brain barrier permeability prediction, BACE for $\beta$-secretase inhibition prediction, and HIV for antiviral activity prediction.

**Large-Scale AirGraph Dataset.** comprises hourly PM2.5 measurements from the China National Environmental Monitoring Center collected at 1,341 stations across 28 provinces in mainland China between 2021 and 2023. For each province the data are represented as a graph in which monitoring stations are nodes and stations are connected based on distance. Each node encodes the preceding 24 hours of PM2.5 readings, and each graph is labeled by the spatially averaged PM2.5 over the subsequent 24 hours, categorized into three levels: low, medium, and high pollution. With 30,660 graph instances, AirGraph is substantially larger than most existing graph classification datasets and exhibits a natural class imbalance: the high pollution category accounts for only 6.86% of instances. More details are provided in **Appendix G**.

**Baselines.** We compare our model against four categories of baselines, encompassing a total of 23 representative baselines: ❶ *Classic Graph Classification Models*: GIN (Xu et al.), GCN (Kipf & Welling, 2016), GraphSAGE (Hamilton et al., 2017), InfoGraph (Sun et al., 2019), and GraphCL (You et al., 2020); ❷ *Graph Transformer Models (GTs)*: GraphGPS (Rampášek et al., 2022), Exphormer (Shirzad et al., 2023), and Graph-Mamba (Wang et al., 2024b); ❸ *3D Graph Models*: SchNet (Schütt et al., 2017), DimeNet (Gasteiger et al., 2020), and SphereNet (Liu et al., 2022b); ❹ *Class-imbalance Methods*: upsampling (Kubat et al., 1997) and reweighting strategies (Yuan & Ma, 2012) applied to various backbones, as well as G$^2$GNN with edge and node masking, ImGKB (Tang & Liang, 2023), and DataDec (Zhang et al., 2023); ❺ *Topology-imbalance Methods*: SOLT-GNN (Liu et al., 2022c), ImbGNN (Xu et al., 2024), and TopoImb (Zhao et al., 2022).

**Experiment Setting.** We implement our proposed model on an 40GB NVIDIA A100 GPU with Pytorch. For the class-imbalanced graph classification task, we split the dataset into 25% for training and 25% for validation, and the remaining data is used for testing. For the topological-imbalanced graph classification task, we split the dataset into 10% for training and 10% for validation, and the remaining data is used for testing. We use the Adam optimizer (Kingma & Ba, 2014) with learning rate 0.001. All experiments are repeated 20 times, and we report the average and standard deviation. For the REDDIT-B, IMDB-MULTI, and COLLAB datasets, we employ one-hot encodings of node degrees as node features (Sun et al., 2019; You et al., 2020). We use widely metrics in the graph imbalance tasks including Macro-F1 and Micro-F1 (Xia et al., 2014; Lipton et al., 2014). And $\varphi$ is set to 0.001. The number of prototypes K is set to $\{16, 16, 24, 24, 24, 32\}$ on six datasets. We stack L = 5 layers of GNN. Other key hyperparameters of each dataset are shown in Table 13.

Table 2: Performance on *class imbalance* datasets with *extreme imbalance degree*. The best results are marked and the runner-ups are underlined. We report the average and standard deviation over 20 runs. Numbers marked with * indicate that the improvement is statistically significant compared with the best baseline (Wilcoxon Signed-Rank Test with *p-value < 0.05*).

| Method | Backbone | PROTEINS | | D&D | | NCI1 | | REDDIT-B | | COLLAB | | IMDB-MULTI | |
|---|---|---|---|---|---|---|---|---|---|---|---|---|---|
| | | Macro-F1 | Micro-F1 | Macro-F1 | Micro-F1 | Macro-F1 | Micro-F1 | Macro-F1 | Micro-F1 | Macro-F1 | Micro-F1 | Macro-F1 | Micro-F1 |
| Classic | GIN | 25.33 ± 7.53 | 28.50 ± 5.82 | 9.99 ± 7.44 | 11.88 ± 9.49 | 18.24 ± 7.58 | 18.94 ± 7.12 | 33.19 ± 14.26 | 36.02 ± 17.38 | 32.58 ± 3.66 | 57.31 ± 4.12 | 13.25 ± 6.19 | 14.92 ± 5.43 |
| | InfoGraph | 35.91 ± 7.58 | 36.81 ± 6.51 | 21.41 ± 4.51 | 27.68 ± 7.52 | 33.09 ± 3.30 | 34.03 ± 3.68 | 57.67 ± 3.80 | 67.10 ± 4.91 | 43.48 ± 4.29 | 59.10 ± 4.88 | 17.28 ± 7.28 | 29.18 ± 4.47 |
| | GraphCL | 40.86 ± 6.94 | 41.24 ± 6.38 | 21.02 ± 3.05 | 26.80 ± 4.95 | 31.02 ± 2.69 | 31.62 ± 3.05 | 53.40 ± 4.06 | 62.19 ± 5.68 | 45.02 ± 5.61 | 60.22 ± 3.47 | 16.30 ± 9.22 | 32.18 ± 8.90 |
| GTs | GraphGPS | 25.79 ± 7.05 | 28.71 ± 5.46 | 10.12 ± 4.41 | 11.97 ± 3.91 | 14.94 ± 2.41 | 15.62 ± 2.07 | 11.68 ± 7.76 | 12.71 ± 8.13 | 25.58 ± 12.93 | 39.92 ± 13.06 | 14.20 ± 5.61 | 28.54 ± 13.78 |
| | Exphormer | 25.52 ± 4.79 | 28.38 ± 3.57 | 9.79 ± 4.18 | 10.85 ± 4.28 | 14.56 ± 3.92 | 15.36 ± 3.60 | 22.68 ± 10.79 | 27.33 ± 21.65 | 32.61 ± 17.44 | 42.02 ± 15.44 | 20.81 ± 5.43 | 28.14 ± 8.30 |
| | Garph-Mamba | 31.12 ± 5.10 | 32.79 ± 4.02 | 4.99 ± 7.93 | 6.12 ± 10.36 | 14.11 ± 3.26 | 14.94 ± 3.82 | 15.27 ± 12.46 | 17.02 ± 14.71 | 42.53 ± 11.15 | 50.63 ± 5.38 | 16.89 ± 4.57 | 28.69 ± 12.47 |
| up-sampling | GIN | 65.64 ± 2.67 | 71.55 ± 3.19 | 41.15 ± 3.74 | 70.56 ± 10.28 | 59.19 ± 4.39 | 71.80 ± 7.02 | 66.71 ± 3.92 | 83.00 ± 5.18 | 64.30 ± 2.67 | 66.10 ± 3.28 | 22.27 ± 10.01 | 38.32 ± 10.04 |
| | InfoGraph | 62.68 ± 2.70 | 66.02 ± 3.18 | 41.55 ± 2.32 | 71.34 ± 6.76 | 53.38 ± 1.88 | 62.20 ± 2.63 | 67.01 ± 3.34 | 78.68 ± 3.71 | 63.28 ± 2.90 | 65.14 ± 3.29 | 21.79 ± 6.68 | 37.29 ± 7.02 |
| | GraphCL | 64.21 ± 2.53 | 65.76 ± 2.61 | 38.96 ± 3.01 | 64.23 ± 8.10 | 49.92 ± 2.15 | 58.29 ± 3.30 | 62.01 ± 3.97 | 75.84 ± 3.98 | 64.57 ± 5.20 | 66.79 ± 4.11 | 23.62 ± 6.91 | 40.29 ± 6.90 |
| re-weight | GIN | 54.54 ± 6.29 | 55.77 ± 7.11 | 28.49 ± 5.92 | 40.79 ± 11.84 | 36.84 ± 8.46 | 39.19 ± 10.05 | 45.17 ± 8.46 | 51.92 ± 12.29 | 57.83 ± 3.03 | 60.09 ± 4.59 | 22.07 ± 11.13 | 36.69 ± 11.14 |
| | InfoGraph | 65.73 ± 3.10 | 69.60 ± 3.68 | 41.92 ± 2.28 | 72.43 ± 6.63 | 53.05 ± 1.12 | 62.45 ± 1.89 | 55.79 ± 3.38 | 77.35 ± 3.96 | 62.22 ± 3.90 | 64.48 ± 3.14 | 21.16 ± 6.79 | 38.02 ± 5.70 |
| | GraphCL | 63.46 ± 2.42 | 64.97 ± 2.41 | 40.29 ± 3.31 | 67.96 ± 8.98 | 50.05 ± 2.09 | 58.18 ± 3.08 | 62.79 ± 6.93 | 76.15 ± 9.15 | 63.18 ± 4.55 | 65.29 ± 3.87 | 22.48 ± 6.82 | 39.57 ± 5.89 |
| G$^2$GNN | remove edge | 67.70* ± 2.96 | 73.10 ± 4.05 | 43.25 ± 3.91 | 77.03 ± 9.98 | 63.60 ± 1.57 | 72.97 ± 1.81 | 68.39 ± 2.97 | 86.35 ± 2.27 | 38.93 ± 3.22 | 54.98 ± 4.28 | 20.67 ± 9.88 | 36.89 ± 11.73 |
| | mask node | 67.39 ± 2.99 | 73.30 ± 4.19 | 43.93 ± 3.46 | 79.03 ± 10.78 | 64.78 ± 2.86 | 74.91 ± 2.14 | 67.52 ± 2.60 | 85.43 ± 1.80 | 37.63 ± 5.19 | 53.92 ± 6.37 | 21.54 ± 9.49 | 35.78 ± 11.08 |
| TopoImb | / | 53.95 ± 6.68 | 56.00 ± 7.88 | 7.72 ± 5.68 | 9.47 ± 4.26 | 16.41 ± 5.19 | 17.14 ± 5.68 | 10.38 ± 2.94 | 11.20 ± 2.69 | 19.24 ± 0.02 | 40.71 ± 0.05 | 9.37 ± 1.51 | 15.24 ± 0.90 |
| DataDec | dynamic sparsity | 29.48 ± 3.98 | 31.25 ± 2.98 | 15.79 ± 2.38 | 18.86 ± 3.47 | 18.14 ± 3.86 | 18.52 ± 3.64 | 56.11 ± 4.65 | 65.35 ± 6.16 | 41.73 ± 1.20 | 57.27 ± 0.65 | 11.08 ± 1.15 | 16.29 ± 0.72 |
| ImGKB | | 53.99 ± 7.22 | 55.31 ± 8.17 | 31.15 ± 6.29 | 46.31 ± 11.72 | 32.93 ± 5.46 | 34.85 ± 7.14 | 11.00 ± 8.34 | 14.00 ± 1.34 | 18.84 ± 1.96 | 39.58 ± 4.90 | 16.53 ± 5.23 | 34.12 ± 12.71 |
| ImbGNN | / | 67.69 ± 2.95 | 73.24 ± 3.85 | 46.06 ± 7.23 | 83.24 ± 19.03 | 65.48 ± 3.39 | 74.58 ± 5.49 | 68.36 ± 8.10 | 86.56 ± 4.77 | 56.62 ± 4.28 | 62.79 ± 3.83 | 17.52 ± 8.98 | 34.54 ± 8.24 |
| UniImb | / | 70.44* ± 4.72 | 74.50 ± 4.99 | 46.63* ± 3.42 | 83.60* ± 6.50 | 68.30* ± 5.19 | 80.68* ± 4.22 | 76.24* ± 4.09 | 88.82* ± 2.93 | 75.73* ± 2.52 | 76.34* ± 2.60 | 33.45* ± 7.83 | 45.72* ± 4.87 |
| Promotion | | 4.05%↑ | 1.64%↑ | 1.24%↑ | 0.43%↑ | 4.31%↑ | 7.70%↑ | 11.48%↑ | 2.61%↑ | 17.28%↑ | 14.30%↑ | 41.62%↑ | 13.48%↑ |

Table 3: Performance comparison on *topological imbalance* datasets with *extreme imbalance degree*.

| Method | PROTEINS | | D&D | | NCI1 | | REDDIT-B | | COLLAB | | IMDB-MULTI | |
|---|---|---|---|---|---|---|---|---|---|---|---|---|
| | Macro-F1 | Micro-F1 | Macro-F1 | Micro-F1 | Macro-F1 | Micro-F1 | Macro-F1 | Micro-F1 | Macro-F1 | Micro-F1 | Macro-F1 | Micro-F1 |
| GIN | 53.48 ± 2.03 | 58.00 ± 4.19 | 57.98 ± 5.51 | 60.68 ± 6.89 | 61.60 ± 2.20 | 61.84 ± 2.29 | 66.60 ± 2.27 | 67.41 ± 2.23 | 64.92 ± 2.18 | 67.05 ± 4.26 | 20.80 ± 4.91 | 34.73 ± 2.16 |
| InfoGraph | 55.31 ± 3.30 | 60.68 ± 3.52 | 59.46 ± 4.03 | 64.22 ± 4.11 | 62.25 ± 1.53 | 57.99 ± 2.01 | 69.56 ± 3.46 | 69.63 ± 4.28 | 63.28 ± 1.68 | 66.20 ± 1.33 | 33.36 ± 1.60 | 36.42 ± 1.10 |
| GraphCL | 57.62 ± 5.50 | 61.91 ± 3.84 | 62.01 ± 5.18 | 65.29 ± 2.33 | 63.62 ± 5.43 | 56.93 ± 4.75 | 67.25 ± 7.92 | 68.31 ± 6.36 | 58.61 ± 4.16 | 59.88 ± 4.01 | 30.39 ± 2.53 | 33.90 ± 1.57 |
| GraphGPS | 65.54 ± 4.22 | 69.26 ± 2.48 | 63.33 ± 12.97 | 63.85 ± 2.08 | 62.96 ± 3.51 | 64.02 ± 2.40 | 66.16 ± 4.19 | 68.42 ± 5.32 | 24.11 ± 8.74 | 56.65 ± 2.81 | 16.87 ± 0.53 | 34.49 ± 0.62 |
| Exphormer | 64.01 ± 2.18 | 67.33 ± 2.78 | 59.43 ± 8.06 | 60.26 ± 5.92 | 62.16 ± 3.19 | 63.23 ± 7.31 | 66.48 ± 14.59 | 67.81 ± 7.45 | 21.03 ± 7.89 | 36.71 ± 15.27 | 25.52 ± 5.03 | 33.60 ± 2.70 |
| Graph-Mamba | 68.46 ± 3.91 | 72.09 ± 3.16 | 43.86 ± 9.73 | 54.45 ± 9.53 | 63.09 ± 2.82 | 63.63 ± 2.14 | 64.81 ± 12.47 | 67.06 ± 8.99 | 13.64 ± 5.97 | 27.37 ± 14.82 | 17.63 ± 3.07 | 33.21 ± 0.52 |
| TopoImb | 44.79 ± 14.19 | 54.89 ± 13.58 | 63.97 ± 2.78 | 64.16 ± 2.96 | 63.57 ± 1.69 | 64.17 ± 1.33 | 68.41 ± 5.34 | 69.14 ± 4.83 | 65.65 ± 6.03 | 67.52 ± 0.77 | 32.59 ± 5.45 | 40.63 ± 3.12 |
| ImbGNN | 66.87 ± 2.59 | 67.07 ± 2.75 | 68.67 ± 6.33 | 70.71 ± 6.99 | 61.24 ± 1.72 | 61.66 ± 1.17 | 65.19 ± 6.11 | 66.96 ± 5.03 | 51.35 ± 0.12 | 52.31 ± 0.14 | 23.85 ± 5.05 | 33.91 ± 1.78 |
| SOLT-GNN | 70.70 ± 2.20 | 72.14 ± 2.18 | 58.50 ± 10.48 | 64.97 ± 3.24 | 61.99 ± 1.44 | 62.13 ± 1.44 | 54.80 ± 3.23 | 60.24 ± 2.21 | 64.68 ± 2.21 | 67.12 ± 3.28 | 33.55 ± 4.77 | 39.69 ± 2.00 |
| UniImb | 71.32 ± 1.88 | 74.89* ± 1.12 | 74.49* ± 1.13 | 76.73* ± 1.04 | 64.99* ± 9.58 | 65.76* ± 7.24 | 77.14* ± 10.05 | 78.22* ± 7.41 | 73.51* ± 1.48 | 75.54* ± 1.63 | 40.45* ± 3.91 | 46.67* ± 2.31 |
| Promotion | 0.88%↑ | 3.80%↑ | 8.48%↑ | 8.51%↑ | 2.15%↑ | 2.48%↑ | 10.90%↑ | 12.34%↑ | 11.97%↑ | 11.88%↑ | 20.57%↑ | 14.87%↑ |

## 5.2 PERFORMANCE ANALYSIS ON IMBALANCE GRAPH CLASSIFICATION

Owing to space limitations, we present the experiment results on all datasets under medium and low imbalance degrees in **Appendix I.1.1** and **I.3.1**.

❶ **Class Imbalance Graph Classification.** As shown in Table 2, classic graph representation models perform the worst due to their inability to address class imbalance. While Graph Transformers (GTs) augment GNN with global attention mechanisms to capture long-range dependencies, they remain ineffective in mitigating the issue of class label imbalance. Upsampling methods require resampling tail classes, which may lead to overfitting on tail features and result in a loss of generalization. TopoImb and ImGKB exhibit highly unstable performance, performing even worse than naive models on most datasets (Qin et al., 2024). *DataDec inadvertently introduces data leakage by using test samples to fine-tune the SVM classifier. To ensure a fair evaluation, we restrict all fine-tuning to the training set. Under this setting, DataDec shows significantly degraded performance.* $G^2$GNN and ImbGNN utilize the $G^2$G framework for data augmentation. However, they underperform compared to simple upsampling baselines on multi-label datasets such as COLLAB. Our model achieves the best performance across all metrics.

❷ **Topological Imbalance Graph Classification.** As shown in Table 3, GIN consistently achieves the worst performance across all topologically imbalanced datasets. GraphCL further improves the performance through contrastive learning with graph augmentations, yielding significant gains over GIN. Graph Transformer models, including GraphGPS, Exphormer, and Graph-Mamba, build upon GNN by integrating global attention mechanisms, which help address topological imbalance. TopoImb alleviates substructure imbalance by incorporating topology-aware modeling and instance weight adjustment. ImbGNN constructs an enhanced $G^2$G based on topology-friendly random walks, which helps mitigate the impact of topology-imbalance in the data. Our UniImb integrates a personalized graph perturbation strategy, a comprehensive graph structure encoding module, and the DBP module. The DBP module effectively extracts prototype features, significantly enhancing the representation capability for graphs of various scales. Our model achieves competitive performance.

## 5.3 COMPREHENSIVE EVALUATION OF UNIIMB: APPLICABILITY AND SCALABILITY

❶ **Applicability with various backbones.** We evaluated various GNN variants and Graph Transformers as backbone networks under class-imbalanced settings. The results in Table 4 show that all backbone networks achieved significant performance improvements when integrated with UniImb. This validates the plug-and-play applicability of our method, demonstrating its ability to significantly enhance the imbalanced graph classification performance of GNNs. For experiments on the applicable scenarios of topological imbalance, see the **Appendix I.7**.

❷ **Scalability on the large graph dataset**. Figure 2 illustrates the prediction accuracy (Macro-F1) and complexity (training time and memory usage) of several advanced models on the large-scale dataset AirGraph, which is naturally imbalanced. We observe that UniImb consistently outperforms strong baseline models, achieving superior performance in predicting air pollution levels. This provides valuable insights for environmental protection and human health. Furthermore, UniImb demonstrates good computational efficiency. Compared with methods specifically designed for imbalance handling, such as TopoImb, ImbGNN, and $G^2$GNN, **UniImb achieves more significant improvements in both performance and computational complexity.**

Table 4: Graph classification performance of various backbones combined with UniImb under the Class Imbalance setting with extreme imbalance degree.

| Method | D&D | | REDDIT-B | | IMDB-MULTI | |
|---|---|---|---|---|---|---|
| | Macro-F1 | Micro-F1 | Macro-F1 | Micro-F1 | Macro-F1 | Micro-F1 |
| GIN | 9.99 ± 7.44 | 11.88 ± 9.49 | 33.19 ± 14.26 | 36.02 ± 14.38 | 13.25 ± 6.19 | 14.92 ± 5.43 |
| GIN + Ours | 46.63 ± 3.42 | 83.60 ± 6.50 | 76.24 ± 4.09 | 88.82 ± 2.93 | 33.45 ± 7.83 | 45.72 ± 4.87 |
| GCN | 7.20 ± 2.29 | 7.77 ± 2.53 | 29.19 ± 1.38 | 31.00 ± 2.52 | 10.75 ± 4.10 | 12.82 ± 5.46 |
| GCN + Ours | 44.85 ± 4.15 | 81.63 ± 3.02 | 74.22 ± 4.39 | 86.94 ± 2.51 | 30.52 ± 3.19 | 43.93 ± 2.28 |
| GraphSage | 6.68 ± 3.01 | 7.15 ± 2.63 | 27.23 ± 2.21 | 29.62 ± 1.99 | 12.26 ± 2.72 | 14.01 ± 2.32 |
| GraphSage + Ours | 42.10 ± 5.54 | 79.38 ± 4.62 | 67.35 ± 2.92 | 83.02 ± 4.53 | 32.29 ± 3.28 | 45.70 ± 4.17 |
| GraphGPS | 10.12 ± 4.41 | 11.97 ± 3.91 | 11.68 ± 7.76 | 12.71 ± 8.13 | 14.20 ± 5.61 | 28.54 ± 13.78 |
| GraphGPS + Ours | 40.17 ± 2.19 | 68.27 ± 4.36 | 66.07 ± 4.36 | 79.79 ± 2.66 | 32.77 ± 2.98 | 43.07 ± 1.22 |
| Exphormer | 9.79 ± 4.18 | 10.85 ± 4.28 | 22.68 ± 10.79 | 27.33 ± 21.65 | 20.81 ± 5.43 | 28.14 ± 8.30 |
| Exphormer + Ours | 45.48 ± 0.94 | 83.43 ± 3.28 | 68.19 ± 1.38 | 79.59 ± 2.09 | 31.33 ± 2.17 | 43.95 ± 3.15 |
| Graph-Mamba | 4.99 ± 7.93 | 6.12 ± 10.36 | 15.27 ± 12.46 | 17.02 ± 14.71 | 16.89 ± 4.57 | 28.69 ± 12.47 |
| Graph-Mamba + Ours | 42.18 ± 2.11 | 72.96 ± 3.65 | 72.13 ± 1.41 | 82.58 ± 4.97 | 29.48 ± 5.64 | 41.02 ± 4.29 |

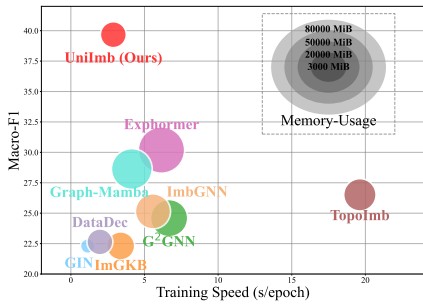

Figure 2: Model performance and efficiency comparison on AirGraph dataset.

## 5.4 PERFORMANCE ON INTERTWINED CLASS AND TOPOLOGY IMBALANCES SCENARIO

We further evaluated the effectiveness of various models in complex scenarios involving class and topological imbalance. Details of the data processing are provided in **Appendix G.1.5**. Additionally, AirGraph also exhibits a combined case of categorical and topological imbalance. Figures 3 and 2 compare our method with several advanced models. We found that methods designed to address topological imbalance generally perform worse than those targeting class imbalance. This is because the distribution of categorical features has a crucial impact on the decision boundary, directly influencing model performance. Our method achieved the best overall performance.

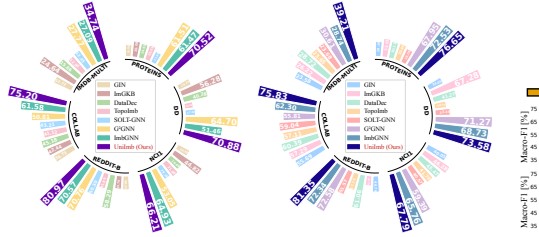

Figure 3: Macro-F1 (left) and Micro-F1 (right) on intertwined imbalance datasets.

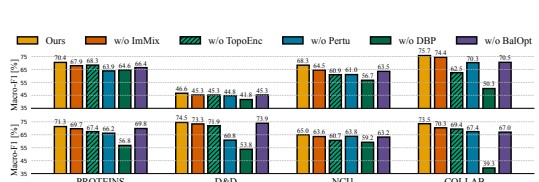

Figure 4: Ablation experiments on class imbalance (upper) and topological imbalance (lower).

## 5.5 ABLATION STUDY

To evaluate the effectiveness of the key components of the model, we create five variants: (1) '**w/o ImMix**' means that we remove imbalanced feature mixup strategy; (2) '**w/o TopoEnc**' means that we remove graph topology encoding; (3) '**w/o Pertu**' means that the graph perturbation strategy strategy is completely removed; (4) '**w/o DBP**' means that the dynamic balanced prototype strategy is removed; (5) '**w/o BalOpt**' means that the we remove the load balancing optimization.

As shown in Figure 4, the w/o TopoEnc variant achieves notably lower accuracy, indicating that incorporating structural information enables the model to better capture graph-level features. Among all ablated versions, the w/o DBP variant—without the proposed load-balanced prototype learning—exhibits the worst performance, demonstrating that explicitly modeling prototype features is critical for handling imbalanced graph data. The w/o BalOpt variant also suffers a performance drop, particularly on class-imbalanced datasets, which further confirms that the proposed load-balancing strategy facilitates the extraction of generalizable prototypes. Overall, the inferior performance of all ablated variants compared to UniImb provides strong evidence for the effectiveness and necessity of each component in our framework. For more ablation studies, see the **Appendix I.4 and I.5**.

## 5.6 HYPERPARAMETER SENSITIVITY ANALYSIS

We further evaluate the model's sensitivity to the number of prototypes. As shown in Figure 5, we observe that model performance first increases and then decreases with respect to K. When K is smaller than the optimal value, it is insufficient to capture enough discriminative prototype information. On the other hand, when K exceeds the optimal value, having too many prototypes

prevents the model from focusing on representative ones, leading to performance degradation. For sensitivity studies on TopK$_1$ and TopK$_2$, see the **Appendix I.6**.

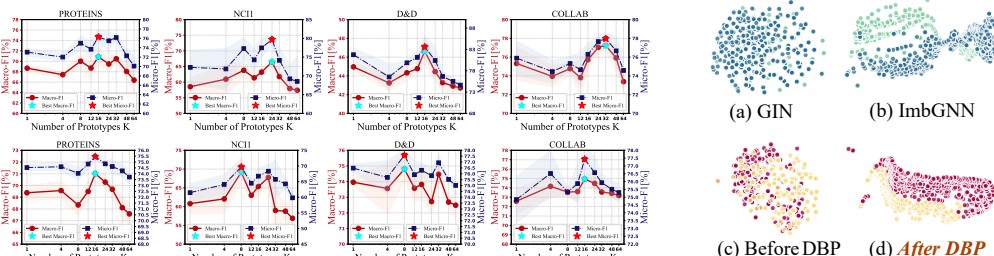

Figure 5: Sensitivity study on class imbalance (upper) and topological imbalance (lower).

Figure 6: Representation visualization on the PROTEINS dataset with extreme imbalance degree.

### 5.7 EVALUATION ON 3D CONFORMATION DATASETS

Beyond conventional graph datasets, we introduce 3D conformational graph datasets from MoleculeNet: BBBP for blood-brain barrier permeability prediction, BACE for $\beta$-secretase inhibition activity prediction, and HIV for antiviral activity prediction. To accommodate this graph structure, we incorporate specific models, including SchNet, DimeNet and SphereNet, which serve as backbone networks for graph imbalance methods. We follow established protocols for handling extremly imbalanced class distributions. And we select SphereNet as backbone of G$^2$GNN, TopoImb, and ImbGNN. For our model, we retain only the core DBP module. As shown in Table 5, the proposed method achieves excellent performance on these 3D Conformation datasets, validating its applicability to domains with complex structural dependencies.

Table 5: Performance comparison on class imbalance datasets with extreme imbalance degree.

| Model | HIV | | BBBP | | BACE | |
|---|---|---|---|---|---|---|
| | Macro-F1 | Micro-F1 | Macro-F1 | Micro-F1 | Macro-F1 | Micro-F1 |
| SchNet | 49.13 ± 2.16 | 96.57 ± 0.30 | 43.44 ± 0.94 | 74.95 ± 2.50 | 57.09 ± 4.18 | 60.00 ± 2.28 |
| SphereNet | 50.68 ± 1.50 | 96.82 ± 0.27 | 48.62 ± 8.48 | 76.27 ± 6.92 | 45.25 ± 5.27 | 52.12 ± 5.28 |
| DimeNet | 49.68 ± 0.66 | 96.69 ± 0.35 | 41.72 ± 2.97 | 71.57 ± 1.87 | 53.13 ± 7.01 | 56.50 ± 4.30 |
| G$^2$GNN | 54.96 ± 4.99 | 97.29 ± 3.62 | 62.65 ± 2.85 | 78.63 ± 4.31 | 52.25 ± 1.03 | 57.83 ± 2.14 |
| TopoImb | 52.40 ± 2.31 | 95.97 ± 2.44 | 53.21 ± 6.72 | 75.95 ± 5.83 | 47.81 ± 6.71 | 55.20 ± 6.29 |
| ImbGNN | 56.29 ± 2.62 | 97.63 ± 2.60 | 65.92 ± 2.30 | 82.74 ± 1.97 | 56.62 ± 2.43 | 63.52 ± 3.26 |
| **Ours** | **60.29** ± 3.18 | **98.95** ± 2.37 | **76.34** ± 2.16 | **86.80** ± 2.26 | **62.30** ± 3.99 | **68.34** ± 3.99 |

### 5.8 GRAPH REPRESENTATION VISUALIZATION

We extract the node-level representations from the PROTEINS dataset using GIN and ImbGNN, as well as the representations before and after applying the DBP module. These are then visualized in Figure 6. The representations generated by the naive GIN (sub-figure (a)) for imbalanced graphs are highly entangled and poorly separated, making it difficult for the classifier to distinguish between classes. While ImbGNN—a specialized method for imbalanced learning—shows relatively better separation, many samples still exhibit overlapping feature distributions. We observe that, after incorporating the Dynamic Balanced Prototype mechanism, the learned label-wise representations become significantly more discriminative. This enhancement greatly benefits the classifier in distinguishing between different classes and improves overall performance.

## 6 CONCLUSION

In this paper, we propose a unified framework for addressing imbalance graph classification task, effectively tackling both class imbalance and topological imbalance. Our core contribution lies in the introduction of a dynamic balanced prototype strategy, which extracts prototype features from graph data to enhance robust representation learning under imbalanced scenarios. We further impose a prototype-balancing optimization strategy to encourage balanced activation across all prototypes. Additionally, we incorporate several advanced techniques, including graph topology encoding, graph perturbation and feature mixup to comprehensively improve model performance. Extensive experiments on over 19 datasets demonstrate the impressive effectiveness of our framework.

## ETHICS STATEMENT

This work addresses graph learning and does not involve individual-level sensitive data or direct interaction with users. We do not identify any inherent ethical risks in the method itself. Nevertheless, downstream applications in sensitive domains may introduce harms, and practitioners should assess ethical implications before deployment.

## REPRODUCIBILITY STATEMENT

The main text formalizes the model architecture with equations. The Appendix contains full implementation details, including dataset descriptions, evaluation metrics, model architectures, hyperparameters, training procedures, and experiment configurations. The code is provided at an anonymous link.

## ACKNOWLEDGMENT

This paper is partially supported by the National Natural Science Foundation of China (No.12227901) and the Natural Science Foundation of Jiangsu Province (BK20250482). The AI-driven experiments, simulations and model training were performed on the robotic AI-Scientist platform of Chinese Academy of Science.

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

# Appendix

## Table of Contents

## A  USE OF LLM

We employed large language models solely to refine wording and improve readability and fluency. LLM assistance was limited to language editing and did not alter model design, data processing, or experimental results.

## B  RELATED WORK

### B.1  GRAPH REPRESENTATION LEARNING

Graphs serve as fundamental data structures for modeling complex relationships among entities (Yuan et al., 2025a; Wu et al., 2022; Derr et al., 2018), with broad applications in social networks (Li et al., 2023), energy systems (Yuan et al., 2025b), transportation (Wang et al., 2023; 2024a), and beyond. A central task in graph machine learning is graph representation learning, which aims to extract latent information from structured graph data and encode it into vectorized representations that support downstream reasoning (Hamilton, 2020). Early work in this area primarily focused on spectral methods based on the Laplacian operator (Noutahi et al., 2019) and random walks (Verdier et al., 2021). In recent years, the success of deep learning on Euclidean data has spurred the development of GNNs, which leverage a message-passing paradigm to effectively integrate topological structure with node features, yielding richer and more informative representations. Several GNN variants have since been proposed to address distinct challenges: graph attention network (Veličković et al., 2017) introduces an attention mechanism to dynamically weight neighbor contributions; GraphSAGE employs neighborhood sampling for scalable learning on large graphs (Hamilton et al., 2017); and GIN (Xu et al., 2018) enhances expressiveness by refining the aggregation function during message passing. However, most existing approaches focus on node-level embeddings, which may be insufficient for capturing higher-order structural patterns. Moreover, their use of topological information is often implicit and coarse. Recently, inspired by the transformative impact of Transformer architectures in NLP and vision, researchers have begun adapting them to graph representation learning (Yun et al., 2019; Cai & Lam, 2020). For example, Graphormer (Ying et al., 2021) uses a single dense self-attention mechanism in Transformer but adds structural features in the form of centrality and spatial encodings. GraphGPS (Rampášek et al., 2022) proposed a general framework for combining message-passing networks with attention mechanisms, while allowing for the mixing and matching of positional and structural embeddings. These emerging approaches better capture global dependencies and structural properties of graphs, marking a significant step forward in the evolution of graph learning methodologies.

### B.2  NODE-LEVEL IMBALANCE LEARNING ON GRAPH

Unlike graph classification, the node classification task aims to predict fine-grained labels for individual nodes (Liu et al., 2020; Ju et al., 2023; Yun et al., 2022; Fu et al., 2023). Class imbalance in node classification arises when the distribution of labeled nodes is uneven across different categories. To address this issue, researchers have explored various cutting-edge methods, which can be broadly categorized into algorithm-level and data-level approaches. Algorithm-level methods focus on designing representations optimized for minority classes. For example, ECGN (Thapaliya et al., 2024) enhances graph node classification by integrating cluster-aware node updates with synthetic minority class node generation. RLP-HGNN (Sun et al., 2025) designs a relation-aware label propagation mechanism to obtain pseudo-labels for nodes in heterogeneous graphs. DPGNN (Wang et al., 2021) adopts a class prototype-driven training strategy to balance losses across classes and utilizes metric learning to determine the positional relationship between nodes and their corresponding prototypes. At the data augmentation level, GraphSMOTE (Zhao et al., 2021b) generates synthetic minority-class node representations through interpolation between minority-class node pairs. GraphENS synthesizes complete ego-networks for minority classes (tail nodes and their one-hop neighbors) based on similarity. Tail-GNN (Liu et al., 2021) enhances tail node representations by transferring neighborhood patterns from high-degree nodes and locally adapting them to each target node. ImGAGN (Qu et al., 2021) mitigates class imbalance in node classification by using a graph generator to produce synthetic minority-class nodes, thereby balancing the class distribution.

**In contrast, graph classification assigns a single label to each graph, which means there may be no node-level labels. Techniques that rely on node annotations or node-level augmentation**

**cannot be directly transferred to this scenario, demanding specific strategies to handle class imbalance at the graph level.**

### B.3 EDGE-LEVEL IMBALANCE LEARNING ON GRAPH

Edge-level class imbalance is an important and understudied problem with practical relevance to tasks such as link prediction, edge-level anomaly detection, and inductive recommendation (Liu et al., 2025). The work (Wang & Derr, 2022) were among the first to expose degree-related biases in link prediction. The influential TopoEdge (Cheng et al., 2025) formally introduced the notion of topological imbalance for edge classification and proposed topological entropy as a mechanism for weighting edge importance, which catalyzed subsequent work in this area. Follow-up approaches include EdgeSMOTE (Gong et al., 2025), which synthesizes minority-class edges by generating high-quality node features to rebalance the class distribution. In application domains where edge imbalance is inherent—e.g., few-shot link prediction (Yang et al., 2022) and edge-level anomaly detection (Liu et al., 2022a)—researchers have pursued improvements via architecture adaptations and tailored loss functions.

### B.4 SUBGRAPH EXTRACTION ON GRAPH

Subgraph extraction is a fundamental technique for identifying the most influential local structures within a graph. It plays a pivotal role in fields such as protein analysis, biomedicine, and community detection (Yang et al., 2023). Recent work has proposed a variety of subgraph-oriented methods. Density-based discovery approaches have been adapted to network science, biological analysis, and graph databases (Nguyen & Vullikanti, 2021). GNN-AK (Zhao et al., 2021a) leverages star pattern subgraphs with a base graph neural network kernel to perform convolution across all subgraphs, thereby producing diverse and informative node embeddings. SubGNN (Alsentzer et al., 2020) decomposes a graph's topology into three channels that reflect position, neighborhood, and structure, and it designs corresponding subgraph patterns to capture each aspect . Despite these advances, the issue of subgraph imbalance remains largely unaddressed. In many graphs, critical structures may occur infrequently but play a decisive role, as exemplified by key molecular motifs in chemical compounds. Accurately extracting and characterizing such minority subgraphs remains an important and underexplored challenge.

## C METHOD

### C.0.1 IMBALANCED FEATURE MIXUP

Given the representation of N graphs denoted as $\mathbf{H}$, the feature mixup strategy randomly shuffle the order of the representation vectors of the graph instances, and the output is denoted as $\bar{\mathbf{H}} \in \mathbb{R}^{N \times d_h}$. Then, we sample a mixing ratio $\lambda \in [0, 1]$ from a Beta($\alpha, \alpha$) distribution, and then we perform a linear interpolation between $\mathbf{H}$ and $\bar{\mathbf{H}}$ using the mixing ratio $\lambda$ as follows,

$$\widehat{\mathbf{H}}_i = \lambda \mathbf{H}_i + (1 - \lambda)\bar{\mathbf{H}}_i \in \mathbb{R}^{d_h}, \quad i \in [1, \cdots, N] \tag{13}$$

Finally, we concatenate the mixed instances with the previous graph instance representations, and the output is denoted as $\widetilde{\mathbf{H}} = [\widehat{\mathbf{H}}, \mathbf{H}] \in \mathbb{R}^{2N \times d_h}$. In this way, the representation of the tail graphs can gain supplementary information from the head graph representation.

## D THEORY PROOF

### D.1 EXPLANATION FOR THEORY 1

For the training graphs $\mathbf{G}$, prototype features $\mathbf{S}$, and labels $\mathbf{Y}$, the objective of the information bottleneck is expressed as minimizing the following objective function:

$$\min \mathrm{I}(\mathbf{S}; \mathbf{G}) - \beta \mathrm{I}(\mathbf{S}; \mathbf{Y}) \tag{14}$$

The Lagrange multiplier $\beta$ determines the level of relevant information captured by the representation $\mathbf{S}, \mathrm{I}(\mathbf{S}; \mathbf{Y})$. For the prototype representation variable $\mathbf{S}$, which is obtained via a nonlinear mapping

from the graph representation $\mathbf{G}$, its statistical properties can be characterized either by its joint distribution with $\mathbf{G}$ and the label $\mathbf{Y}$, or equivalently by the encoder distribution $P(\mathbf{S}|\mathbf{G})$ and the decoder distribution $P(\mathbf{Y}|\mathbf{S})$. Given the joint distribution $P(\mathbf{G}, \mathbf{Y})$, the prototype representation $\mathbf{S}$ can be uniquely mapped to a point on the information plane (Shwartz-Ziv & Tishby, 2017), with coordinates $(I(\mathbf{G}; \mathbf{S}), I(\mathbf{S}; \mathbf{Y}))$, which represent the mutual information between the prototype and the input graph, and between the prototype and the label, respectively. Based on this perspective, we apply the information-theoretic analysis to our proposed model, **UniImb**, whose architecture satisfies the Markov chain condition (Tishby & Zaslavsky, 2015). Accordingly, the variable $\mathbf{S}$ can be mapped to a point on the information plane and forms a unique information path together with the input $\mathbf{G}$, the label $\mathbf{Y}$, and the predicted label $\hat{\mathbf{Y}}$:

$$\mathbf{Y} \leftrightarrow \mathbf{G} \rightarrow \mathbf{S} \rightarrow \hat{\mathbf{Y}} \tag{15}$$

This path satisfies the following Data Processing Inequality (DPI) chains (Cover, 1999):

$$I(\mathbf{G}; \mathbf{Y}) \geq I(\mathbf{S}; \mathbf{Y}) \geq I(\hat{\mathbf{Y}}; \mathbf{Y}) \quad \text{and} \quad H(\mathbf{G}) \geq I(\mathbf{G}; \mathbf{S}) \geq I(\mathbf{G}; \hat{\mathbf{Y}}) \tag{16}$$

These inequalities indicate that the prototype representation compresses the input information while retaining task-relevant signals, thus revealing a fundamental trade-off between representation capacity and generalization from an information-theoretic perspective. Therefore, we can maximize the mutual information between the predicted labels and the ground-truth labels, $I(\hat{\mathbf{Y}}; \mathbf{Y})$, through supervised learning, which in turn maximizes the mutual information between the prototype representation and the labels, $I(\mathbf{S}; \mathbf{Y})$. Based on this objective, the optimization process can be further simplified to minimizing the mutual information between the training graph set $\mathbf{G}$ and the prototype feature $\mathbf{S}$, $I(\mathbf{S}; \mathbf{G})$, i.e., reducing the redundancy in the input information caused by class-imbalance.

### D.2 PROOF FOR PROPOSITION 1

According to Theorem 1, our goal is to minimize the mutual information between the prototype distribution $\mathbf{S}$ and the input graph data $\mathbf{G}$, in order to reduce the dependency on the imbalanced distribution of the training data. This ensures that the prototypes are not overly tailored to specific graphs, but instead are shared in a more general and consistent manner across the entire dataset.

Formally, the mutual information between prototype variable $\mathbf{S}$ and the graph identity $\mathbf{G}$ is defined as follows:

$$I(\mathbf{S}; \mathbf{G}) = \sum_{\mathcal{G}, s} \mathrm{p}(\mathcal{G}) \mathrm{p}(s|\mathcal{G}) \log \frac{\mathrm{p}(s|\mathcal{G})}{\mathrm{p}(s)} \tag{17}$$

Directly computing the marginal distribution of the prototype variable, $\mathrm{p}(s) = \sum_{i=1}^{2N} \mathrm{p}(s, \mathcal{G}_i) = \sum_{i=1}^{2N} \mathrm{p}(s|\mathcal{G}_i) \mathrm{p}(\mathcal{G}_i)$, is generally intractable. To address this, we introduce an auxiliary distribution $r(s)$ to approximate the true marginal. By the non-negativity of the Kullback-Leibler divergence, i.e.,

$$\mathrm{KL}[\mathrm{p}(s)\|r(s)] = \sum_s \mathrm{p}(s) \log \frac{\mathrm{p}(s)}{r(s)} \geq 0 \tag{18}$$

we obtain the inequality

$$\sum_s \mathrm{p}(s) \log \mathrm{p}(s) \geq \sum_s \mathrm{p}(s) \log r(s), \tag{19}$$

with equality holding if and only if $\mathrm{p}(s) = r(s)$ for every $s \in \mathcal{S}$. This result leads to an upper bound on the mutual information, which allows us to replace the intractable marginal term with a more manageable surrogate $r(s)$. Therefore, we arrive at an upper bound on the mutual information:

$$I(\mathbf{S}; \mathbf{G}) \leq \sum_{\mathcal{G}, s} \mathrm{p}(\mathcal{G}) \mathrm{p}(s|\mathcal{G}) \log \frac{\mathrm{p}(s|\mathcal{G})}{r(s)} \tag{20}$$

Then we approximate the true joint distribution $p(\mathcal{G}, y) = p(\mathcal{G}) p(y|\mathcal{G})$ using the empirical data distribution (Alemi et al., 2016) as follows:

$$p(\mathcal{G}, y) = \frac{1}{2N} \sum_{i=1}^{2N} \delta_{\mathcal{G}_i}(\mathcal{G}) \, \delta_{y_i}(y) \tag{21}$$

where the Dirac delta function $\delta(\cdot)$ is used to express the empirical distribution of the joint random variables $\mathcal{G}$ and $y$, approximating the true joint distribution $p(\mathcal{G}, y)$. Therefore, we can obtain this formula:

$$I(\mathbf{S}; \mathbf{G}) \leq \frac{1}{2N} \sum_{i=1}^{2N} \sum_{s} p(s|\mathcal{G}_i) \log \frac{p(s|\mathcal{G}_i)}{r(s)} \tag{22}$$

Assuming that our choice of $p(s|\mathcal{G}_i)$ and the auxiliary distribution $r(s)$ allows for an analytical evaluation of the Kullback–Leibler divergence, we can directly incorporate this term into the objective. Thus, we obtain the following optimizable objective, which we aim to minimize:

$$\min I(\mathbf{S}; \mathbf{G}) \implies \min \frac{1}{2N} \sum_{i=1}^{2N} \sum_{s} \mathbb{E}_{\mathcal{G}_i}[KL(p(s|\mathcal{G}_i)||r(s))] \tag{23}$$

In the paper, the prototypes are derived from the interaction between graph representations and similarity coefficients. Therefore, the formula can be written as:

$$
\begin{aligned}
\min I(\mathbf{S}; \mathbf{G}) \implies & \min \frac{1}{2N} \sum_{i=1}^{2N} \sum_{s} \mathbb{E}_{\mathcal{G}_i}[KL(p(s|\mathcal{G}_i)||r(s))] \\
= & \min \frac{1}{2N} \sum_{i=1}^{2N} \sum_{k=1}^{K} \mathbb{E}_{\mathcal{G}_i}[KL(p(s_k|\mathcal{G}_i)||r(s_k))] \\
= & \min \frac{1}{2N} \sum_{i=1}^{2N} \sum_{k=1}^{K} \mathbb{E}_{\mathcal{G}_i}[KL\left(p(p_k f(\mathcal{G}_i)|\mathcal{G}_i)||r\left(u_k f\left(\widetilde{\mathcal{G}}\right)\right)|\widetilde{\mathcal{G}}\right)] \\
\implies & \min \frac{1}{2N} \sum_{i=1}^{2N} \sum_{k=1}^{K} \mathbb{E}_{\mathcal{G}_i}[KL(p(p_k f(\mathcal{G}_i)|\mathcal{G}_i)||r(u_k f(\mathcal{G}_i))|\mathcal{G}_i)]
\end{aligned}
\tag{24}
$$

where $f(\cdot)$ means the mapping function of the graph. In the last line, to align the two terms, we introduce auxiliary variable $\widetilde{\mathcal{G}}$, which denotes any graph. And $\mathbf{P} = \{p_1, p_2, \ldots, p_K\}$ represents the similarity probability distribution of K prototypes. From step three to step four, we set $\widetilde{\mathcal{G}}$ to $\mathcal{G}_i$, if so, we still satisfy the distribution $r(s)$ rather than the general case, by modulating the auxiliary similarity probability distribution $\mathbf{U} = \{u_1, u_2, \ldots, u_K\}$.

Thus, the original objective can be optimized by minimizing the Kullback–Leibler divergence between the similarity probability distribution $\mathbf{P}$ and the probability distribution $\mathbf{U}$:

$$\min I(\mathbf{S}; \mathbf{G}) \Rightarrow \min KL(\mathbf{P} \| \mathbf{U}) = \min \sum_{k=1}^{K} p_k \log \frac{p_k}{u_k} \tag{25}$$

Then we let:

$$p_k = u_k + \delta_k \quad \text{with} \quad \sum_{k=1}^{K} \delta_k = 0 \tag{26}$$

Therefore, we have the following formula,

$$KL(\mathbf{P} \| \mathbf{U}) = \sum_{k=1}^{K} (u_k + \delta_k) \log \left(1 + \frac{\delta_k}{u_k}\right) \tag{27}$$

We aim for $p_k \approx u_k$, and $\delta_k \to 0$, $\frac{\delta_k}{u_k} \to 0$, $\forall k = 1, 2, \ldots, K$, so we use the Taylor expansion for $\log(1+x)$ around $x = 0$. Apply this to $\log(1 + \frac{\delta_k}{u_k})$:

$$\log\left(1 + \frac{\delta_k}{u_k}\right) = \frac{\delta_k}{u_k} - \frac{1}{2}\left(\frac{\delta_k}{u_k}\right)^2 + \frac{1}{3}\left(\frac{\delta_k}{u_k}\right)^3 + \mathcal{O}\left(\left(\frac{\delta_k}{u_k}\right)^4\right) \tag{28}$$

Substitute into the KL divergence:

$$\begin{aligned}
\text{KL}(\mathbf{P} \parallel \mathbf{U}) &\approx \sum_{k=1}^{K}(u_k + \delta_k)\left[\frac{\delta_k}{u_k} - \frac{1}{2}\left(\frac{\delta_k}{u_k}\right)^2 + \frac{1}{3}\left(\frac{\delta_k}{u_k}\right)^3 + \mathcal{O}\left(\left(\frac{\delta_k}{u_k}\right)^4\right)\right] \\
&= \sum_{k=1}^{K}\left[\delta_k + \frac{1}{2}\cdot\frac{\delta_k^2}{u_k} - \frac{1}{6}\cdot\frac{\delta_k^3}{u_k^2} + \frac{1}{3}\cdot\frac{\delta_k^4}{u_k^3} + \mathcal{O}\left(\left(\frac{\delta_k}{u_k}\right)^4\right)\right]
\end{aligned} \tag{29}$$

Since the first-order term $\sum_{k=1}^{K}\delta_k = 0$, the KL divergence simplifies to a second-order term $\sum_{k=1}^{K}\frac{1}{2}\frac{\delta_k^2}{u_k}$, a third-order correction $\sum_{k=1}^{K}-\frac{1}{6}\frac{\delta_k^3}{u_k^2}$ and a forth-order term $\frac{1}{3}\frac{\delta_k^4}{u_k^3}$.

Thus, the approximation of the KL divergence is:

$$\text{KL}(\mathbf{P} \parallel \mathbf{U}) \approx \frac{1}{2}\sum_{k=1}^{K}\frac{\delta_k^2}{u_k} - \frac{1}{6}\sum_{k=1}^{K}\frac{\delta_k^3}{u_k^2} + \frac{1}{3}\sum_{k=1}^{K}\frac{\delta_k^4}{u_k^3} + \sum_{k=1}^{K}\mathcal{O}\left(\left(\frac{\delta_k}{u_k}\right)^4\right) \tag{30}$$

Since $\delta_k$ is a relatively small quantity (i.e., $p_k$ is close to the uniform distribution $\frac{1}{K}$), we only retain the first two terms as an approximation. Since the first-order term is zero, we have:

$$\begin{aligned}
\text{KL}(\mathbf{P} \parallel \mathbf{U}) &\approx \frac{1}{2}\sum_{k=1}^{K}\frac{\delta_k^2}{u_k} \\
&= \frac{1}{2}\sum_{k=1}^{K}\frac{(p_k - u_k)^2}{u_k}
\end{aligned} \tag{31}$$

Since our goal is to minimize $\text{KL}(\mathbf{P} \parallel \mathbf{U})$, we can observe that the KL divergence between the probability $\mathbf{P}$ and the auxiliary distribution $\mathbf{U}$ its minimum when these two distributions are equal. For intuitive optimization, we convert it into hit counts. For 2N graphs, each graph hits $\text{TopK}_2$ prototypes, and we have a total hit count of $\mathcal{K}=2\text{N}\times\text{TopK}_2$. The actual number of hits for each prototype is $\mathbb{N} = \{n_1, n_2, \cdots, n_K\}$, where $n_k = p_k\mathcal{K}$, $k \in \{1, \cdots, K\}$. If the attention coefficient of each prototype and a graph is greater than zero, then we consider this prototype to be hit once. In order word, Each value in $\mathbb{N} \in \mathbb{R}^K$ is equal to the number of values greater than zero in each column of $\text{TopK}_2\left(\widetilde{\mathbf{H}}\mathbf{S}^\top/\sqrt{d_h} + \eta\right) \in \mathbb{R}^{2\text{N}\times K}$.

However, we aim to optimize $\mathbb{N}$ to approach $\mathcal{K}\mathbf{U}$ through a loss function. Nevertheless, the TopK operation is non-differentiable in practice. To address this, we introduce a regulation coefficient $\eta \in \mathbb{R}^K$ into the similarity computation process as follows,

$$\widehat{\mathbf{H}} = \text{Sigmoid}\left(\text{TopK}_2\left(\widetilde{\mathbf{H}}\mathbf{S}^\top/\sqrt{d_h} + \eta\right) + \boldsymbol{\gamma}\right)\widetilde{\mathbf{H}}_{\text{S}}, \tag{32}$$

### D.3 Prototype-balancing Optimization

To meet the above objectives, we introduce a specific constraint loss $\mathcal{L}_{\mathcal{M}}$ that iteratively models the prototype activation process during training.

$$\begin{aligned}
\mathcal{L}_{\mathcal{M}} &= \frac{1}{2}\sum_{k=1}^{K}\left|n_k - \frac{2*\text{N}*\text{TopK}_2}{\frac{1}{u_k}}\right|^2 \\
&= \frac{1}{2}\sum_{k=1}^{K}\left|\eta + \text{StopGrad}(n_k - \eta) - \frac{2*\text{N}*\text{TopK}_2}{\frac{1}{u_k}}\right|^2
\end{aligned} \tag{33}$$

where $n_k \in \mathbb{R}^K$ represents the sequence of activation counts for $K$ prototypes, $\text{StopGrad}(\cdot)$ denotes the stop-gradient operator (Chen & He, 2021), which *behaves as identity in the forward pass but yields zero in the backward pass*, and this is a common approach used for non-differentiable operations. The gradient of the loss $\mathcal{L}_\mathcal{M}$ with respect to $\eta$ is:

$$\nabla_\eta \mathcal{L}_\mathcal{M} = n_k - \frac{2 * \text{N} * \text{TopK}_2}{\frac{1}{u_k}} \tag{34}$$

Therefore, $\eta$ can be optimized by the following objectives:

$$\eta \leftarrow \eta - \varphi \, \text{sgn} \left( \nabla_\eta \mathcal{L}_\mathcal{M} \right) = \eta - \varphi \, \text{sgn} \left( n_k - \frac{2 * \text{N} * \text{TopK}_2}{\frac{1}{u_k}} \right) \tag{35}$$

where $\varphi$ is the learning rate for updating $\eta$. The sign function $\text{sgn}(\cdot)$ is used to control the attenuation direction. Next, we select several commonly used **discrete distributions** as the auxiliary distributions $\mathbf{U}$ to show.

❶ **Uniform Distribution.** when the auxiliary distribution $\mathbf{U}$ follows a uniform distribution (Kuipers & Niederreiter, 2012), i.e., $\mathbf{U} = \underbrace{\left( \frac{1}{\text{K}}, \ldots, \frac{1}{\text{K}} \right)}_{\text{K}}$. our load balancing optimization goal can be written as:

$$\eta \leftarrow \eta - \varphi \, \text{sgn} \left( \nabla_\eta \mathcal{L}_\mathcal{M} \right) = \eta - \varphi \, \text{sgn} \left( n_k - \frac{2 * \text{N} * \text{TopK}_2}{\text{K}} \right) \tag{36}$$

When the load $\mathcal{M}_{\text{hit}}$ for a prototype is higher than the average, it will be penalized to reduce its activation frequency. This mechanism encourages a balanced distribution of attention between head and tail graph information during training.

❷ **Zipf Distribution.** when the auxiliary distribution $\mathbf{U}$ follows a Zipf distribution (Axtell, 2001), it can be written as,

$$u_k = \frac{1/k^s}{Z}, \quad \text{where} \quad Z = \sum_{j=1}^{\text{K}} \frac{1}{j^s}, \quad s > 1, \quad k \in \{1, \cdots, \text{K}\} \tag{37}$$

where $s$ is the parameter of zipf distribution and our optimization goal is as follows,

$$\eta \leftarrow \eta - \varphi \, \text{sgn} \left( n_k - \frac{2 * \text{N} * \text{TopK}_2}{k^s Z} \right) \tag{38}$$

where $\varphi$ is the learning rate, and $Z = \sum_{j=1}^{\text{K}} \frac{1}{j^s}$.

❸ **Exponential Distribution.** when the auxiliary distribution $\mathbf{U}$ follows an exponential distribution (Marshall & Olkin, 1967), it can be written as,

$$u_k = \frac{e^{-\lambda k}}{Z}, \quad \text{where} \quad Z = \sum_{j=1}^{\text{K}} e^{-\lambda j}, \quad \lambda > 0 \tag{39}$$

where the decay parameter $\lambda$ governs the steepness of decline. A larger $\lambda$ implies a stronger preference toward early-index prototypes, effectively enforcing sparsity and focus in the prototype activation pattern. And our optimization goal is as follows,

$$\eta \leftarrow \eta - \varphi \, \text{sgn} \left( n_k - \frac{2 * \text{N} * \text{TopK}_2}{Z} e^{-\lambda k} \right) \tag{40}$$

❹ **Poisson Distribution.** when the auxiliary distribution $\mathbf{U}$ follows a poisson distribution (Consul & Jain, 1973), it can be written as,

$$u_k = \frac{\lambda^k e^{-\lambda}}{k! Z}, \quad \text{where} \quad Z = \sum_{j=1}^{K} \frac{\lambda^j e^{-\lambda}}{j!} \tag{41}$$

where $\lambda$ acts as both the mean and the mode of the distribution. When $\lambda$ is small, attention is concentrated on early prototypes; when $\lambda$ increases, the most attended prototypes shift to middle indices, promoting a more balanced activation pattern centered around a specific region in the prototype space. And our optimization goal is as follows,

$$\eta \leftarrow \eta - \varphi \operatorname{sgn}\left(n_k - \frac{2 * \mathrm{N} * \mathrm{TopK}_2}{Z} * \frac{\lambda^k e^{-\lambda}}{k!}\right) \tag{42}$$

In **Appendix D.4**, *we provide extensive experimental evidence demonstrating that the model achieves optimal performance when the auxiliary distribution* $\mathbf{U}$ *follows a uniform distribution*, i.e., $\mathbf{U} = \underbrace{\left(\frac{1}{\mathrm{K}}, \ldots, \frac{1}{\mathrm{K}}\right)}_{\mathrm{K}}$. This indicates that when each prototype has a balanced load, the extracted prototype features are more generalizable.

In fact, a prior study, DeepSeek (Wang et al., 2024c), also follows a similar intuition and applies the concept of balancing to the design of large language models, claiming that balanced expert allocation can enhance model performance. While their work differs fundamentally in task setting, our core contribution lies in rigorously justifying this intuitive principle from the perspective of information bottleneck theory—particularly in the context of imbalanced graph learning. Furthermore, we provide a more detailed theoretical analysis of the balancing optimization strategy, along with formal derivations that establish its effectiveness. This complements and extends previous work by offering a principled formulation of the balancing objective and demonstrating its benefits through both theoretical perspectives.

Finally, our load balancing optimization goal can be written as:

$$\eta \leftarrow \eta - \varphi \operatorname{sgn}\left(\nabla_\eta \mathcal{L}_{\mathcal{M}}\right) = \eta - \varphi \operatorname{sgn}\left(n_k - \frac{2 * \mathrm{N} * \mathrm{TopK}_2}{\mathrm{K}}\right) \tag{43}$$

### D.4 COMPARISON ACROSS DIFFERENT DISTRIBUTIONS OF U

In this section, we present evidence for setting $\mathbf{U}$ to an uniform distribution. Using four class-imbalanced datasets with extreme imbalance ratio as example, we compared five settings of $\mathbf{U}$, including Zipf distribution (Axtell, 2001), Exponential distribution (Marshall & Olkin, 1967), Poisson distribution (Consul & Jain, 1973), and Uniform distribution (Kuipers & Niederreiter, 2012)). These settings correspond to different optimization losses described in **Appendix D.3**. As shown in Table 7 and Table 6, we observe that the model achieves the best performance when U follows a uniform distribution — that is, when prototypes are expected to be activated uniformly. This setting encourages better feature extraction for tail classes, leading to improved generalization.

Table 6: Macro-F1 and Micro-F1 scores on *class imbalance* datasets with *extreme imbalance degree* under *different* U *distributions*. The best results are marked and the runner-ups are underlined. We report the average and standard deviation over 20 runs.

| Distribution | PROTEINS | | D&D | | NCI1 | | COLLAB | |
|---|---|---|---|---|---|---|---|---|
| | Macro-F1 | Micro-F1 | Macro-F1 | Micro-F1 | Macro-F1 | Micro-F1 | Macro-F1 | Micro-F1 |
| Zipf (Axtell, 2001) | 67.80 ± 3.71 | 73.29 ± 4.00 | 44.05 ± 2.29 | 79.02 ± 7.38 | 65.69 ± 3.11 | 79.78 ± 1.58 | 69.25 ± 10.26 | 69.79 ± 10.08 |
| Exponential (Marshall & Olkin, 1967) | 66.45 ± 3.34 | 71.99 ± 4.94 | 44.99 ± 2.95 | 82.31 ± 9.36 | 65.03 ± 2.41 | 79.85 ± 2.09 | 73.51 ± 6.73 | 74.47 ± 6.09 |
| Poisson (Consul & Jain, 1973) | 68.49 ± 1.70 | 73.94 ± 3.40 | 41.62 ± 3.72 | 71.98 ± 10.64 | 67.67 ± 5.50 | 80.60 ± 5.09 | 75.56 ± 2.03 | 75.85 ± 2.18 |
| Uniform (Kuipers & Niederreiter, 2012) | 70.44 ± 4.72 | 74.50 ± 4.99 | 46.63 ± 3.42 | 83.60 ± 6.50 | 68.30 ± 5.19 | 80.68 ± 4.22 | 75.73 ± 2.52 | 76.34 ± 2.60 |

## E ALGORITHM PSEUDOCODE

We present the pseudocode for UniImb in Algorithm 1, and comprehensive implementation details are available at our anonymized code repository link.

Table 7: Macro-F1 and Micro-F1 scores on *topological imbalance* datasets with *extreme imbalance degree* under *different* **U** *distributions*.

| Distribution | PROTEINS | | D&D | | NCI1 | | COLLAB | |
|---|---|---|---|---|---|---|---|---|
| | Macro-F1 | Micro-F1 | Macro-F1 | Micro-F1 | Macro-F1 | Micro-F1 | Macro-F1 | Micro-F1 |
| Zipf (Axtell, 2001) | 69.39 ± 1.54 | 74.01 ± 0.92 | 74.32 ± 0.71 | 76.13 ± 0.52 | 63.05 ± 3.20 | 63.94 ± 1.83 | 69.61 ± 6.89 | 72.47 ± 1.78 |
| Exponential (Marshall & Olkin, 1967) | 69.87 ± 2.27 | 74.57 ± 0.97 | 74.17 ± 1.20 | 75.70 ± 1.01 | 63.09 ± 1.60 | 64.15 ± 1.54 | 64.75 ± 15.75 | 69.30 ± 11.20 |
| Poisson (Consul & Jain, 1973) | 69.10 ± 1.21 | 74.35 ± 0.43 | 74.41 ± 0.62 | 76.38 ± 0.32 | 58.65 ± 9.14 | 59.88 ± 3.27 | 68.09 ± 9.21 | 70.33 ± 8.84 |
| **Uniform** (Kuipers & Niederreiter, 2012) | **71.32** ± 1.88 | **74.89** ± 1.12 | **74.49** ± 1.13 | **76.73** ± 1.04 | **64.99** ± 9.58 | **65.76** ± 7.24 | **73.51** ± 1.48 | **75.54** ± 1.63 |

---

**Algorithm 1:** UniImb for Imbalanced Graph Classification

---

**Input:** Input Graph $G = \{\mathcal{G}_1, \mathcal{G}_2, \ldots, \mathcal{G}_N\}$ with N graph instances where the graph instance $\mathcal{G}_i = \{\mathcal{V}^{\mathcal{G}_i}, \mathcal{E}^{\mathcal{G}_i}, \mathbf{A}^{\mathcal{G}_i}, \mathbf{X}^{\mathcal{G}_i}\}$ and the corresponding label set $\mathbf{Y} = \{y_1, y_2, \ldots, y_N\}$.

**Output:** Well-trained parameters $\eta$ of UniImb.

1 **while** *maximum epochs nor reached or not converged* **do**

2      **for** *patience* $= 1, 2, \ldots, P$ **do**

3          Computing graph topology encoding **LE** and **GE** in Eq.1 and Eq.2;

4          Implementing personalized graph perturbation strategy;

5          $\widetilde{\mathbf{H}} \leftarrow \text{ReadOut}\left(\text{GNN}\left(\widetilde{\mathbf{X}}, \mathbf{GE}, \mathbf{LE}\right)\right);$      `// Generate graph-level representations`

6          $\widetilde{\mathbf{H}}_S = \text{Softmax}\left(\text{TopK}_1\left(\mathbf{S}\widetilde{\mathbf{H}}^\top / \sqrt{d_h}\right)\right)\widetilde{\mathbf{H}}\mathbf{W}_v;$      `// Prototype feature perception`

7          $\widehat{\mathbf{H}} = \text{Sigmoid}\left(\text{TopK}_2\left(\widetilde{\mathbf{H}}\mathbf{S}^\top / \sqrt{d_h} + \eta\right) + \gamma\right)\widetilde{\mathbf{H}}_S;$      `// Dynamic balanced prototype`

8          $\Theta \leftarrow \Theta - \alpha\nabla_\Theta\mathcal{L}^{ce};$      `// Update the parametners of UniImb using cross-entropy classification loss`

9          $\eta \leftarrow \eta - \varphi\,\text{sgn}\left(n_k - \frac{2*N*\text{TopK}_2}{K}\right);$      `// Update the balance regulation coefficient using prototype-balancing Optimization loss`

10      **end**

11 **end**

---

# F   DEFINITION OF GRAPH CLASS IMBALANCE AND TOPOLOGICAL IMBALANCE

## F.1   CLASS IMBALANCE

**Definition.** There is a notable imbalance in the number of labeled samples (nodes or graphs) across different classes, resulting in a long-tailed distribution of quantities (Qin et al., 2024).

**Definition 1 (Graph-Level Class Imbalance Ratio).** Graph-level Class Imbalance occurs when there is a disproportionate number of labeled graphs across different classes, resulting in a long-tailed distribution. Given a set of labeled graphs $G = \{\mathcal{G}_1, \mathcal{G}_2, \ldots, \mathcal{G}_N\}$ with $N$ graphs, where each $\mathcal{G}_i$ represents the graphs in class $i$, the imbalance ratio $\rho$ is defined as the ratio between the size of the largest class and the smallest class, i.e.,

$$\rho = \frac{|\mathcal{G}_{\max}|}{|\mathcal{G}_{\min}|} \tag{44}$$

where $\mathcal{G}_{\max}$ denotes the number of graphs in the largest class and $\mathcal{G}_{\min}$ represents the number of graphs in the smallest class.

## F.2   TOPOLOGICAL IMBALANCE

**Definition.** The imbalance is facilitated by the uneven graph size (the number of nodes) distribution (Qin et al., 2024).

**Definition 2 (Graph-Level Topological Imbalance Ratio).** Following by the work (Qin et al., 2024), Graph-level Topology Imbalance refers to the uneven distribution of graph sizes (i.e., number of nodes) across a dataset, which often leads to biased model performance. Let $G = \{\mathcal{G}_1, \mathcal{G}_2, \ldots, \mathcal{G}_N\}$ with $N$ graphs denote the set of labeled training graphs. To capture the long-tailed nature of graph size distributions, we follow the Pareto principle (i.e., the 20/80 rule) (Sanders, 1987) and divide the labeled graph set $G$ into two parts: the top 20% of graphs with the largest number of nodes constitute the head subset $\mathcal{H}_\mathcal{G}$, while the remaining 80% form the tail subset $\mathcal{T}_\mathcal{G}$. Then we sample the training set generated by the graph based on different settings. The topology-imbalance ratio $\rho$ is then defined as the ratio between the average graph size in the head set and that in the tail set:

$$\rho = \frac{\frac{1}{|\mathcal{H}_\mathcal{G}|} \sum_{\mathcal{G}_i \in \mathcal{H}_\mathcal{G}} |\mathcal{V}^{\mathcal{G}_i}|}{\frac{1}{|\mathcal{T}_\mathcal{G}|} \sum_{\mathcal{G}_j \in \mathcal{T}_\mathcal{G}} |\mathcal{V}^{\mathcal{G}_j}|} \tag{45}$$

where $\mathcal{V}^{\mathcal{G}_i}$ denotes the number of nodes in graph $\mathcal{G}_i$. Such imbalance is especially prevalent in graph classification tasks such as molecular property prediction or protein function analysis. Larger graphs typically encode richer structural information, making them more expressive and leading models to perform better on them. As a result, models may become biased toward large graphs, while underperforming on small ones, which contributes to performance disparity.

## G  DATASET

### G.1  INTRODUCTION

#### G.1.1  COMMON GRAPH CLASSIFICATION DATASETS

We conducted extensive experiments on **12** widely-adopted graph datasets, including nine **binary-classification** datasets and three **multi-class classification** datasets. As shown in Table 8, these datasets cover a range of domains: ❶ *Chemical compounds*: PTC-MR, NCI1, MUTAG, FRANKENSTEIN, and AIDS. ❷ *Protein compounds*: PROTEINS, D&D, and DHFR. ❸ *Social and collaboration networks*: REDDIT-B, IMDB-MULTI, and COLLAB. ❹ *Synthetic graphs*: Synthie.

Table 8: Statistics of all benchmark datasets for imbalanced graph classification.

| Dataset | Domain | # Graphs | # Avg-Node | # Avg-Edge | # Features | # Classes | #$\mathcal{G}_{\text{head}}$ |
|---|---|---|---|---|---|---|---|
| MUTAG (Debnath et al., 1991) | Chemical | 188 | 17.93 | 19.79 | 7 | 2 | - |
| PTC-MR (Toivonen et al., 2003) | Chemical | 344 | 14.29 | 14.69 | 18 | 2 | 67 |
| PROTEINS (Yanardag & Vishwanathan, 2015) | Protein | 1113 | 39.06 | 72.82 | 3 | 2 | 218 |
| D&D (Shervashidze et al., 2011) | Protein | 1178 | 284.32 | 715.66 | 89 | 2 | 234 |
| NCI1 (Wale et al., 2008) | Chemical | 4110 | 29.87 | 32.30 | 37 | 2 | 744 |
| DHFR (Sutherland et al., 2003) | Protein | 756 | 42.43 | 44.54 | 53 | 2 | - |
| AIDS (Riesen & Bunke, 2008) | Chemical | 2000 | 15.69 | 16.20 | 38 | 2 | - |
| FRANKENSTEIN (Orsini et al., 2015) | Chemical | 4337 | 16.9 | 17.88 | - | 2 | 757 |
| REDDIT-B (Yanardag & Vishwanathan, 2015) | Social | 2000 | 429.63 | 497.75 | - | 2 | 400 |
| COLLAB (Leskovec et al., 2005) | Collaboration | 5000 | 74.49 | 2457.22 | - | 3 | 991 |
| Synthie (Morris et al., 2016) | Synthetic | 400 | 95.00 | 172.93 | - | 4 | - |
| IMDB-MULTI (Yanardag & Vishwanathan, 2015) | Social | 1500 | 13.00 | 65.94 | - | 3 | 272 |

#### G.1.2  AIRGRAPH DATASET

To evaluate the practicality of the model, we release an air quality graph dataset, **AirGraph**, for research on air pollution level prediction. **This dataset contains 30,660 graphs, significantly surpassing the size of most commonly used benchmark datasets in the graph learning domain.** Below, we provide a detailed introduction to the dataset.

**Data Source.**  The dataset was collected by the China National Environmental Monitoring Center, covering 1,341 monitoring sensors across 28 provinces in mainland China from 2021 to 2023, with hourly PM2.5 readings recorded.

**Graph Generating.**    We generated the graph structure and node features of the dataset based on the following rules to align with the graph classification task.

- **Node construction**: All monitoring stations within each province are treated as nodes in the graph;

- **Edge construction**: Edges are established based on the geographic distances between monitoring stations, reflecting spatial dependencies;

- **Node features**: Each node's features consist of the past 24 hours of PM2.5 observations, capturing temporal information;

- **Graph labels**: Each province-level graph label represents the air quality level corresponding to the average PM2.5 index of all sensors in the province over the next 24 hours, categorized into three levels: low pollution ($< 25$ $\mu$g/m$^3$), medium pollution (25–75 $\mu$g/m$^3$), and high pollution ($> 75$ $\mu$g/m$^3$), following international air quality standards.

Based on this processing procedure, we constructed a large-scale dataset comprising **30,660 graphs** (28 provinces × 365 days × 3 years), which significantly exceeds the scale of most commonly used benchmark datasets in the graph learning field.

**Graph Imbalance Characteristics.**    As shown in Figure 7, **AirGraph exhibits a naturally imbalanced distribution of pollution levels**: high pollution accounts for 6.86%, medium pollution for 42.84%, and low pollution for 50.30%, with the ratio between the largest and smallest classes reaching 7.33. Due to the heterogeneity in the distribution of monitoring stations across provinces, **the graph topologies in AirGraph also exhibit naturally imbalanced distributions**, providing a realistic and ideal resource for studying imbalanced graph learning problems.

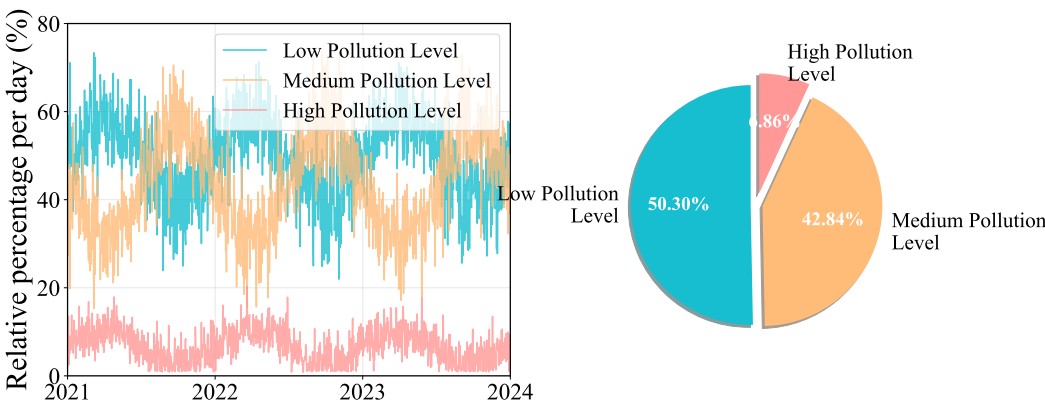

Figure 7: Distribution of Pollution Levels in the AirGraph Dataset: High (6.86%), Medium (42.84%), and Low (50.30%).

### G.1.3    PROCESSING OF CLASS IMBALANCE GRAPH DATASETS

**Dataset Setting.** We perform graph classification tasks on twelve modified Class Imbalanced graph datasets. Among them, nine datasets are used for binary classification tasks, and three datasets are used for multi-class classification tasks. Therefore, we adopt two different processing methods for the imbalance settings as follows. Final detailed class-imbalance dataset summary can be found in Table 9.

❶ For the **binary classification** task, we follow the setup in the G$^2$GNN paper (Wang et al., 2022b), where the dataset is split into 25% training and 25% validation, with the remaining samples assigned to the test set, resulting in specified numbers of graphs in each subset. We treat the class with more graphs as the head class and the other as the tail class. The imbalance ratio $\rho$ is defined as the ratio between the number of head and tail graphs. According to three different ratios — highly imbalanced setting $\rho \approx 10$ (**extreme**), moderately imbalanced setting $\rho \approx 2$ (**medium**), and balanced setting $\rho \approx 2$ (**low**) — we sample the desired number of graphs for the training and validation sets. All remaining graph samples are allocated to the test set.

❷ For the **multi-class classification** task, we allocate 25% of the dataset for training, 10% for validation, and the remaining 65% for testing, allowing us to obtain the desired sizes for each subset. Then, we sort the classes in a specific order and sample training graphs for each class following an exponential decay: $N_i = N_0\, e^{-\lambda t}$, where $N_0$ means the ratio of the largest class, which we set to 1. $N_i$ represents the number of sorted i-th class graphs. We control coefficient $\lambda$ to meet three different imbalance degree: highly imbalanced setting with $\rho \approx 100$ (**extreme**), moderately imbalanced setting with $\rho \approx 20$ (**medium**), and balanced setting with $\rho \approx 1$ (**low**). In the validation set, we maintain an equal number of samples across all classes. All remaining graph samples are assigned to the test set.

Table 9: Statistics of the *Class imbalance* datasets for graph classification. In the graph-level class imbalance setting, the imbalance ratio $\rho$ is defined as the ratio between the number of graphs in the Head Class and that in the Tail Class. For *binary classification* tasks, the validation set shares the same class distribution as the training set. For *multi-class classification* tasks, the number of graphs per class in the validation set is equal to ensure fair evaluation.

| Dataset | Task | Level | # Graphs (Head Class) | # Graphs (Tail Class) | Imbalance Ratio $\rho$ |
|---|---|---|---|---|---|
| MUATG (Debnath et al., 1991) | Binary | extreme | 45 | 5 | 9.00 |
| | | medium | 35 | 15 | 2.33 |
| | | low | 25 | 25 | 1.00 |
| PTC-MR (Toivonen et al., 2003) | Binary | extreme | 81 | 9 | 9.00 |
| | | medium | 63 | 27 | 2.33 |
| | | low | 45 | 45 | 1.00 |
| PROTEINS (Yanardag & Vishwanathan, 2015) | Binary | extreme | 270 | 30 | 9.00 |
| | | medium | 210 | 90 | 2.33 |
| | | low | 150 | 150 | 1.00 |
| D&D (Shervashidze et al., 2011) | Binary | extreme | 270 | 30 | 9.00 |
| | | medium | 210 | 90 | 2.33 |
| | | low | 150 | 150 | 1.00 |
| NCI1 (Wale et al., 2008) | Binary | extreme | 900 | 100 | 9.00 |
| | | medium | 700 | 300 | 2.33 |
| | | low | 500 | 500 | 1.00 |
| DHFR (Sutherland et al., 2003) | Binary | extreme | 108 | 12 | 9.00 |
| | | medium | 84 | 36 | 2.33 |
| | | low | 60 | 60 | 1.00 |
| AIDS (Riesen & Bunke, 2008) | Binary | extreme | 450 | 50 | 9.00 |
| | | medium | 350 | 150 | 2.33 |
| | | low | 250 | 250 | 1.00 |
| FRANKENSTEIN (Orsini et al., 2015) | Binary | extreme | 900 | 100 | 9.00 |
| | | medium | 700 | 300 | 2.33 |
| | | low | 500 | 500 | 1.00 |
| REDDIT-B (Yanardag & Vishwanathan, 2015) | Binary | extreme | 450 | 50 | 9.00 |
| | | medium | 350 | 150 | 2.33 |
| | | low | 250 | 250 | 1.00 |
| COLLAB (Leskovec et al., 2005) | Multi-Class | extreme | 1100 | 11 | 100.00 |
| | | medium | 980 | 49 | 20.00 |
| | | low | 417 | 417 | 1.00 |
| Synthie (Morris et al., 2016) | Multi-Class | extreme | 100 | 1 | 100.00 |
| | | medium | 60 | 3 | 20.00 |
| | | low | 25 | 25 | 1.00 |
| IMDB-MULTI (Yanardag & Vishwanathan, 2015) | Multi-Class | extreme | 300 | 3 | 100.00 |
| | | medium | 300 | 15 | 20.00 |
| | | low | 125 | 125 | 1.00 |

### G.1.4 PROCESSING OF TOPOLOGICAL IMBALANCE GRAPH DATASETS

**Dataset Settings.** We perform graph classification tasks on eight modified Topological Imbalanced graph datasets, including six for binary classification and two for multi-class classification. Following the experimental settings of IGL-Bench (Qin et al., 2024), we adopt a unified processing approach for both binary and multi-class graph classification tasks. **Since the test sets of some datasets may not include all classes when generating topological imbalance data under the predefined settings, four datasets are excluded from the evaluation on topological imbalance performance.** The detailed topological imbalance dataset settings are summarized in Table 10.

Specifically, following the setting in the f IGL-Bench(Qin et al., 2024), we partition the dataset into training and validation sets with a 10%/10% split, while the remaining 80% of samples are assigned to the test set. To ensure fair evaluation under imbalanced graph distributions, we adopt a class-balanced splitting strategy, where each class is represented by an equal number of graphs in both the training and validation sets. As formally defined in Definition 2, we define the head graph subset as the top 20 % of graphs with the largest number of nodes, while the remaining 80%

constitute the tail graph subset (Sanders, 1987). According to the desired imbalance ratio $\rho$, we sample from the two subsets to construct the corresponding training set. To comprehensively evaluate the robustness of UniImb under varying degrees of topology-imbalance, we create three representative splits with different levels of balance: extreme, medium, and low. Due to the complexity of the graph size distribution, it is challenging to strictly satisfy the target sparsity ratio in practice. After multiple attempts, we obtain the final splits whose imbalance levels closely approximate the intended scenarios. The resulting topology-imbalance statistics are summarized in Table 10, reflecting the practical difficulty in achieving exact sparsity control under real-world graph distributions.

Table 10: Statistics of the modified ***Topological Imbalance*** datasets for graph classification. In the graph-level topological imbalance setting, the imbalance ratio $\rho$ is defined as the ratio between the average size of the head graphs and the average size of the tail graphs. To ensure class balance, the number of graphs per class is equal in both the training and validation sets.

| Dataset | Task | Level | # Head Graphs (per Class) | # Tail Graphs (per Class) | Avg. # Size (Head Graphs) | Avg. # Size (Tail Graphs) | Imbalance Ratio $\rho$ |
|---|---|---|---|---|---|---|---|
| PTC-MR (Toivonen et al., 2003) | Binary | extreme medium low | 2 | 15 | 56.00 34.25 29.50 | 11.37 11.80 18.4 | 4.93 2.90 1.60 |
| PROTEINS (Yanardag & Vishwanathan, 2015) | Binary | extreme medium low | 6 | 50 | 276.00 90.25 93.50 | 22.76 22.26 45.81 | 12.13 4.05 2.04 |
| D&D (Shervashidze et al., 2011) | Binary | extreme medium low | 6 | 53 | 1765.00 524.83 548.00 | 198.11 207.7 350.58 | 8.91 2.53 1.56 |
| NCI1 (Wale et al., 2008) | Binary | extreme medium low | 21 | 185 | 83.62 49.90 50.48 | 24.82 25.03 36.06 | 3.37 1.99 1.4 |
| FRANKENSTEIN (Orsini et al., 2015) | Binary | extreme medium low | 22 | 195 | 77.55 32.43 33.09 | 13.78 13.78 21.19 | 5.63 2.35 1.56 |
| REDDIT-B (Yanardag & Vishwanathan, 2015) | Binary | extreme medium low | 10 | 90 | 2442.70 1097.75 1180.75 | 222.73 223.04 423.23 | 10.97 4.92 2.79 |
| COLLAB (Leskovec et al., 2005) | Multi-Class | extreme medium low | 17 | 150 | 309.41 147.22 141.41 | 50.77 50.04 80.14 | 6.09 2.94 1.76 |
| IMDB-MULTI (Yanardag & Vishwanathan, 2015) | Multi-Class | extreme medium low | 5 | 45 | 56.33 25.33 26.13 | 10.10 9.95 15.63 | 5.58 2.55 1.67 |

### G.1.5 PROCESSING OF DATASETS WITH INTERTWINED CLASS AND TOPOLOGICAL IMBALANCES

**Dataset Settings.** To simulate the complex scenario of Class Imbalance and Topological Imbalance interwoven in real-world graph classification tasks, we conducted experiments on six modified graph datasets that exhibit both class-imbalance and topological-imbalance. These datasets include four binary classification datasets and two multi-class classification dataset. We divided both the training and validation sets with a 10%/10% ratio, and the remaining samples were allocated to the test set. For these two types of datasets, we adopted two different processing methods. The detailed dataset configurations are provided in Table 11.

❶ For the **binary classification** task, we adopt a Class Imbalance ratio similar to that used in G²GNN. Specifically, the imbalance ratio in the training and validation sets is set to 1:9, with the remaining samples allocated to the test set. Within each class, the topology-imbalance ratio is defined as the ratio of the average graph size (in terms of node count) between the head set and the tail set, as outlined in **Definition 2**. To achieve this, we employ the Pareto principle (20%/80%) to partition the dataset into head and tail sets. Consequently, for each class in the training set, we allocate 10% of the graphs to the head set and 90% to the tail set based on their graph size, thus creating a scenario where both class-imbalance and topology-imbalance are interwoven.

❷ For **multi-class classification** task, similar to the aforementioned class-imbalance setting, we assume that the number of graph samples per class in the training set follows an exponential decay distribution. Given the total number of training samples and a class-imbalance ratio $\rho$, we determine the number of samples per class such that the ratio between the Head Class and the combined Tail Classes is 9:1, thereby creating an extremely Class Imbalanced training environment. Building on this, we further introduce Topological Imbalance within each class. Specifically, we rank the graphs in each class by their size (i.e., number of nodes), and partition them into head and tail subsets

with a 1:9 ratio. That is, only 10% of the graphs in each class belong to the structurally complex head subset, while the remaining 90% belong to the simpler tail subset. This strategy enables us to construct a challenging experimental scenario where class-imbalance and topological-imbalance are interwoven. Investigating such intertwined imbalance settings is of significant importance, as it closely mirrors the complex distributions commonly encountered in real-world graph classification tasks. In many practical applications, such as social network analysis, protein-protein interaction prediction, and recommender systems, class-imbalance often coexists with disparities in structural complexity—some classes consist of graphs with more nodes and complex structures, while others are relatively simple. By conducting experiments under this intertwined imbalance setting, we can comprehensively evaluate the robustness and generalization ability of UniImb in realistic and challenging scenarios.

Table 11: Statistics of the modified ***Intertwined Class and Topological Imbalanced*** datasets for graph classification. In the graph-level class-imbalance setting, the imbalance ratio $\rho$ is defined as the ratio between the number of graphs in the Head Class and that in the Tail Class. And in the graph-level topology-imbalance setting, the imbalance ratio $\rho$ is defined as the ratio between the average size of the head graphs and the average size of the tail graphs.

| Dataset | Task | # Graphs (Head Class) | | # Graphs (Tail Class) | | Avg. # Size (Head Graphs) | Avg. # Size (Tail Graphs) | Imbalance Ratio $\rho$ (class & topology) |
|---|---|---|---|---|---|---|---|---|
| | | # Head Graphs | # Tail Graphs | # Head Graphs | # Tail Graphs | | | |
| PROTEINS (Yanardag & Vishwanathan, 2015) | Binary | 10 | 90 | 1 | 10 | 103.91 | 27.53 | 9.09 & 3.77 |
| D&D (Shervashidze et al., 2011) | Binary | 11 | 95 | 1 | 10 | 538.82 | 165.68 | 9.64 & 3.25 |
| NCI1 (Wale et al., 2008) | Binary | 37 | 333 | 4 | 37 | 52.27 | 26.45 | 9.02 & 1.98 |
| REDDIT-B (Yanardag & Vishwanathan, 2015) | Binary | 18 | 162 | 2 | 18 | 874.10 | 164.13 | 9.00 & 5.33 |
| COLLAB (Leskovec et al., 2005) | Multi-Class | 50 | 450 | 1 | 4 | 150.35 | 52.58 | 100.00 & 2.86 |
| IMDB-MULTI (Yanardag & Vishwanathan, 2015) | Multi-Class | 10 | 90 | 0 | 1 | 30.64 | 10.55 | 100.00 & 2.90 |

## H EXPERIMENT SETTING

### H.1 HYPERPARAMETER SETTING

We summarize in Table 13 the detailed hyperparameter configurations of UniImb across the six datasets discussed in the main text. To ensure a fair comparison, the remaining models are based on publicly available optimal hyperparameters.

Table 12: Detailed hyperparameters of UniImb with Class Imbalance datasets.

| Dataset | Learning_rate | Hidden_dim | GNN_layers L | Batch_size | K | TopK$_1$ | TopK$_2$ |
|---|---|---|---|---|---|---|---|
| PROTEINS (Yanardag & Vishwanathan, 2015) | 0.001 | 128 | 5 | 32 | 16 | 16 | 8 |
| D&D (Shervashidze et al., 2011) | 0.001 | 128 | 5 | 64 | 16 | 32 | 8 |
| NCI1 (Wale et al., 2008) | 0.001 | 128 | 5 | 64 | 24 | 32 | 12 |
| REDDIT-B (Yanardag & Vishwanathan, 2015) | 0.001 | 128 | 5 | 64 | 32 | 32 | 16 |
| COLLAB (Leskovec et al., 2005) | 0.001 | 128 | 5 | 64 | 32 | 32 | 16 |
| IMDB-MULTI (Yanardag & Vishwanathan, 2015) | 0.001 | 128 | 5 | 32 | 32 | 16 | 8 |

Table 13: Detailed hyperparameters of UniImb with Topological Imbalance datasets.

| Dataset | Learning_rate | Hidden_dim | GNN_layers L | Batch_size | K | TopK$_1$ | TopK$_2$ |
|---|---|---|---|---|---|---|---|
| PROTEINS (Yanardag & Vishwanathan, 2015) | 0.001 | 128 | 5 | 32 | 16 | 24 | 8 |
| D&D (Shervashidze et al., 2011) | 0.001 | 128 | 5 | 64 | 8 | 32 | 4 |
| NCI1 (Wale et al., 2008) | 0.001 | 128 | 5 | 64 | 8 | 32 | 4 |
| REDDIT-B (Yanardag & Vishwanathan, 2015) | 0.001 | 128 | 5 | 64 | 32 | 32 | 16 |
| COLLAB (Leskovec et al., 2005) | 0.001 | 128 | 5 | 64 | 16 | 32 | 8 |
| IMDB-MULTI (Yanardag & Vishwanathan, 2015) | 0.001 | 128 | 5 | 32 | 32 | 16 | 8 |

### H.2 EVALUATION METRICS

Following existing work in Class Imbalanced graph classification, we adopt two evaluation metrics, Macro-F1 and Micro-F1 (Wang et al., 2022b), to assess the performance of UniImb and other baseline models. Macro-F1 calculates the accuracy for each class independently and then averages these values,

treating all classes equally. In contrast, Micro-F1 computes accuracy over all samples collectively, which may result in the head class dominating the evaluation. A more detailed explanation of these evaluation metrics can be found as follows.

❶ **Macro-F1**(Xia et al., 2014) is a commonly used metric for evaluating multi-class classification performance. It computes the F1 score for each class independently and then averages them, treating all classes equally regardless of their size. The formula is given by:

$$\text{Macro-F1} = \frac{1}{N} \sum_{i=1}^{N} \frac{2 \cdot \text{Precision}_i \cdot \text{Recall}_i}{\text{Precision}_i + \text{Recall}_i} \tag{46}$$

where $\text{Precision}_i = \frac{TP_i}{TP_i + FP_i}$ and $\text{Recall}_i = \frac{TP_i}{TP_i + FN_i}$. Macro-F1 equally considers all classes, regardless of their frequency, and emphasizes the balance between precision and recall. It is especially suitable for imbalanced datasets. Since each class contributes equally, Macro-F1 can be overly influenced by rare classes and may not reflect the overall performance if class distribution is extremely skewed.

❷ **Micro-F1** (Lipton et al., 2014) aggregates the contributions of all classes to compute global precision and recall, and then calculates the F1 score. Unlike Macro-F1, it does not treat classes separately but considers the total number of true positives, false positives, and false negatives across all classes. The formula is:

$$\text{Micro-F1} = \frac{2 \cdot \text{Precision} \cdot \text{Recall}}{\text{Precision} + \text{Recall}} \tag{47}$$

where $\text{Precision} = \frac{\sum_i TP_i}{\sum_i (TP_i + FP_i)}$ and $\text{Recall} = \frac{\sum_i TP_i}{\sum_i (TP_i + FN_i)}$. Micro-F1 provides a global measure of performance and is robust to class imbalance, as it gives more weight to larger classes. Therefore, it may overlook poor performance on minority classes, as these contribute less to the overall count of predictions.

# I ADDITIONAL EXPERIMENTAL RESULTS

## I.1 CLASS IMBALANCE PERFORMANCE ANALYSIS

### I.1.1 COMPLETE EXPERIMENTS ON ALL DATASETS WITH VARING IMBALANCE DEGREE

The performance on the 12 class-imbalanced datasets is presented in Table 14, Figure 16, and Table 16. Classic graph representation models perform the worst due to their inability to address the challenge of class-imbalance. Oversampling methods involve resampling tail classes, which may lead to overfitting of tail features and consequently degrade generalization performance. DataDec identifies an informative subset for model training through dynamic sparse graph contrastive learning and leverages abundant unlabeled data to achieve limited improvements. **It is worth noting that the original code of DataDec provided by its authors uses the test set for classifier fine-tuning. For a fair comparison, we only use the training set for fine-tuning, which may result in performance slightly lower than what was originally reported.** TopoImb and ImGKB, as well as G$^2$GNN and ImbGNN, improve representation learning under class-imbalance by selecting similar graphs for data augmentation within tail classes, thereby achieving higher prediction accuracy. Our model achieves the best performance across all metrics on the class-imbalanced datasets. This superior performance is attributed to our prototype-based strategy, which effectively enhances the model's ability to learn robust and discriminative representations from class-imbalanced graph data.

We further compare our model with baseline models under scenarios of medium and low class-imbalance (i.e., close to balanced). As shown in Table 15 and Table 16, we find that as the class distribution approaches balance, classic graph classification models perform increasingly better, while models specifically designed for imbalanced settings tend to become less effective. For instance, G$^2$GNN constructs the graphs of graphs that can easily lead to information redundancy, thereby degrading performance. In contrast, DataDec performs increasingly well and becomes the best-performing baseline in the low-imbalance setting. On the other hand, our model achieves consistently strong performance across all scenarios, demonstrating its robustness in handling more complex and varying class distributions.

Table 14: Macro-F1 and Micro-F1 score on *class imbalance* datasets with *extreme imbalance degree*. The best results are marked and the runner-ups are underlined. We report the average and standard deviation over 20 runs.

| Model | Backbone | MUTAG Macro-F1 | MUTAG Micro-F1 | PROTEINS Macro-F1 | PROTEINS Micro-F1 | D&D Macro-F1 | D&D Micro-F1 | NCI1 Macro-F1 | NCI1 Micro-F1 | PTC-MR Macro-F1 | PTC-MR Micro-F1 | DHFR Macro-F1 | DHFR Micro-F1 |
|---|---|---|---|---|---|---|---|---|---|---|---|---|---|
| vanilla | GIN | 52.50 ± 18.70 | 56.77 ± 14.14 | 25.33 ± 7.53 | 28.50 ± 5.82 | 9.99 ± 7.44 | 11.88 ± 9.49 | 18.24 ± 7.58 | 18.94 ± 7.12 | 17.74 ± 6.49 | 20.30 ± 6.06 | 35.96 ± 8.87 | 49.46 ± 4.90 |
| | InfoGraph | 69.11 ± 9.03 | 69.68 ± 7.77 | 35.91 ± 7.58 | 36.81 ± 6.51 | 21.41 ± 4.51 | 27.68 ± 7.52 | 33.09 ± 3.30 | 34.03 ± 3.68 | 25.85 ± 6.14 | 26.71 ± 6.50 | 50.62 ± 4.83 | 56.28 ± 4.58 |
| | GraphCL | 66.82 ± 11.56 | 67.77 ± 9.78 | 40.86 ± 6.94 | 41.24 ± 6.38 | 21.02 ± 3.05 | 26.80 ± 4.95 | 31.02 ± 2.69 | 31.62 ± 3.05 | 24.22 ± 6.21 | 25.16 ± 5.25 | 50.55 ± 10.01 | 56.31 ± 6.12 |
| GTs | GraphGPS | 36.87 ± 7.04 | 44.55 ± 4.18 | 25.79 ± 7.05 | 28.71 ± 5.46 | 10.12 ± 4.41 | 11.97 ± 3.91 | 14.94 ± 2.41 | 15.62 ± 2.07 | 22.53 ± 3.79 | 23.66 ± 3.20 | 41.16 ± 7.23 | 51.65 ± 3.86 |
| | Exphormer | 40.19 ± 7.62 | 46.59 ± 4.85 | 25.52 ± 4.79 | 28.38 ± 3.57 | 9.79 ± 4.18 | 10.85 ± 4.28 | 14.56 ± 3.92 | 15.36 ± 3.60 | 23.64 ± 3.71 | 24.57 ± 3.10 | 47.49 ± 11.56 | 55.33 ± 6.50 |
| | Garph-Mamba | 46.61 ± 13.76 | 51.82 ± 10.04 | 31.12 ± 5.30 | 32.79 ± 4.02 | 4.99 ± 7.93 | 6.12 ± 10.36 | 14.11 ± 3.26 | 14.94 ± 3.82 | 23.28 ± 3.24 | 24.21 ± 2.59 | 45.05 ± 8.74 | 53.76 ± 5.09 |
| up-sampling | GIN | 78.03 ± 7.62 | 78.77 ± 7.67 | 65.64 ± 2.67 | 71.55 ± 3.19 | 41.15 ± 3.74 | 70.56 ± 10.28 | 59.19 ± 4.39 | 71.80 ± 7.02 | 44.78 ± 8.01 | 55.43 ± 14.25 | 55.96 ± 10.06 | 59.39 ± 6.52 |
| | InfoGraph | 78.62 ± 6.84 | 79.09 ± 6.86 | 62.68 ± 2.70 | 66.02 ± 3.18 | 41.55 ± 2.22 | 71.34 ± 6.76 | 53.38 ± 1.88 | 62.20 ± 2.63 | 44.29 ± 4.69 | 48.91 ± 7.49 | 59.49 ± 5.20 | 61.62 ± 4.18 |
| | GraphCL | 80.06 ± 7.79 | 80.45 ± 7.86 | 64.21 ± 2.53 | 65.76 ± 2.61 | 38.96 ± 3.01 | 64.23 ± 8.10 | 49.92 ± 2.15 | 58.29 ± 3.30 | 45.12 ± 7.33 | 53.50 ± 13.31 | 60.29 ± 9.04 | 61.71 ± 6.75 |
| re-weight | GIN | 77.00 ± 9.59 | 77.68 ± 9.30 | 54.54 ± 6.29 | 55.77 ± 7.11 | 28.49 ± 5.92 | 40.79 ± 11.84 | 36.84 ± 8.46 | 39.19 ± 10.05 | 36.96 ± 14.08 | 43.09 ± 20.01 | 55.16 ± 9.47 | 57.78 ± 6.69 |
| | InfoGraph | 80.85 ± 7.75 | 81.68 ± 7.83 | 65.73 ± 3.10 | 69.60 ± 3.68 | 41.92 ± 2.28 | 72.43 ± 6.63 | 53.05 ± 1.12 | 62.45 ± 1.89 | 44.09 ± 5.62 | 49.17 ± 8.78 | 58.67 ± 5.82 | 60.24 ± 4.80 |
| | GraphCL | 80.20 ± 7.27 | 80.84 ± 7.43 | 63.46 ± 2.42 | 64.97 ± 2.41 | 40.29 ± 3.31 | 67.96 ± 8.98 | 50.05 ± 2.09 | 58.18 ± 3.08 | 44.75 ± 7.62 | 52.22 ± 13.24 | 60.87 ± 6.33 | 61.93 ± 5.15 |
| G²GNN | remove edge | 80.37 ± 6.73 | 81.25 ± 6.87 | 67.70 ± 2.96 | 73.10 ± 4.05 | 43.25 ± 3.91 | 77.03 ± 9.98 | 63.60 ± 1.57 | 72.97 ± 1.81 | 46.40 ± 7.73 | 56.61 ± 13.72 | 61.63 ± 10.02 | 63.61 ± 6.05 |
| | mask node | 83.01 ± 7.01 | 83.59 ± 7.14 | 67.39 ± 2.99 | 73.30 ± 4.19 | 43.93 ± 3.46 | 79.03 ± 10.78 | 64.78 ± 2.86 | 74.91 ± 2.14 | 46.61 ± 8.27 | 56.70 ± 14.81 | 59.72 ± 6.83 | 61.27 ± 5.40 |
| TopoImb | dynamic sparsity | 54.53 ± 9.91 | 59.51 ± 8.57 | 53.95 ± 6.68 | 56.00 ± 7.88 | 7.72 ± 5.68 | 9.47 ± 4.26 | 16.41 ± 5.19 | 17.14 ± 5.68 | 29.32 ± 6.76 | 30.61 ± 6.22 | 36.02 ± 9.17 | 49.10 ± 5.85 |
| DataDec | | 52.63 ± 14.62 | 55.91 ± 10.97 | 29.48 ± 3.98 | 31.25 ± 2.98 | 15.79 ± 2.38 | 18.86 ± 3.47 | 18.14 ± 3.86 | 18.52 ± 3.64 | 24.25 ± 5.15 | 25.06 ± 3.33 | 42.35 ± 7.94 | 51.94 ± 4.18 |
| ImGKB | | 54.02 ± 7.82 | 55.74 ± 6.04 | 53.99 ± 7.22 | 55.31 ± 8.17 | 31.15 ± 6.29 | 46.31 ± 11.72 | 32.93 ± 5.46 | 34.85 ± 7.14 | 30.71 ± 10.51 | 32.23 ± 10.90 | 41.51 ± 7.19 | 49.07 ± 3.11 |
| ImbGNN | | 84.89 ± 7.24 | 85.23 ± 7.46 | 67.69 ± 2.95 | 73.24 ± 3.85 | 46.06 ± 7.23 | 83.24 ± 19.03 | 65.48 ± 3.39 | 74.58 ± 5.49 | 47.34 ± 4.91 | 59.33 ± 12.99 | 61.80 ± 8.06 | 61.82 ± 4.57 |
| **UniImb** | / | **86.70 ± 4.61** | **87.06 ± 4.68** | **70.44 ± 4.72** | **74.50 ± 4.99** | **46.63 ± 3.42** | **83.60 ± 6.50** | **68.30 ± 5.19** | **80.68 ± 4.22** | **47.54 ± 8.30** | **59.70 ± 13.54** | **66.24 ± 4.65** | **67.05 ± 4.06** |

| Model | Backbone | REDDIT-B Macro-F1 | REDDIT-B Micro-F1 | AIDS Macro-F1 | AIDS Micro-F1 | FRANKENSTEIN Macro-F1 | FRANKENSTEIN Micro-F1 | COLLAB Macro-F1 | COLLAB Micro-F1 | IMDB-MULTI Macro-F1 | IMDB-MULTI Micro-F1 | Synthie Macro-F1 | Synthie Micro-F1 |
|---|---|---|---|---|---|---|---|---|---|---|---|---|---|
| vanilla | GIN | 33.19 ± 14.26 | 36.02 ± 17.38 | 96.42 ± 0.27 | 97.68 ± 0.62 | 20.45 ± 6.72 | 25.72 ± 9.80 | 32.58 ± 3.66 | 57.31 ± 4.12 | 13.25 ± 6.19 | 14.92 ± 5.43 | 33.58 ± 1.88 | 33.54 ± 0.39 |
| | InfoGraph | 57.67 ± 3.80 | 67.10 ± 4.91 | 97.16 ± 1.34 | 97.89 ± 1.78 | 35.64 ± 4.89 | 40.16 ± 2.67 | 43.48 ± 4.29 | 59.10 ± 4.88 | 17.28 ± 7.28 | 29.18 ± 4.47 | 35.08 ± 2.90 | 36.13 ± 2.47 |
| | GraphCL | 53.40 ± 4.06 | 62.19 ± 5.68 | 97.26 ± 2.43 | 98.02 ± 2.90 | 36.73 ± 3.86 | 42.16 ± 2.91 | 45.02 ± 5.61 | 60.22 ± 3.47 | 16.30 ± 9.22 | 32.18 ± 8.90 | 34.47 ± 2.78 | 35.82 ± 1.97 |
| GTs | GraphGPS | 11.68 ± 7.76 | 12.71 ± 8.13 | 74.39 ± 13.58 | 79.10 ± 7.65 | 21.13 ± 0.57 | 26.15 ± 0.36 | 25.58 ± 12.93 | 39.92 ± 13.06 | 14.20 ± 5.61 | 28.54 ± 13.78 | 11.16 ± 8.53 | 18.81 ± 14.85 |
| | Exphormer | 22.68 ± 10.79 | 27.33 ± 21.65 | 60.62 ± 6.50 | 76.10 ± 3.84 | 21.01 ± 0.46 | 26.07 ± 0.30 | 32.61 ± 17.44 | 42.02 ± 15.44 | 20.81 ± 5.43 | 28.14 ± 8.30 | 15.15 ± 4.36 | 32.95 ± 5.39 |
| | Garph-Mamba | 15.27 ± 12.46 | 17.02 ± 14.71 | 99.09 ± 0.44 | 99.24 ± 0.36 | 22.28 ± 2.45 | 26.92 ± 1.68 | 42.53 ± 11.15 | 50.63 ± 5.38 | 16.89 ± 4.57 | 28.69 ± 12.47 | 31.98 ± 9.75 | 44.77 ± 8.45 |
| up-sampling | GIN | 66.71 ± 3.92 | 83.00 ± 5.18 | 98.54 ± 0.39 | 98.78 ± 0.32 | 63.93 ± 8.65 | 68.80 ± 6.46 | 64.30 ± 2.67 | 66.10 ± 3.28 | 22.27 ± 10.01 | 38.32 ± 10.04 | 40.09 ± 2.58 | 43.37 ± 9.88 |
| | InfoGraph | 67.01 ± 3.34 | 78.68 ± 3.71 | 98.67 ± 0.89 | 99.01 ± 1.45 | 59.78 ± 5.75 | 67.62 ± 6.86 | 63.28 ± 2.90 | 65.14 ± 3.29 | 21.79 ± 6.68 | 37.29 ± 7.02 | 39.33 ± 4.12 | 41.29 ± 4.29 |
| | GraphCL | 62.01 ± 3.97 | 75.84 ± 3.98 | 97.98 ± 1.65 | 98.79 ± 2.16 | 62.17 ± 7.37 | 66.14 ± 5.64 | 64.57 ± 5.20 | 66.79 ± 4.11 | 23.62 ± 6.91 | 40.29 ± 6.90 | 41.28 ± 5.10 | 43.48 ± 4.27 |
| re-weight | GIN | 45.17 ± 8.46 | 51.92 ± 12.29 | 97.45 ± 1.64 | 98.73 ± 1.25 | 22.89 ± 8.17 | 26.57 ± 7.43 | 57.83 ± 3.03 | 60.09 ± 4.59 | 22.07 ± 11.13 | 36.69 ± 11.14 | 36.14 ± 3.42 | 47.05 ± 12.96 |
| | InfoGraph | 65.79 ± 3.38 | 77.35 ± 3.96 | 98.65 ± 2.10 | 98.79 ± 1.43 | 58.62 ± 6.12 | 67.89 ± 5.60 | 62.22 ± 3.90 | 64.48 ± 3.14 | 21.16 ± 6.79 | 38.02 ± 5.70 | 38.20 ± 4.02 | 46.20 ± 3.90 |
| | GraphCL | 62.79 ± 6.93 | 76.15 ± 9.15 | 97.78 ± 3.24 | 98.89 ± 2.40 | 63.19 ± 5.14 | 66.10 ± 6.71 | 63.18 ± 4.55 | 65.29 ± 3.87 | 22.48 ± 6.82 | 39.57 ± 5.89 | 39.46 ± 6.27 | 47.00 ± 3.39 |
| G²GNN | remove edge | 68.39 ± 2.97 | 86.35 ± 2.27 | 98.66 ± 0.29 | 98.87 ± 0.24 | 64.46 ± 11.05 | 70.64 ± 11.83 | 38.93 ± 3.22 | 54.98 ± 4.28 | 20.67 ± 9.88 | 36.89 ± 11.73 | 41.08 ± 6.59 | 45.99 ± 11.17 |
| | mask node | 67.52 ± 2.60 | 85.43 ± 1.80 | 98.90 ± 0.28 | 98.84 ± 0.24 | 64.20 ± 9.34 | 71.80 ± 11.03 | 37.63 ± 5.19 | 53.92 ± 6.37 | 21.54 ± 9.49 | 35.78 ± 11.08 | 41.29 ± 6.90 | 47.76 ± 13.31 |
| TopoImb | dynamic sparsity | 10.38 ± 2.94 | 11.20 ± 2.69 | 49.48 ± 4.97 | 72.67 ± 1.68 | 27.12 ± 10.14 | 40.39 ± 12.20 | 19.24 ± 0.02 | 40.71 ± 0.05 | 9.37 ± 1.51 | 15.24 ± 0.90 | 17.75 ± 6.89 | 29.77 ± 11.95 |
| DataDec | | 56.11 ± 4.65 | 65.35 ± 6.16 | 89.71 ± 3.25 | 91.88 ± 2.46 | 23.40 ± 1.39 | 27.64 ± 0.92 | 41.73 ± 1.20 | 57.27 ± 0.65 | 11.08 ± 1.15 | 16.29 ± 0.72 | 22.24 ± 1.45 | 33.71 ± 1.49 |
| ImGKB | | 11.00 ± 8.34 | 14.00 ± 1.34 | 79.90 ± 2.94 | 83.71 ± 2.99 | 21.58 ± 4.83 | 28.15 ± 10.58 | 18.84 ± 1.96 | 39.58 ± 4.90 | 16.53 ± 5.23 | 34.12 ± 12.71 | 15.42 ± 5.87 | 22.35 ± 10.09 |
| ImbGNN | | 68.36 ± 8.10 | 86.56 ± 4.77 | 98.79 ± 0.62 | 99.03 ± 0.42 | 63.80 ± 6.28 | 72.58 ± 6.70 | 56.62 ± 4.28 | 62.79 ± 3.83 | 17.52 ± 8.98 | 34.54 ± 8.24 | 42.16 ± 6.25 | 47.72 ± 13.24 |
| **UniImb** | / | **76.24 ± 4.09** | **88.82 ± 2.93** | **99.70 ± 0.25** | **99.75 ± 0.21** | **68.74 ± 6.78** | **74.29 ± 5.92** | **75.73 ± 2.52** | **76.34 ± 2.60** | **33.45 ± 7.83** | **45.72 ± 4.87** | **44.38 ± 3.12** | **66.35 ± 0.75** |

Table 15: Macro-F1 and Micro-F1 score on *class imbalance* datasets with *medium imbalance degree*.

| Model | Backbone | MUTAG Macro-F1 | MUTAG Micro-F1 | PROTEINS Macro-F1 | PROTEINS Micro-F1 | D&D Macro-F1 | D&D Micro-F1 | NCI1 Macro-F1 | NCI1 Micro-F1 | PTC-MR Macro-F1 | PTC-MR Micro-F1 | DHFR Macro-F1 | DHFR Micro-F1 |
|---|---|---|---|---|---|---|---|---|---|---|---|---|---|
| vanilla | GIN | 78.77 ± 4.09 | 80.19 ± 3.46 | 65.75 ± 2.41 | 66.82 ± 2.06 | 46.20 ± 5.14 | 51.31 ± 7.12 | 58.76 ± 2.93 | 58.93 ± 3.08 | 31.52 ± 5.39 | 41.54 ± 2.97 | 37.82 ± 6.42 | 57.23 ± 1.78 |
| | InfoGraph | 79.94 ± 5.15 | 81.07 ± 3.44 | 66.18 ± 3.30 | 67.12 ± 3.29 | 52.11 ± 3.90 | 60.28 ± 3.44 | 60.67 ± 2.24 | 62.11 ± 3.27 | 34.68 ± 4.10 | 44.20 ± 3.19 | 49.97 ± 7.20 | 56.14 ± 6.49 |
| | GraphCL | 79.23 ± 6.36 | 80.25 ± 4.09 | 66.90 ± 4.14 | 67.46 ± 3.99 | 53.57 ± 3.89 | 61.95 ± 4.03 | 58.79 ± 3.99 | 59.90 ± 4.47 | 32.90 ± 4.97 | 43.19 ± 4.50 | 48.33 ± 10.90 | 57.01 ± 8.82 |
| GTs | GraphGPS | 66.00 ± 7.45 | 72.84 ± 4.14 | 57.99 ± 6.03 | 61.07 ± 4.14 | 36.88 ± 13.24 | 41.97 ± 18.39 | 54.93 ± 3.57 | 55.04 ± 3.46 | 41.87 ± 6.22 | 46.52 ± 2.98 | 66.25 ± 6.19 | 69.75 ± 3.84 |
| | Exphormer | 72.23 ± 11.19 | 76.14 ± 5.11 | 59.72 ± 2.19 | 63.73 ± 4.41 | 37.67 ± 7.11 | 40.86 ± 9.52 | 56.72 ± 4.27 | 59.39 ± 2.92 | 44.67 ± 7.32 | 47.44 ± 4.86 | 63.18 ± 6.27 | 67.33 ± 5.93 |
| | Graph-Mamba | 70.65 ± 12.77 | 76.59 ± 6.59 | 65.21 ± 3.62 | 66.59 ± 2.69 | 40.26 ± 4.12 | 45.82 ± 4.52 | 64.61 ± 4.54 | 65.09 ± 4.99 | 43.16 ± 7.51 | 48.84 ± 5.65 | 65.16 ± 4.72 | 67.91 ± 3.20 |
| up-sampling | GIN | 80.78 ± 4.95 | 81.24 ± 5.14 | 67.80 ± 2.89 | 68.17 ± 2.89 | 56.63 ± 4.65 | 67.24 ± 6.88 | 70.82 ± 1.09 | 73.83 ± 1.89 | 37.17 ± 9.53 | 46.55 ± 6.99 | 55.22 ± 8.37 | 58.67 ± 7.54 |
| | InfoGraph | 81.02 ± 3.03 | 82.06 ± 2.37 | 66.56 ± 3.40 | 67.20 ± 4.44 | 57.78 ± 3.80 | 68.22 ± 3.92 | 68.12 ± 2.40 | 71.75 ± 3.40 | 36.66 ± 2.13 | 46.02 ± 4.01 | 54.28 ± 7.72 | 59.22 ± 11.03 |
| | GraphCL | 81.66 ± 4.47 | 82.38 ± 3.76 | 67.57 ± 5.90 | 68.09 ± 4.59 | 55.19 ± 6.17 | 66.15 ± 5.90 | 66.15 ± 5.90 | 68.91 ± 2.78 | 38.01 ± 4.12 | 46.49 ± 5.53 | 56.07 ± 8.83 | 62.14 ± 7.05 |
| re-weight | GIN | 78.85 ± 4.88 | 80.34 ± 3.98 | 67.37 ± 3.26 | 68.36 ± 2.60 | 51.28 ± 6.39 | 58.98 ± 9.24 | 62.34 ± 2.18 | 62.72 ± 2.46 | 40.94 ± 7.30 | 45.81 ± 4.21 | 45.57 ± 11.87 | 59.07 ± 4.59 |
| | InfoGraph | 80.17 ± 5.61 | 81.20 ± 4.05 | 67.98 ± 4.46 | 68.79 ± 5.01 | 53.20 ± 4.76 | 62.70 ± 4.23 | 67.18 ± 4.99 | 70.89 ± 3.57 | 42.80 ± 3.13 | 46.54 ± 3.34 | 49.99 ± 7.14 | 60.62 ± 5.70 |
| | GraphCL | 81.29 ± 6.03 | 81.99 ± 4.97 | 67.04 ± 6.44 | 68.19 ± 5.18 | 56.38 ± 3.92 | 68.00 ± 2.70 | 66.59 ± 4.43 | 69.02 ± 3.29 | 41.09 ± 2.22 | 46.20 ± 3.85 | 57.43 ± 6.38 | 63.85 ± 7.90 |
| G²GNN | remove edge | 81.34 ± 2.96 | 81.82 ± 2.90 | 68.09 ± 1.78 | 68.48 ± 1.69 | 63.72 ± 5.28 | 78.46 ± 4.55 | 60.82 ± 6.55 | 65.82 ± 3.88 | 48.12 ± 8.14 | 53.26 ± 5.12 | 54.61 ± 5.33 | 54.61 ± 5.33 |
| | mask node | 82.26 ± 3.60 | 82.78 ± 3.49 | 67.81 ± 1.80 | 68.26 ± 1.60 | 61.80 ± 3.63 | 76.03 ± 5.58 | 63.82 ± 3.22 | 66.97 ± 3.72 | 37.46 ± 9.54 | 46.68 ± 7.18 | 46.54 ± 9.82 | 53.92 ± 5.92 |
| TopoImb | dynamic sparsity | 65.30 ± 12.94 | 66.94 ± 10.97 | 52.04 ± 12.41 | 57.78 ± 8.59 | 14.60 ± 6.74 | 15.66 ± 6.46 | 28.23 ± 10.72 | 34.76 ± 8.84 | 50.69 ± 3.42 | 52.66 ± 3.85 | 48.78 ± 10.28 | 61.65 ± 6.19 |
| DataDec | | 76.62 ± 6.76 | 78.75 ± 5.73 | 57.24 ± 2.63 | 59.71 ± 1.67 | 45.38 ± 2.67 | 52.84 ± 3.43 | 59.78 ± 3.98 | 59.98 ± 4.06 | 49.58 ± 3.43 | 50.12 ± 2.98 | 65.33 ± 3.65 | 67.79 ± 3.23 |
| ImGKB | | 64.59 ± 6.72 | 68.58 ± 5.41 | 67.51 ± 1.64 | 67.64 ± 1.61 | 56.29 ± 3.51 | 69.31 ± 4.97 | 57.68 ± 1.58 | 58.98 ± 2.10 | 53.10 ± 3.29 | 53.51 ± 3.23 | 51.49 ± 4.48 | 54.42 ± 2.94 |
| ImbGNN | | 80.52 ± 4.82 | 81.14 ± 4.02 | 69.15 ± 2.40 | 69.71 ± 2.03 | 61.07 ± 4.31 | 74.94 ± 6.57 | 59.29 ± 2.72 | 61.61 ± 4.08 | 40.60 ± 8.34 | 50.12 ± 7.59 | 54.18 ± 8.35 | 57.96 ± 4.72 |
| **UniImb** | / | **87.29 ± 2.98** | **87.75 ± 2.97** | **76.33 ± 3.89** | **76.40 ± 3.87** | **68.77 ± 3.26** | **82.87 ± 4.59** | **72.70 ± 3.36** | **76.40 ± 2.99** | **53.29 ± 8.12** | **53.84 ± 7.99** | **69.84 ± 6.89** | **70.36 ± 6.98** |

| Model | Backbone | REDDIT-B Macro-F1 | REDDIT-B Micro-F1 | AIDS Macro-F1 | AIDS Micro-F1 | FRANKENSTEIN Macro-F1 | FRANKENSTEIN Micro-F1 | COLLAB Macro-F1 | COLLAB Micro-F1 | IMDB-MULTI Macro-F1 | IMDB-MULTI Micro-F1 | Synthie Macro-F1 | Synthie Micro-F1 |
|---|---|---|---|---|---|---|---|---|---|---|---|---|---|
| vanilla | GIN | 61.67 ± 9.80 | 62.05 ± 10.12 | 96.55 ± 0.41 | 98.84 ± 0.14 | 43.44 ± 9.56 | 49.55 ± 5.09 | 34.70 ± 7.28 | 61.43 ± 5.20 | 13.51 ± 0.45 | 15.52 ± 0.16 | 43.55 ± 3.63 | 45.12 ± 4.11 |
| | InfoGraph | 65.29 ± 3.44 | 66.08 ± 2.91 | 97.28 ± 0.37 | 99.01 ± 0.21 | 49.70 ± 8.20 | 57.28 ± 6.43 | 44.39 ± 3.18 | 64.20 ± 2.97 | 20.26 ± 4.89 | 36.78 ± 3.77 | 35.47 ± 3.90 | 45.12 ± 4.11 |
| | GraphCL | 64.06 ± 2.79 | 67.26 ± 4.01 | 96.74 ± 0.11 | 98.86 ± 0.18 | 52.22 ± 4.83 | 45.08 ± 4.01 | 45.08 ± 4.01 | 63.15 ± 4.33 | 24.47 ± 3.02 | 38.35 ± 4.71 | 34.29 ± 5.00 | 47.38 ± 5.76 |
| GTs | GraphGPS | 47.39 ± 25.39 | 51.25 ± 22.65 | 94.63 ± 12.16 | 94.70 ± 16.60 | 53.32 ± 5.13 | 55.73 ± 3.79 | 28.94 ± 4.10 | 45.65 ± 3.72 | 14.28 ± 5.37 | 28.61 ± 13.25 | 26.17 ± 2.15 | 39.79 ± 6.34 |
| | Exphormer | 56.74 ± 2.40 | 58.28 ± 4.38 | 96.52 ± 3.29 | 97.41 ± 5.03 | 46.22 ± 10.17 | 51.50 ± 5.91 | 32.20 ± 11.87 | 45.36 ± 6.65 | 15.27 ± 7.03 | 23.34 ± 9.40 | 20.44 ± 4.76 | 30.83 ± 4.70 |
| | Graph-Mamba | 49.65 ± 12.77 | 52.34 ± 6.21 | 98.24 ± 0.96 | 98.93 ± 0.84 | 50.29 ± 3.48 | 55.74 ± 4.06 | 38.93 ± 2.19 | 50.27 ± 5.41 | 20.83 ± 5.42 | 30.77 ± 9.54 | 24.53 ± 15.74 | 34.91 ± 13.05 |
| up-sampling | GIN | 70.59 ± 4.12 | 72.91 ± 4.80 | 98.65 ± 0.36 | 99.03 ± 0.12 | 67.56 ± 4.78 | 68.06 ± 5.23 | 66.26 ± 4.19 | 69.41 ± 2.58 | 30.38 ± 7.86 | 50.95 ± 6.27 | 40.38 ± 6.73 | 49.91 ± 4.97 |
| | InfoGraph | 69.67 ± 3.44 | 71.39 ± 4.28 | 99.01 ± 0.23 | 99.34 ± 0.35 | 64.87 ± 5.98 | 66.03 ± 4.29 | 63.20 ± 4.53 | 65.14 ± 3.78 | 28.36 ± 3.90 | 47.10 ± 4.21 | 39.69 ± 7.03 | 47.55 ± 5.25 |
| | GraphCL | 70.24 ± 6.25 | 71.48 ± 5.69 | 98.78 ± 0.28 | 99.29 ± 0.30 | 62.10 ± 8.29 | 65.90 ± 3.38 | 64.22 ± 3.82 | 66.00 ± 1.12 | 28.78 ± 4.15 | 48.20 ± 5.58 | 41.68 ± 4.55 | 50.78 ± 3.08 |
| re-weight | GIN | 71.83 ± 6.09 | 73.20 ± 6.22 | 97.22 ± 0.83 | 99.05 ± 0.30 | 61.35 ± 4.11 | 61.68 ± 3.44 | 56.18 ± 4.73 | 60.05 ± 2.10 | 16.25 ± 10.99 | 39.33 ± 9.21 | 34.67 ± 1.26 | 51.29 ± 5.72 |
| | InfoGraph | 72.29 ± 4.43 | 75.48 ± 3.33 | 98.88 ± 0.22 | 99.16 ± 0.46 | 64.38 ± 5.79 | 66.27 ± 5.01 | 58.06 ± 3.80 | 61.28 ± 4.02 | 18.26 ± 4.29 | 42.68 ± 2.10 | 39.52 ± 7.54 | 45.99 ± 5.02 |
| | GraphCL | 72.45 ± 5.01 | 76.38 ± 3.77 | 99.02 ± 1.70 | 99.38 ± 1.03 | 63.80 ± 13.92 | 67.11 ± 10.09 | 58.20 ± 4.10 | 60.48 ± 5.02 | 19.33 ± 4.58 | 45.04 ± 5.14 | 40.06 ± 6.79 | 52.46 ± 6.02 |
| G²GNN | remove edge | 65.34 ± 6.45 | 67.17 ± 4.80 | 98.24 ± 0.79 | 99.36 ± 0.29 | 51.08 ± 10.08 | 56.90 ± 4.78 | 36.92 ± 1.91 | 50.89 ± 2.48 | 30.45 ± 10.41 | 44.09 ± 5.58 | 40.59 ± 8.43 | 48.23 ± 3.93 |
| | mask node | 68.47 ± 3.91 | 71.22 ± 4.39 | 98.36 ± 0.78 | 99.42 ± 0.27 | 52.27 ± 11.13 | 58.43 ± 6.33 | 43.42 ± 3.28 | 54.41 ± 2.09 | 22.68 ± 8.15 | 37.90 ± 11.32 | 36.65 ± 4.78 | 48.83 ± 4.16 |
| TopoImb | dynamic sparsity | 22.90 ± 0.15 | 29.96 ± 0.21 | 52.16 ± 2.46 | 90.55 ± 0.31 | 34.80 ± 3.01 | 49.55 ± 6.30 | 20.54 ± 1.04 | 44.68 ± 1.30 | 8.97 ± 0.13 | 15.49 ± 0.14 | 23.03 ± 7.31 | 36.33 ± 8.61 |
| DataDec | | 80.62 ± 1.82 | 81.92 ± 1.77 | 92.38 ± 1.36 | 97.05 ± 0.79 | 51.32 ± 1.65 | 54.42 ± 1.07 | 47.84 ± 2.13 | 62.10 ± 0.74 | 13.44 ± 1.96 | 18.18 ± 1.23 | 34.43 ± 2.40 | 43.21 ± 1.17 |
| ImGKB | | 27.60 ± 7.84 | 40.00 ± 17.32 | 78.10 ± 5.22 | 90.35 ± 3.97 | 32.28 ± 3.05 | 47.83 ± 6.84 | 19.87 ± 2.31 | 42.68 ± 2.10 | 18.16 ± 4.77 | 36.06 ± 10.50 | 9.86 ± 1.92 | 24.85 ± 5.65 |
| ImbGNN | | 67.87 ± 3.01 | 69.99 ± 3.66 | 59.78 ± 29.83 | 74.37 ± 34.90 | 39.81 ± 9.92 | 53.35 ± 7.11 | 50.02 ± 3.44 | 58.60 ± 3.69 | 23.55 ± 9.50 | 40.62 ± 10.46 | 37.52 ± 7.92 | 52.94 ± 6.71 |
| **UniImb** | / | **90.90 ± 2.66** | **92.10 ± 2.17** | **99.83 ± 0.18** | **99.94 ± 0.07** | **71.12 ± 3.60** | **72.54 ± 3.20** | **75.50 ± 3.70** | **75.73 ± 3.71** | **35.55 ± 11.77** | **49.84 ± 12.20** | **42.10 ± 4.36** | **57.92 ± 0.57** |

## I.2 MULTI-CLASS DATASET

To further validate the performance of UniImb in multi-class settings, we incorporate additional datasets that cover diverse domains. Specifically, we include Enzymes (Schomburg et al., 2004) (a bioinformatics dataset with 6 classes) and Letter-high (Riesen & Bunke, 2008) and Letter-low

Table 16: Macro-F1 and Micro-F1 score on *class imbalance* datasets with *low imbalance degree*.

| Model | Backbone | MUTAG Macro-F1 | MUTAG Micro-F1 | PROTEINS Macro-F1 | PROTEINS Micro-F1 | D&D Macro-F1 | D&D Micro-F1 | NCI1 Macro-F1 | NCI1 Micro-F1 | PTC-MR Macro-F1 | PTC-MR Micro-F1 | DHFR Macro-F1 | DHFR Micro-F1 |
|---|---|---|---|---|---|---|---|---|---|---|---|---|---|
| vanilla | GIN | 72.43 ± 5.39 | 80.23 ± 5.80 | 68.99 ± 1.84 | 72.70 ± 2.85 | 68.62 ± 5.38 | 70.39 ± 6.12 | 72.99 ± 1.23 | 73.03 ± 1.34 | 49.16 ± 5.31 | 53.54 ± 6.78 | 41.45 ± 14.48 | 45.40 ± 11.63 |
|  | InfoGraph | 74.22 ± 4.67 | 81.24 ± 5.16 | 69.78 ± 1.14 | 70.09 ± 2.23 | 69.19 ± 5.02 | 72.43 ± 4.88 | 73.56 ± 4.92 | 73.99 ± 3.24 | 53.29 ± 6.10 | 54.22 ± 4.98 | 43.29 ± 10.78 | 48.15 ± 13.90 |
|  | GraphCL | 75.45 ± 5.13 | 82.00 ± 3.97 | 69.44 ± 3.34 | 71.34 ± 2.55 | 68.92 ± 4.14 | 71.57 ± 4.39 | 73.08 ± 5.49 | 73.48 ± 5.32 | 52.19 ± 5.44 | 54.02 ± 4.67 | 45.74 ± 8.99 | 50.16 ± 9.36 |
| GTs | GraphGPS | 61.95 ± 16.64 | 73.86 ± 19.85 | 67.74 ± 1.76 | 72.63 ± 2.38 | 65.82 ± 5.83 | 69.19 ± 7.46 | 70.94 ± 1.95 | 71.08 ± 1.91 | 47.10 ± 8.62 | 50.85 ± 8.68 | 65.83 ± 8.76 | 67.48 ± 9.11 |
|  | Exphormer | 64.49 ± 11.10 | 73.30 ± 13.01 | 62.62 ± 2.89 | 67.13 ± 3.57 | 63.39 ± 4.83 | 66.75 ± 6.35 | 67.73 ± 1.75 | 67.84 ± 1.67 | 50.89 ± 7.99 | 53.72 ± 6.89 | 65.91 ± 4.86 | 68.88 ± 3.17 |
|  | Graph-Mamba | 73.20 ± 5.09 | 82.39 ± 5.29 | 71.35 ± 1.77 | 75.05 ± 1.78 | 64.06 ± 4.84 | 67.91 ± 6.63 | 74.58 ± 1.60 | 74.64 ± 1.58 | 48.14 ± 7.54 | 53.72 ± 6.64 | 69.19 ± 3.92 | 71.16 ± 4.54 |
| G²GNN | remove edge | 72.57 ± 5.69 | 80.06 ± 5.77 | 68.29 ± 2.54 | 70.44 ± 3.21 | 72.83 ± 5.78 | 75.69 ± 4.32 | 59.56 ± 1.71 | 60.01 ± 1.55 | 47.92 ± 9.41 | 53.35 ± 7.74 | 47.90 ± 12.57 | 51.35 ± 11.56 |
|  | mask node | 76.02 ± 7.16 | 83.41 ± 6.48 | 67.41 ± 2.14 | 69.33 ± 2.52 | 72.96 ± 4.67 | 75.87 ± 5.28 | 74.58 ± 1.60 | 74.64 ± 1.58 | 39.01 ± 9.39 | 47.62 ± 9.37 | 49.78 ± 11.25 | 54.91 ± 10.56 |
| TopoImb | / | 49.41 ± 13.07 | 56.56 ± 15.89 | 22.89 ± 0.76 | 28.21 ± 3.32 | 31.94 ± 13.90 | 38.66 ± 10.94 | 39.52 ± 7.99 | 52.38 ± 2.75 | 48.12 ± 2.48 | 48.87 ± 2.38 | 62.99 ± 3.41 | 69.56 ± 4.83 |
| DataDec | dynamic sparsity | 73.55 ± 5.67 | 81.25 ± 5.14 | 62.99 ± 2.21 | 66.90 ± 3.07 | 60.75 ± 1.85 | 68.88 ± 1.21 | 73.20 ± 0.74 | 73.21 ± 0.76 | 53.78 ± 3.33 | 54.45 ± 3.38 | 66.28 ± 2.82 | 67.64 ± 3.82 |
| ImGKB | / | 61.49 ± 4.32 | 71.08 ± 5.02 | 66.51 ± 2.56 | 70.72 ± 2.61 | 67.43 ± 3.03 | 70.96 ± 3.17 | 61.88 ± 1.03 | 61.97 ± 1.03 | 51.82 ± 4.37 | 53.35 ± 4.67 | 50.32 ± 4.61 | 51.30 ± 4.93 |
| ImbGNN | / | 73.66 ± 5.64 | 82.61 ± 4.80 | 70.67 ± 3.17 | 72.40 ± 3.84 | 68.79 ± 3.85 | 70.74 ± 4.64 | 61.01 ± 1.25 | 61.50 ± 1.78 | 44.07 ± 7.50 | 54.70 ± 7.83 | 53.46 ± 12.41 | 57.07 ± 11.51 |
| **UniImb** | / | 77.28 ± 4.80 | 85.11 ± 4.30 | 73.54 ± 3.02 | 77.50 ± 2.87 | 76.87 ± 5.56 | 79.24 ± 4.63 | 76.49 ± 4.53 | 76.72 ± 4.36 | 59.88 ± 3.57 | 62.80 ± 3.05 | 70.62 ± 3.27 | 72.83 ± 3.47 |

| Model | Backbone | REDDIT-B Macro-F1 | REDDIT-B Micro-F1 | AIDS Macro-F1 | AIDS Micro-F1 | FRANKENSTEIN Macro-F1 | FRANKENSTEIN Micro-F1 | COLLAB Macro-F1 | COLLAB Micro-F1 | IMDB-MULTI Macro-F1 | IMDB-MULTI Micro-F1 | Synthie Macro-F1 | Synthie Micro-F1 |
|---|---|---|---|---|---|---|---|---|---|---|---|---|---|
| vanilla | GIN | 78.24 ± 4.94 | 78.51 ± 4.73 | 50.00 ± 0.00 | 100.00 ± 0.00 | 66.79 ± 1.72 | 66.98 ± 1.77 | 54.46 ± 2.10 | 65.12 ± 1.56 | 37.34 ± 3.63 | 45.44 ± 3.08 | 36.65 ± 6.67 | 50.49 ± 4.35 |
|  | InfoGraph | 82.44 ± 3.87 | 83.02 ± 4.28 | 50.00 ± 0.00 | 100.00 ± 0.00 | 67.18 ± 3.24 | 68.88 ± 4.13 | 56.14 ± 3.48 | 65.08 ± 2.01 | 38.29 ± 4.42 | 46.30 ± 2.99 | 38.14 ± 5.90 | 52.88 ± 4.43 |
|  | GraphCL | 81.47 ± 5.05 | 82.59 ± 4.46 | 50.00 ± 0.00 | 100.00 ± 0.00 | 66.92 ± 5.12 | 69.05 ± 4.38 | 55.09 ± 4.04 | 65.19 ± 4.01 | 40.47 ± 4.44 | 47.18 ± 5.15 | 40.47 ± 4.78 | 51.65 ± 5.34 |
| GTs | GraphGPS | 45.13 ± 13.70 | 54.62 ± 7.49 | 28.72 ± 7.75 | 42.15 ± 17.27 | 61.93 ± 13.28 | 65.16 ± 8.72 | 13.74 ± 11.23 | 27.91 ± 24.31 | 16.67 ± 2.77 | 33.33 ± 2.55 | 24.96 ± 2.93 | 29.38 ± 2.99 |
|  | Exphormer | 72.48 ± 12.53 | 73.37 ± 11.52 | 33.74 ± 10.73 | 50.10 ± 16.22 | 67.34 ± 1.08 | 67.93 ± 1.25 | 21.50 ± 9.18 | 34.75 ± 13.71 | 20.35 ± 3.36 | 33.10 ± 1.14 | 18.99 ± 4.68 | 29.38 ± 2.99 |
|  | Graph-Mamba | 67.34 ± 11.24 | 69.27 ± 8.44 | 49.93 ± 0.03 | 99.74 ± 0.10 | 65.74 ± 6.81 | 69.59 ± 6.32 | 38.75 ± 16.79 | 52.35 ± 16.27 | 20.32 ± 4.35 | 34.02 ± 0.99 | 38.87 ± 13.04 | 47.65 ± 10.01 |
| G²GNN | remove edge | 70.33 ± 3.21 | 70.35 ± 5.47 | 49.95 ± 0.03 | 99.82 ± 0.10 | 55.60 ± 7.54 | 58.70 ± 6.11 | 20.63 ± 4.53 | 37.15 ± 3.83 | 31.83 ± 8.29 | 37.93 ± 3.28 | 41.46 ± 5.53 | 47.91 ± 4.41 |
|  | mask node | 68.22 ± 4.67 | 68.43 ± 5.10 | 49.93 ± 0.02 | 98.82 ± 0.10 | 58.10 ± 5.75 | 60.94 ± 3.49 | 25.82 ± 4.46 | 40.83 ± 3.14 | 28.55 ± 10.90 | 37.36 ± 2.81 | 35.53 ± 6.97 | 47.28 ± 4.36 |
| TopoImb | / | 33.21 ± 0.11 | 49.98 ± 0.20 | 40.67 ± 1.84 | 88.78 ± 5.26 | 34.49 ± 10.07 | 43.61 ± 6.56 | 13.78 ± 5.02 | 26.96 ± 10.48 | 36.90 ± 7.61 | 42.88 ± 3.94 | 14.02 ± 3.73 | 25.49 ± 6.39 |
| DataDec | dynamic sparsity | 86.64 ± 1.70 | 86.66 ± 1.71 | 48.92 ± 0.29 | 95.76 ± 1.15 | 68.36 ± 0.71 | 68.76 ± 0.72 | 59.05 ± 0.71 | 64.89 ± 0.78 | 46.29 ± 1.47 | 47.89 ± 1.47 | 43.48 ± 5.32 | 45.19 ± 4.27 |
| ImGKB | / | 34.25 ± 0.36 | 50.41 ± 0.17 | 46.72 ± 2.83 | 88.15 ± 8.96 | 33.56 ± 4.38 | 50.92 ± 9.84 | 14.04 ± 8.43 | 28.35 ± 2.87 | 17.60 ± 0.59 | 33.78 ± 0.28 | 10.06 ± 0.99 | 25.27 ± 3.12 |
| ImbGNN | / | 71.42 ± 1.13 | 71.46 ± 1.11 | 40.25 ± 16.03 | 76.15 ± 33.05 | 42.72 ± 8.64 | 53.15 ± 8.25 | 36.36 ± 3.16 | 49.25 ± 2.60 | 26.76 ± 11.52 | 35.93 ± 3.23 | 33.84 ± 10.82 | 48.73 ± 7.87 |
| **UniImb** | / | 91.18 ± 4.07 | 91.19 ± 4.05 | 50.00 ± 0.00 | 100.00 ± 0.00 | 68.61 ± 4.19 | 69.23 ± 4.31 | 63.67 ± 5.08 | 69.46 ± 4.23 | 45.07 ± 9.83 | 46.97 ± 5.72 | 37.59 ± 2.86 | 54.23 ± 4.49 |

(Riesen & Bunke, 2008) (two visual-domain datasets, each containing 15 classes). These datasets introduce greater label granularity and structural diversity, enabling a more comprehensive evaluation of UniImb's generalization ability in complex multi-class graph imbalance learning scenarios [2]. As shown in Table 17, experimental results indicate that ImbGNN exhibits inferior performance, achieving performance close to GIN, whereas our model UniImb maintains competitive performance on multi-label datasets.

Table 17: Macro-F1 and Micro-F1 score on multi-class datasets with *extreme imbalance degree*.

| Model | ENZYMES Macro-F1 | ENZYMES Micro-F1 | Letter-high Macro-F1 | Letter-high Micro-F1 | Letter-low Macro-F1 | Letter-low Micro-F1 |
|---|---|---|---|---|---|---|
| GIN | 10.15 ± 4.36 | 12.28 ± 2.04 | 25.62 ± 2.77 | 27.84 ± 1.70 | 23.08 ± 3.83 | 40.85 ± 3.13 |
| G²GNN | 17.94 ± 1.35 | 19.33 ± 4.16 | 29.40 ± 5.39 | 33.67 ± 4.92 | 31.28 ± 2.47 | 47.92 ± 4.94 |
| TopoImb | 7.26 ± 4.76 | 8.94 ± 5.35 | 18.27 ± 9.26 | 19.41 ± 11.31 | 21.94 ± 6.28 | 39.22 ± 7.36 |
| ImbGNN | 14.76 ± 2.06 | 17.19 ± 3.10 | 31.96 ± 3.47 | 35.84 ± 2.92 | 27.29 ± 6.54 | 45.27 ± 5.43 |
| **UniImb** | **41.38** ± 2.37 | **56.86** ± 3.14 | **39.62** ± 1.38 | **46.94** ± 2.16 | **43.86** ± 2.48 | **64.97** ± 3.59 |

## I.3 Topological imbalance Performance Analysis

### I.3.1 Complete Experiments on All Datasets Various Imbalance Degrees

In Table 18, we report the performance gains of our model over other baselines on both large-sacle and small-scale graphs under extreme topological imbalance. And the performance across the 8 datasets is shown in Table 19, Table 20, and Table 23. SOLT-GNN identifies co-occurring topologies in larger graphs (or "head" graphs) and transfers this information to smaller graphs, thereby improving their performance. It achieves strong results, demonstrating the effectiveness of knowledge transfer mechanisms in alleviating imbalanced classification. ImbGNN designs a topology-aware $G^2G$ construction method that is friendly to imbalanced structures, showcasing its potential in handling topology-imbalance. Although TopoImb was primarily proposed to address uneven substructure distributions, our results also show that it can effectively mitigate topology-imbalance across multiple datasets. **Our model achieves competitive performance regardless of whether the graphs are large-scale or small-scale**, which can be attributed to two main factors: first, our topological local encoding and graph perturbation strategies enhance the model's robustness to data variations; second, the proposed prototype-based strategy improves the representation learning of small-scale graphs.

---

[2]https://chrsmrrs.github.io/datasets/docs/datasets/

Table 18: Comparison of Micro-F1 performance *topological imbalanced* datasets with *extreme imbalance degree* across graphs of *different scales*.

| Model | PROTEINS | | NCI1 | | IMDB-MULTI | |
|---|---|---|---|---|---|---|
| | Small-scale | Large-scale | Small-scale | Large-scale | Small-scale | Large-scale |
| GIN | 56.12 ± 3.18 | 64.28 ± 3.52 | 58.39 ± 3.12 | 70.88 ± 6.01 | 32.19 ± 4.41 | 33.98 ± 5.06 |
| InfoGraph | 58.42 ± 2.67 | 65.10 ± 3.19 | 60.76 ± 3.29 | 70.64 ± 5.17 | 34.22 ± 4.61 | 37.29 ± 2.13 |
| GraphCL | 59.24 ± 2.28 | 65.88 ± 3.41 | 61.33 ± 4.67 | 74.75 ± 5.02 | 31.24 ± 5.94 | 35.98 ± 5.19 |
| GraphGPS | 66.02 ± 2.85 | 80.40 ± 6.46 | 60.81 ± 4.77 | 68.18 ± 7.61 | 34.14 ± 1.10 | 31.38 ± 3.01 |
| Exphormer | 65.42 ± 2.84 | 73.91 ± 9.07 | 61.96 ± 12.87 | 68.83 ± 2.48 | 33.59 ± 3.45 | 33.63 ± 3.79 |
| Graph-Mamba | 66.12 ± 3.61 | 85.25 ± 3.52 | 62.49 ± 3.60 | 68.60 ± 6.11 | 30.45 ± 3.19 | 34.07 ± 1.08 |
| TopoImb | 53.36 ± 6.68 | 62.87 ± 36.75 | 61.72 ± 1.24 | 72.92 ± 1.57 | 38.40 ± 2.56 | 47.62 ± 2.74 |
| ImbGNN | 68.30 ± 0.88 | 63.23 ± 3.07 | 58.68 ± 1.86 | 70.89 ± 1.56 | 32.76 ± 6.13 | 34.07 ± 7.33 |
| SOLT-GNN | 67.34 ± 3.57 | 83.88 ± 3.64 | 58.52 ± 1.91 | 68.77 ± 7.96 | 36.64 ± 1.39 | 42.24 ± 2.76 |
| **UniImb** | **69.68** ± 2.15 | **87.23** ± 2.04 | **64.68** ± 1.44 | **76.19** ± 2.36 | **42.48** ± 1.81 | **48.23** ± 2.72 |

Table 19: Macro-F1, Micro-F1 on *topological imbalance* datasets with *extreme imbalance degree*.

| Model | PROTEINS | | D&D | | NCI1 | | PTC-MR | |
|---|---|---|---|---|---|---|---|---|
| | Macro-F1 | Micro-F1 | Macro-F1 | Micro-F1 | Macro-F1 | Micro-F1 | Macro-F1 | Accuracy |
| GIN | 53.48 ± 2.03 | 58.00 ± 4.19 | 57.98 ± 5.51 | 60.68 ± 6.89 | 61.60 ± 2.20 | 61.84 ± 2.29 | 34.56 ± 6.32 | 48.41 ± 7.07 |
| InfoGraph | 55.31 ± 3.30 | 60.68 ± 3.52 | 59.46 ± 4.03 | 61.22 ± 4.11 | 62.25 ± 1.53 | 62.69 ± 2.01 | 44.68 ± 8.14 | 48.33 ± 4.91 |
| GraphCL | 57.62 ± 5.50 | 61.91 ± 3.84 | 62.01 ± 5.18 | 63.29 ± 2.33 | 63.62 ± 5.43 | 64.01 ± 4.75 | 41.82 ± 6.67 | 49.75 ± 5.25 |
| GraphGPS | 65.54 ± 4.22 | 69.26 ± 2.48 | 63.33 ± 12.97 | 63.85 ± 2.08 | 62.96 ± 3.51 | 64.02 ± 2.40 | 48.45 ± 9.71 | 50.46 ± 6.45 |
| Exphormer | 64.01 ± 2.18 | 67.33 ± 2.78 | 59.43 ± 8.06 | 60.26 ± 5.92 | 62.16 ± 3.19 | 63.23 ± 7.31 | 49.14 ± 9.56 | 51.15 ± 3.86 |
| Graph-Mamba | 68.46 ± 3.91 | 72.09 ± 3.16 | 43.86 ± 9.73 | 54.45 ± 9.53 | 63.09 ± 2.82 | 63.63 ± 2.14 | 49.72 ± 12.58 | 51.01 ± 5.66 |
| TopoImb | 44.79 ± 14.19 | 54.89 ± 13.58 | 63.97 ± 2.78 | 64.16 ± 2.96 | 63.57 ± 1.69 | 64.17 ± 1.33 | 51.65 ± 1.07 | 51.96 ± 1.16 |
| ImbGNN | 66.87 ± 2.59 | 67.07 ± 2.75 | 68.67 ± 6.33 | 70.71 ± 6.99 | 61.24 ± 1.72 | 61.66 ± 1.17 | 43.87 ± 4.75 | 52.66 ± 4.53 |
| SOLT-GNN | 70.70 ± 2.20 | 72.14 ± 2.18 | 58.50 ± 10.48 | 64.97 ± 3.24 | 61.99 ± 1.44 | 62.13 ± 1.44 | 40.70 ± 3.27 | 51.74 ± 5.25 |
| **UniImb** | **71.32** ± 1.88 | **74.89** ± 1.12 | **74.49** ± 1.13 | **76.73** ± 1.04 | **64.99** ± 9.58 | **65.76** ± 7.24 | **53.86** ± 2.59 | **56.38** ± 1.88 |

Table 20: Macro-F1, Micro-F1 on *topological imbalance* datasets with *medium imbalance degree*.

| Model | PROTEINS | | D&D | | NCI1 | | PTC-MR | |
|---|---|---|---|---|---|---|---|---|
| | Macro-F1 | Micro-F1 | Macro-F1 | Micro-F1 | Macro-F1 | Micro-F1 | Macro-F1 | Micro-F1 |
| GIN | 56.46 ± 2.90 | 62.14 ± 2.43 | 60.89 ± 1.84 | 61.46 ± 2.43 | 58.37 ± 6.27 | 60.47 ± 3.60 | 38.58 ± 5.97 | 51.38 ± 6.78 |
| InfoGraph | 58.87 ± 3.41 | 63.07 ± 2.88 | 61.78 ± 4.92 | 62.37 ± 3.26 | 63.21 ± 5.33 | 64.29 ± 3.94 | 45.28 ± 9.13 | 48.76 ± 5.28 |
| GraphCL | 61.48 ± 2.19 | 64.43 ± 2.10 | 63.41 ± 4.03 | 63.78 ± 3.75 | 64.18 ± 5.71 | 66.02 ± 2.13 | 43.46 ± 4.11 | 46.92 ± 2.53 |
| GraphGPS | 59.74 ± 4.68 | 66.99 ± 1.63 | 58.72 ± 2.23 | 59.25 ± 1.05 | 56.11 ± 3.67 | 59.12 ± 4.36 | 40.62 ± 3.41 | 47.59 ± 2.26 |
| Exphormer | 60.32 ± 4.66 | 63.19 ± 3.61 | 60.74 ± 2.13 | 61.04 ± 2.70 | 58.93 ± 1.28 | 61.49 ± 3.02 | 39.25 ± 3.42 | 49.16 ± 2.72 |
| Graph-Mamba | 62.41 ± 3.28 | 65.66 ± 2.03 | 62.85 ± 4.29 | 62.94 ± 1.70 | 57.44 ± 3.52 | 59.83 ± 3.16 | 42.56 ± 3.92 | 50.78 ± 3.02 |
| TopoImb | 59.03 ± 4.35 | 64.03 ± 4.43 | 64.65 ± 1.76 | 65.99 ± 1.25 | 66.94 ± 1.79 | 67.16 ± 1.74 | 49.17 ± 0.95 | 51.59 ± 4.30 |
| ImbGNN | 69.91 ± 1.61 | 70.70 ± 1.99 | 53.71 ± 17.20 | 58.42 ± 17.49 | 58.26 ± 4.48 | 59.83 ± 2.47 | 49.64 ± 6.20 | 52.92 ± 4.40 |
| SOLT-GNN | 70.58 ± 2.03 | 71.95 ± 2.36 | 58.67 ± 4.91 | 63.33 ± 1.86 | 62.76 ± 1.29 | 62.81 ± 1.28 | 43.20 ± 0.36 | 53.04 ± 3.91 |
| **UniImb** | **70.62** ± 1.60 | **74.20** ± 0.94 | **72.70** ± 3.03 | **74.56** ± 3.18 | **69.53** ± 3.50 | **70.07** ± 2.84 | **52.36** ± 2.62 | **53.99** ± 3.07 |

Table 21: Macro-F1, Micro-F1 on *topological imbalance* datasets with *low imbalance degree*.

| Model | PROTEINS | | D&D | | NCI1 | | PTC-MR | |
|---|---|---|---|---|---|---|---|---|
| | Macro-F1 | Micro-F1 | Macro-F1 | Micro-F1 | Macro-F1 | Micro-F1 | Macro-F1 | Micro-F1 |
| GIN | 59.67 ± 1.87 | 63.58 ± 1.76 | 63.39 ± 0.84 | 65.01 ± 0.69 | 61.89 ± 5.08 | 63.30 ± 3.19 | 48.26 ± 3.11 | 52.17 ± 4.36 |
| InfoGraph | 61.18 ± 2.22 | 64.27 ± 2.58 | 62.56 ± 4.09 | 63.83 ± 2.29 | 3.44 ± 4.42 | 64.69 ± 4.10 | 47.19 ± 3.73 | 51.07 ± 2.84 |
| GraphCL | 63.72 ± 3.03 | 65.08 ± 4.29 | 64.54 ± 3.08 | 64.28 ± 4.67 | 65.08 ± 3.27 | 66.82 ± 43.19 | 48.57 ± 4.23 | 52.90 ± 4.65 |
| GraphGPS | 65.95 ± 2.46 | 65.96 ± 3.23 | 64.69 ± 1.33 | 65.94 ± 3.91 | 63.20 ± 0.94 | 63.46 ± 0.81 | 49.81 ± 2.95 | 50.07 ± 2.88 |
| Exphormer | 51.77 ± 6.39 | 54.73 ± 5.82 | 38.95 ± 1.13 | 41.48 ± 1.25 | 49.17 ± 8.89 | 54.77 ± 3.48 | 44.99 ± 3.29 | 45.97 ± 2.60 |
| Graph-Mamba | 60.16 ± 2.54 | 60.88 ± 2.20 | 59.48 ± 3.35 | 61.42 ± 3.79 | 64.49 ± 1.92 | 64.70 ± 1.70 | 47.85 ± 3.22 | 48.19 ± 2.91 |
| TopoImb | 59.94 ± 4.32 | 61.19 ± 4.61 | 64.97 ± 1.65 | 65.16 ± 1.76 | 62.41 ± 0.66 | 62.46 ± 0.66 | 48.83 ± 2.90 | 49.71 ± 1.98 |
| ImbGNN | 62.41 ± 11.37 | 64.33 ± 4.65 | 50.74 ± 7.30 | 58.04 ± 2.77 | 62.01 ± 0.94 | 62.09 ± 0.99 | 48.72 ± 4.22 | 50.33 ± 3.26 |
| SOLT-GNN | 64.60 ± 4.66 | 65.56 ± 4.83 | 62.34 ± 0.66 | 63.78 ± 1.06 | 58.75 ± 0.73 | 59.59 ± 0.77 | 43.63 ± 1.61 | 47.54 ± 4.33 |
| **UniImb** | **65.70** ± 3.65 | **65.80** ± 3.51 | **65.13** ± 3.39 | **65.25** ± 3.49 | 53.50 ± 8.30 | 59.88 ± 4.12 | **50.85** ± 6.46 | **53.32** ± 5.28 |

## I.4 ABLATION EXPERIMENT WITH MICRO-F1 SCORE.

In the main text, we report the Macro-F1 score from the ablation studies. Here, we supplement the results with the Micro-F1 score for completeness. As shown in Figure 8, the findings remain consistent: each component contributes positively to the overall performance, confirming the effectiveness of our design choices.

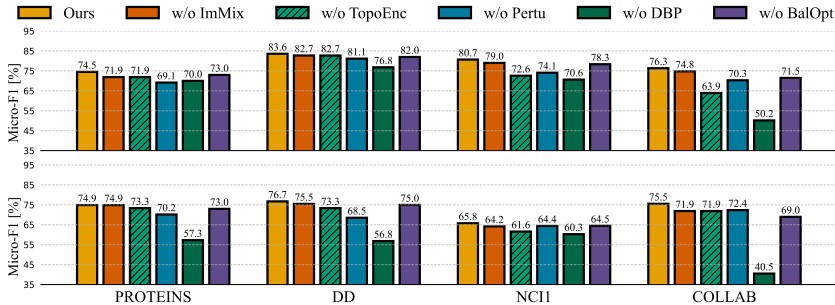

Figure 8: Ablation experiments of on class-imbalance (upper half) and topological-imbalance (lower half), evaluated by Micro-F1 score.

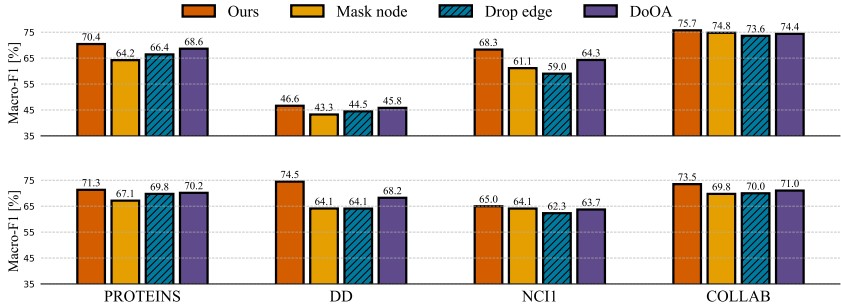

Figure 9: Ablation experiments on class-imbalance (upper half) and topological-imbalance (lower half), evaluated by Macro-F1 score.

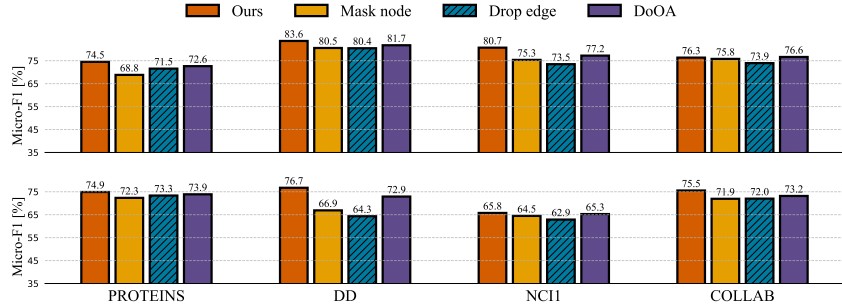

Figure 10: Ablation experiments on class-imbalance (upper half) and topological-imbalance (lower half), evaluated by Micro-F1 score.

## I.5 ADDITIONAL EXPERIMENT OF GRAPH PERTURBATION STRATEGY

We further evaluate the effectiveness of the two graph augmentation strategies. The Mask Node variant denotes our approach with randomly dropped nodes, while the Drop Edge variant refers to the method that randomly masks edges. In both cases, the perturbation operations are applied in a non-learnable and static manner. Degree-oriented Optional Augmentation (DoOA) (Xu et al., 2024) improves upon these baselines by customizing the node masking and edge dropping probabilities for each graph based on its structural properties—specifically, the node degree distribution. As shown

in Figures 9 and 10, all variants underperform compared to our full model to varying degrees. This demonstrates the effectiveness of our proposed learnable node and edge perturbation strategies, which adaptively learn data-driven augmentation policies from the input graphs. These mechanisms facilitate the learning of richer topological representations and contribute to improved overall performance.

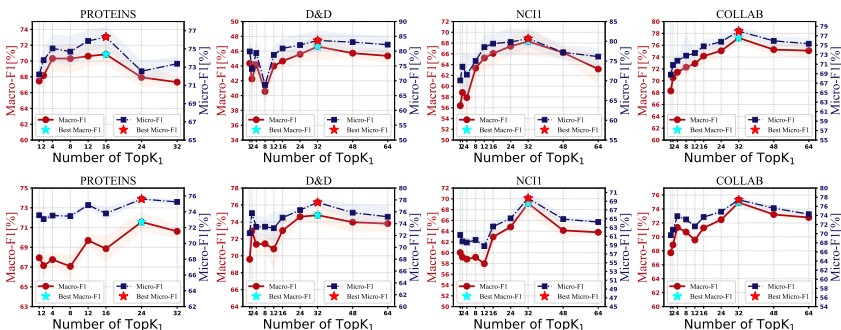

Figure 11: Sensitivity study of $TopK_1$ on class-imbalance (upper half) and topological-imbalance (lower half).

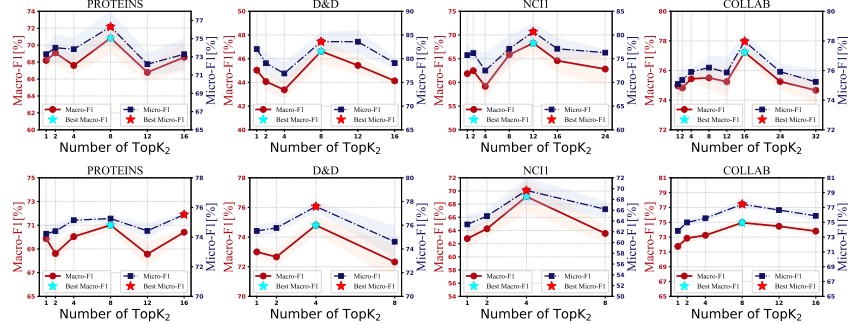

Figure 12: Sensitivity study of $TopK_2$ on class-imbalance (upper half) and topological-imbalance (lower half).

## I.6 HYPERPARAMETER EXPERIMENT

We further evaluate the sensitivity of two key hyperparameters, $TopK_1$ and $TopK_2$, which are involved in the purification operations during the attention computation. Specifically: $TopK_1$ denotes the number of most-matching graphs selected for each prototype during the prototype-awareness stage. $TopK_2$ represents the number of most-affine prototypes selected for each graph during the prototype-balance stage. In this analysis, we fix K to its optimal value for each dataset and report the performance under different settings of $TopK_1$ and $TopK_2$. As shown in Figure 11 and Figure 12, we observe that $TopK_1$ typically performs well within the range of 16 to 64. Values of $TopK_1$ that are too large may lead to the inclusion of redundant prototype features, which can degrade model performance. On the other hand, $TopK_2$ generally achieves strong results when set between 4 and 12. Furthermore, we conducted hyperparameter experiments on the D&D dataset under an extreme class-imbalance scenario, exploring different combinations of $TopK_1$ and $TopK_2$. The model achieves the best performance when $TopK_1$ is set to 32 and $TopK_2$ to 8, as illustrated in Figures 13 and 14. These results indicate that carefully selecting these hyperparameters is essential for capturing representative patterns and achieving balanced learning.

## I.7 APPLICABILITY ANALYSIS OF UNIIMB

We further evaluate the impact of different GNN architectures and Graph Transformers (GTs) as backbones on model performance. Specifically, we experiment with GCN (Kipf & Welling, 2016), GraphSAGE (Hamilton et al., 2017), GraphGPS (Rampášek et al., 2022), Exphormer (Shirzad et al., 2023), and Graph-Mamba (Wang et al., 2024b) as alternative graph learning backbones—in place of the GIN architecture used in our original design. The results are shown in Table 22 and Table

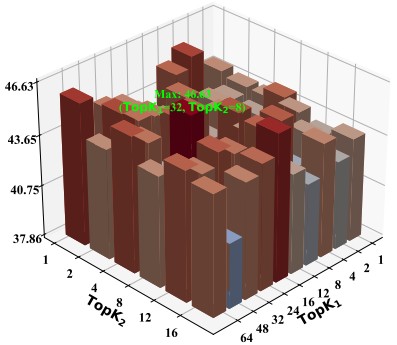

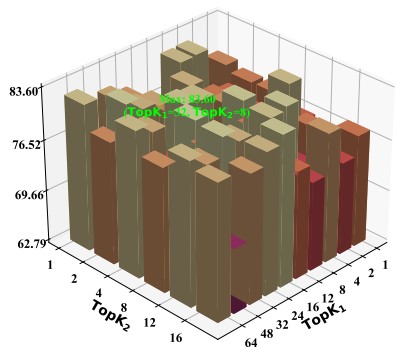

Figure 13: Macro-F1 on the D&D Dataset for $TopK_1$ and $TopK_2$ Hyperparameter Combinations.

Figure 14: Micro-F1 on the D&D Dataset for $TopK_1$ and $TopK_2$ Hyperparameter Combinations.

23, We find that both GraphSAGE and Graph-Mamba achieve superior performance: GraphSAGE benefits from its efficient neighborhood aggregation that preserves informative local structures, while Graph-Mamba excels at jointly modeling node features and long-range structural dependencies. Importantly, when integrated with our framework, all backbone variants demonstrate significantly improved performance on imbalanced graph classification tasks. This result validates the effectiveness and generalizability of our method across various backbone architectures.

Table 22: Graph classification performance of various backbones wiht our UniImb under **Class Imbalance** datasets with **extreme imbalance degree**.

| Method | PROTEINS | | D&D | | NCI1 | | REDDIT-B | | COLLAB | | IMDB-MULTI | |
|---|---|---|---|---|---|---|---|---|---|---|---|---|
| | Macro-F1 | Micro-F1 | Macro-F1 | Micro-F1 | Macro-F1 | Micro-F1 | Macro-F1 | Micro-F1 | Macro-F1 | Micro-F1 | Macro-F1 | Micro-F1 |
| GIN | 25.33 ± 7.53 | 28.50 ± 5.82 | 9.99 ± 7.44 | 11.88 ± 9.49 | 18.24 ± 7.58 | 18.94 ± 7.12 | 33.19 ± 14.26 | 36.02 ± 17.38 | 32.58 ± 3.66 | 57.31 ± 4.12 | 13.25 ± 6.19 | 14.92 ± 5.43 |
| GIN + UniImb | 70.44 ± 4.72 | 74.50 ± 4.99 | 46.63 ± 3.42 | 83.60 ± 6.50 | 68.30 ± 5.19 | 80.68 ± 4.22 | 76.24 ± 4.09 | 88.82 ± 2.93 | 75.73 ± 2.52 | 76.34 ± 2.60 | 33.45 ± 7.83 | 45.72 ± 4.87 |
| GCN | 21.57 ± 4.47 | 25.61 ± 3.27 | 7.20 ± 2.29 | 7.77 ± 2.53 | 10.86 ± 10.23 | 12.18 ± 13.87 | 29.19 ± 1.38 | 31.00 ± 2.52 | 30.16± 2.48 | 55.23 ± 4.33 | 10.75 ± 4.10 | 12.82 ± 5.46 |
| GCN + UniImb | 66.51 ± 2.69 | 70.05 ± 2.46 | 44.85 ± 4.15 | 81.63 ± 3.02 | 62.84 ± 5.13 | 79.23 ± 2.48 | 74.22 ± 4.39 | 86.94 ± 2.51 | 72.18 ± 4.33 | 74.39 ± 5.06 | 30.52 ± 3.19 | 43.93 ± 2.28 |
| GraphSage | 25.83 ± 5.53 | 28.73 ± 4.18 | 6.68 ± 3.01 | 7.15 ± 2.63 | 10.65 ± 1.44 | 11.94 ± 2.01 | 27.23 ± 2.21 | 29.62 ± 1.99 | 31.64 ± 3.06 | 56.77 ± 2.18 | 12.26 ± 2.72 | 14.01 ± 2.32 |
| GraphSage + UniImb | 67.83 ± 3.17 | 71.96 ± 4.29 | 42.10 ± 5.54 | 79.38 ± 4.62 | 61.51 ± 3.08 | 77.44 ± 4.17 | 67.35 ± 2.92 | 83.02 ± 4.53 | 74.10 ± 3.51 | 75.26 ± 4.43 | 32.29 ± 3.28 | 45.70 ± 4.17 |
| GraphGPS | 25.79 ± 7.05 | 28.71 ± 5.46 | 10.12 ± 4.41 | 11.97 ± 3.91 | 14.94 ± 2.41 | 15.62 ± 2.07 | 11.68 ± 7.76 | 12.71 ± 8.13 | 25.58 ± 12.93 | 39.92 ± 13.06 | 14.20 ± 5.61 | 28.54 ± 13.78 |
| GraphGPS + UniImb | 64.34 ± 3.28 | 68.75 ± 2.37 | 40.17 ± 2.19 | 68.27 ± 4.36 | 65.26 ± 3.98 | 78.46 ± 3.79 | 66.07 ± 4.36 | 79.79 ± 2.66 | 69.26 ± 3.29 | 71.53 ± 4.92 | 32.77 ± 2.98 | 43.07 ± 1.22 |
| Exphormer | 25.52 ± 4.79 | 28.38 ± 3.57 | 9.79 ± 4.18 | 10.85 ± 4.28 | 14.56 ± 3.92 | 15.36 ± 3.60 | 22.68 ± 10.79 | 27.33 ± 21.65 | 32.61 ± 17.44 | 42.02 ± 15.44 | 20.81 ± 5.43 | 28.14 ± 8.30 |
| Exphormer + UniImb | 65.95 ± 5.43 | 70.86 ± 4.17 | 45.48 ± 0.94 | 83.43 ± 3.28 | 64.21 ± 2.01 | 77.51 ± 3.63 | 68.19 ± 1.38 | 79.59 ± 2.09 | 74.65 ± 2.17 | 85.22 ± 4.41 | 31.33 ± 2.17 | 43.95 ± 3.15 |
| Graph-Mamba | 31.12 ± 5.10 | 32.79 ± 4.02 | 4.99 ± 7.93 | 6.12 ± 10.36 | 14.11 ± 3.26 | 14.94 ± 3.82 | 15.27 ± 12.46 | 17.02 ± 14.71 | 42.53 ± 11.15 | 50.63 ± 5.38 | 16.89 ± 4.57 | 28.69 ± 12.47 |
| Graph-Mamba + UniImb | 68.62 ± 3.61 | 73.88 ± 3.19 | 42.18 ± 2.11 | 72.96 ± 3.65 | 66.71 ± 3.84 | 78.60 ± 3.02 | 72.13 ± 1.41 | 82.58 ± 4.97 | 77.93 ± 2.16 | 78.57 ± 3.18 | 29.48 ± 5.64 | 41.02 ± 4.29 |

Table 23: Graph classification performance of various backbones wiht our UniImb under **Topological Imbalance** datasets with **extreme imbalance degree**.

| Method | PROTEINS | | D&D | | NCI1 | | REDDIT-B | | COLLAB | | IMDB-MULTI | |
|---|---|---|---|---|---|---|---|---|---|---|---|---|
| | Macro-F1 | Micro-F1 | Macro-F1 | Micro-F1 | Macro-F1 | Micro-F1 | Macro-F1 | Micro-F1 | Macro-F1 | Micro-F1 | Macro-F1 | Micro-F1 |
| GIN | 53.48 ± 2.03 | 58.00 ± 4.19 | 57.98 ± 5.51 | 60.68 ± 6.89 | 61.60 ± 2.20 | 61.84 ± 2.29 | 66.60 ± 2.27 | 67.41 ± 2.23 | 64.92 ± 2.18 | 67.05 ± 4.26 | 20.80 ± 4.91 | 34.73 ± 2.16 |
| GIN + UniImb | 71.32 ± 1.88 | 74.89 ± 1.12 | 74.49 ± 1.13 | 76.73 ± 1.04 | 64.99 ± 9.58 | 65.76 ± 7.24 | 77.14 ± 10.05 | 78.22 ± 7.41 | 73.51 ± 1.48 | 75.54 ± 1.63 | 40.45 ± 3.91 | 46.67 ± 2.31 |
| GCN | 51.29 ± 1.10 | 58.02 ± 5.02 | 52.46 ± 8.79 | 60.98 ± 6.71 | 55.12 ± 2.34 | 59.04 ± 1.13 | 65.91 ± 0.31 | 66.38 ± 0.46 | 66.80 ± 1.08 | 68.99 ± 1.36 | 18.95 ± 2.94 | 32.29 ± 1.32 |
| GCN + UniImb | 66.17 ± 10.15 | 72.90 ± 8.37 | 68.26 ± 1.96 | 70.01 ± 1.59 | 61.92 ± 2.20 | 62.27 ± 1.76 | 71.30 ± 1.68 | 71.75 ± 1.31 | 69.55 ± 1.23 | 72.21 ± 1.45 | 38.95 ± 8.54 | 44.67 ± 4.61 |
| GraphSage | 52.38 ± 0.67 | 60.06 ± 6.17 | 54.57 ± 1.01 | 62.67 ± 4.09 | 57.46 ± 5.49 | 62.10 ± 2.88 | 66.57 ± 3.17 | 68.71 ± 3.28 | 67.19 ± 3.11 | 70.10 ± 2.62 | 23.06 ± 6.41 | 33.78 ± 4.58 |
| GraphSage + UniImb | 69.53 ± 2.60 | 79.65 ± 1.67 | 70.91 ± 1.46 | 72.04 ± 1.48 | 62.36 ± 5.66 | 62.76 ± 5.65 | 73.29 ± 4.13 | 73.75 ± 2.33 | 71.28 ± 0.91 | 73.34 ± 2.41 | 39.13 ± 1.48 | 45.19 ± 1.20 |
| GraphGPS | 65.54 ± 4.22 | 69.26 ± 2.48 | 63.33 ± 12.97 | 63.85 ± 2.08 | 62.96 ± 3.51 | 64.02 ± 2.40 | 66.16 ± 4.19 | 68.42 ± 5.32 | 24.11 ± 8.74 | 56.65 ± 2.81 | 16.87 ± 0.53 | 34.49 ± 0.62 |
| GraphGPS + UniImb | 68.19 ± 3.74 | 77.30 ± 3.52 | 71.76 ± 2.25 | 74.26 ± 3.16 | 63.87 ± 2.03 | 64.25 ± 4.18 | 67.52 ± 1.84 | 69.88 ± 2.95 | 57.41 ± 3.28 | 66.73 ± 3.06 | 35.26 ± 4.08 | 42.64 ± 1.28 |
| Exphormer | 64.01 ± 2.18 | 67.33 ± 2.78 | 59.43 ± 8.06 | 60.26 ± 5.92 | 62.16 ± 3.19 | 63.23 ± 7.31 | 66.48 ± 14.59 | 67.81 ± 7.45 | 21.03 ± 7.89 | 36.71 ± 15.27 | 25.52 ± 5.03 | 33.60 ± 2.70 |
| Exphormer + UniImb | 67.26 ± 4.18 | 73.92 ± 3.60 | 68.86 ± 2.88 | 69.81 ± 1.86 | 65.92 ± 4.43 | 66.71 ± 3.68 | 69.62 ± 2.54 | 70.81 ± 3.22 | 54.37 ± 2.97 | 60.53 ± 3.18 | 42.43 ± 3.62 | 47.92 ± 4.74 |
| Graph-Mamba | 68.46 ± 3.91 | 72.09 ± 3.16 | 43.86 ± 9.73 | 54.45 ± 9.53 | 63.09 ± 2.82 | 63.63 ± 2.14 | 64.81 ± 12.47 | 67.06 ± 8.99 | 13.64 ± 5.97 | 27.37 ± 14.82 | 17.63 ± 3.07 | 33.21 ± 0.52 |
| Graph-Mamba + UniImb | 74.36 ± 4.22 | 76.52 ± 2.95 | 71.09 ± 1.05 | 72.35 ± 2.32 | 67.32 ± 3.14 | 68.53 ± 4.10 | 73.89 ± 2.47 | 73.94 ± 2.85 | 52.33 ± 1.41 | 57.61 ± 3.02 | 37.74 ± 2.25 | 43.30 ± 3.48 |

## I.8 Applicability analysis of Dynamic Balanced Prototype

To further validate the effectiveness of our core module DBP, we conducted experiments by incorporating DBP into GIN, GCN, and GraphSage on the REDDIT-B and IMDB-MULTI datasets under extreme class-imbalance scenarios. The results, as shown in Table 24, indicate that only adding DBP to these vanilla GNNs significantly improves performance, demonstrating its effectiveness in mitigating imbalance. This also confirms that our two-stage design, prototype-perception and prototype-balance, enables balanced interactions between each graph and its most similar prototypes, encouraging the prototypes to capture critical representations and thereby enhancing learning for tail classes.

Table 24: Analysis of Only DBP Performance on class imbalanced datasets with *extreme imbalance degree*.

| Method | REDDIT-B | | IMDB-MULTI | |
|---|---|---|---|---|
| | Macro-F1 | Micro-F1 | Macro-F1 | Micro-F1 |
| GIN | 33.19 ± 14.26 | 36.02 ± 14.38 | 13.25 ± 6.19 | 14.92 ± 5.43 |
| **GIN + DBP** | **73.26** ± 3.30 | **85.31** ± 3.43 | **30.27** ± 2.28 | **43.41** ± 3.62 |
| GCN | 29.19 ± 1.38 | 31.00 ± 2.52 | 10.75 ± 4.10 | 12.82 ± 4.16 |
| **GCN + DBP** | **71.49** ± 1.36 | **84.60** ± 1.40 | **29.27** ± 3.17 | **41.65** ± 2.86 |
| GraphSage | 27.23 ± 2.21 | 29.62 ± 1.99 | 12.26 ± 2.72 | 14.01 ± 2.32 |
| **GraphSage + DBP** | **64.29** ± 4.92 | **78.28** ± 5.37 | **30.69** ± 1.03 | **44.26** ± 0.84 |

## I.9 GRAPH STRUCTURE VISUALIZATION

We further analyze the effectiveness of prototypes. Taking the highly class-imbalanced COLLAB dataset as an example, we extract the attention matrix between prototypes and graph instances for prototype feature extraction (refer to Eq. 6). Based on this matrix, for a given anchor prototype we retrieve the three minority-class graph instances with the highest affinity to that prototype, as shown in Figure 15(a)–(c). We further examine the graph structures of these example graphs and find they all contain the same local topology (as shown in (d)), a topology that is important in each graph (because each node has high degree). This indicates that prototypes can capture discriminative feature for minority classes, thereby improving minority-class classification performance.

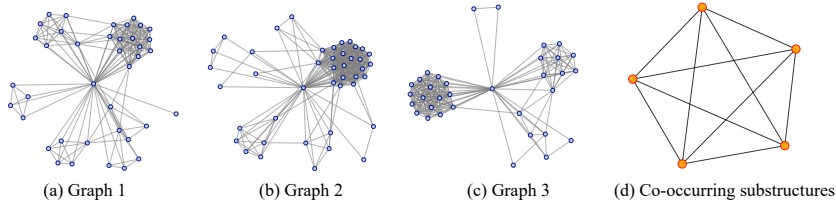

| (a) Graph 1 | (b) Graph 2 | (c) Graph 3 | (d) Co-occurring substructures |

Figure 15: Visualization analysis of prototype effects.

## J LIMITATION DISCUSSION AND FUTURE WORK

❶ Broaden Dataset Evaluation Across Diverse Scenarios : Our current datasets primarily consist of homogeneous graphs, which do not fully capture the diversity and complexity of real-world networks. In the future, we will evaluate the performance on heterogeneous graphs that contain multiple types of nodes and edges, enabling a more comprehensive assessment across varied scenarios.

❷ Compare Imbalanced Graph Learning Algorithms Across Various Tasks : We aim to extend the comparison of imbalanced graph learning algorithms beyond classification to include tasks such as few-shot learning, dynamic graph learning, and anomaly detection. By broadening the range of tasks, we can gain deeper insights into the progress of the field and better understand how different algorithms perform across diverse applications.

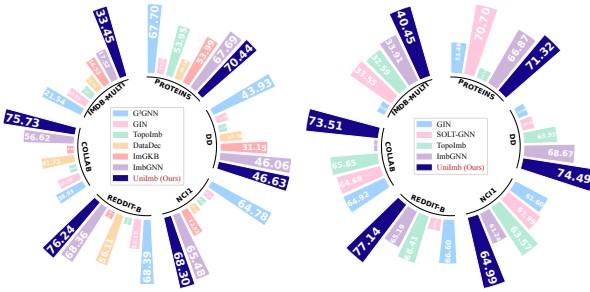

Figure 16: Macro-F1 on class-imbalance and topological-imbalance datasets with extreme imbalance degree.

