# OpenReview forum: "One for Two: A Unified Framework for Imbalanced Graph Classification via Dynamic Balanced Prototype"
_ICLR.cc/2026/Conference — ICLR 2026 Oral_

### Official Review · Reviewer_5aWh · 2025-10-31

**Soundness:** 3
**Presentation:** 4
**Contribution:** 3
**Rating:** 6
**Confidence:** 5

**Summary:**

In this submission, a unified framework, UniImb, for handling both class imbalance and topological imbalance in graph classification tasks is prposed. The core innovation lies in the introduction of a dynamic balanced prototype module, coupled with a prototype load-balancing regularizer, which ensures equitable influence from tail graphs. Theoretically, the authors provide an information-theoretic justification for the prototype balancing mechanism based on the Information Bottleneck principle. Extensive experiments demonstrate that UniImb achieves competitive performance.

**Strengths:**

S1. The authors are the first to unify the modeling of class imbalance and topological imbalance in graph classification, filling a notable gap in existing research.

S2. The proposed dynamic prototype method is elegantly designed and well-grounded theoretically. And the proposed UniImb framework is highly versatile and plug-and-play, which can be seamlessly integrated into various GNN backbones.

S3. Another key strength of this work is its exceptionally comprehensive and thorough experimental evaluation, encompassing 13 datasets, over 20 baselines, and the introduction of a new large-scale graph classification benchmark. Across all settings, UniImb delivers significant performance improvements.

**Weaknesses:**

W1. The model contains several key hyperparameters. Although the authors analyzed the sensitivity of each individually in the original paper, a sensitivity analysis of combinations of two hyperparameters is also necessary.

W2. While the overall framework is interesting, most individual components seems like incremental.

W3. All datasets are from common benchmarks. It would be stronger to include an industrial-scale or biomedical use case with real-world imbalance.

**Questions:**

Q1. What does "Basis" mean in Table 1?

Q2. What is the final form of the model's loss? How is the classification loss combined with Equation 11?

Q3. Why do the authors claim that "Methods focused on topology imbalance adapting to small graphs but typically ignore imbalanced class distributions"? Why can't existing models handle these two types of imbalance effectively?

Q4. Imbalance learning baselines such as upsampling have only been evaluated on GNNs; can their effectiveness be extended or improved when applied to graph Transformers?

Minor errors:

(1) The red boxes in the article's appendix can be deleted; they seem to serve no purpose.

(2) The table caption content is overly redundant.

(3) Bold text in table captions should be used with caution.

---

> ### Author Response · Authors · 2025-11-21
> **Part 1 of Rebuttal (1/3)**
>
> > **W1**.Combinatorial Hyperparameter Experiments
>
> Thanks for your suggestion. We have conducted a sensitivity analysis on $\text{TopK}_1$ and $\text{TopK}_2$ on class-imbalanced D&D dataset. For your convenience, we present a subset of the experimental results, reported in terms of Macro-F1, below. And the complete overview of the hyperparameter experiments can be found in $\underline{\text{Appendix I.6}}$.
>
> | TopK_1 | TopK_2 = 1         | 2              | 4              | **8**              | 12             | 16             |
> |:--------:|:----------------------:|:----------------:|:----------------:|:----------------:|:----------------:|:----------------:|
> | 16     | 43.62 ± 2.35         | 44.04 ± 1.34   | 45.19 ± 1.53   | 41.63 ± 4.33   | 37.86 ± 6.95   | 46.24 ± 1.65   |
> | 24     | 44.42 ± 1.41         | 44.73 ± 1.39   | 45.24 ± 2.34   | 43.91 ± 1.93   | 44.83 ± 2.44   | 45.03 ± 4.92   |
> | **32**     | 44.70 ± 2.13         | 44.90 ± 1.95   | 42.57 ± 6.12   | **46.63 ± 3.42**   | 45.36 ± 1.82   | 44.47 ± 1.78   |
> | 48     | 38.45 ± 11.75        | 45.70 ± 1.59   | 44.82 ± 1.66   | 40.72 ± 6.95   | 44.60 ± 1.52   | 41.56 ± 2.50   |
>
>
> > **W2**. While the overall framework is interesting, most individual components seems like incremental.
>
> Sorry for the confusion. We would like to reiterate that our core contribution is the development of a dynamic balanced prototype strategy, enhanced by the introduction of a prototype balancing regularization term (DBP). This innovation is grounded in information bottleneck theory, which strengthens its theoretical foundation. This design effectively addresses the limitations of existing methods in managing complex graph imbalance scenarios.
>
> To further clarify any remaining concerns, we demonstrate the standalone effectiveness of DBP without integrating any additional strategies through ablation studies, comparative evaluations, and scalability experiments under extremely class-imbalance scenarios.
>
>
> - Ablation & Comparison Experiments
>
> The experimental results demonstrate that the GIN+DBP (only) variant, which retains only the DBP component while removing all other components, achieves performance close to the full model. This indicates that the majority of our model’s performance improvement stems from the core DBP component. Meanwhile, using DBP alone consistently outperforms existing models, confirming its effectiveness.
>
> | Macro-F1 | REDDIT-B | IMDB-MULTI |
> |:--------------:|:--------------:|:--------------:|
> | GIN | 33.19 ± 14.26| 13.25 ± 6.19|
> | G²GNN  |68.39 ± 2.97|20.67 ± 9.88|
> |ImbGNN|68.36 ± 8.10 |17.52 ± 8.98|
> | GIN + DBP (only) | 73.26  ± 3.30| 30.27 ± 2.28|
> | GIN + Full | 76.24 ± 4.09| 33.45 ± 7.83 |
>
> - Scalability Experiment
>
> We integrated the standalone DBP module with multiple GNN variants, and the experimental results are presented in the table below. The results further demonstrate that DBP can significantly improve the imbalanced learning performance of these GNNs.
>
> | Macro-F1 | REDDIT-B | IMDB-MULTI |
> |:--------------:|:--------------:|:--------------:|
> | GIN | 33.19 ± 14.26 | 13.25 ± 6.19 |
> | GIN + DBP (only) | **73.26 ± 3.30** | **30.27 ± 2.28** |
> | GCN | 29.19 ± 1.38 | 10.75 ± 4.10 |
> | GCN + DBP (only) | **71.49 ± 1.36** | **29.27 ± 3.17** |
> | GraphSage | 27.23 ± 2.21 | 12.26 ± 2.72 |
> | GraphSage + DBP | **64.29 ± 4.92** | **30.69 ± 1.03** |

---

> ### Author Response · Authors · 2025-11-21
> **Part 2 of Rebuttal (2/3)**
>
> > **W3**. All datasets are from common benchmarks. It would be stronger to include an industrial-scale or biomedical use case with real-world imbalance.
>
> Sorry for the confusion. In fact, beyond the 12 commonly used datasets, **we have additionally open-sourced a large-scale air quality graph dataset**, which collects air pollution levels from monitoring stations across mainland China from 2021 to 2023. Detailed experiments can be found in Section 5.3 and Appendix G.1.2.
>
> Furthermore, to address your concern, we have supplemented our evaluation with three datasets under extremely class-imbalance scenarios from **biological** and **the visual**: ENZYMES, Letter-high, and Letter-low. The experimental results are shown in the table below.
>
> | Macro-F1 | ENZYMES | Letter-high | Letter-low |
> |:----------:|:----------:|:----------:|:----------:|
> | GIN      | 10.15 ± 4.36 | 25.62 ± 2.77 | 23.08 ± 3.83 |
> | G²GNN    | 17.94 ± 1.35 | 29.40 ± 5.39 | 31.28 ± 2.47 |
> | ImbGNN   | 14.76 ± 2.06 | 31.96 ± 3.47 | 27.29 ± 6.54 |
> | **UniImb**| **41.38 ± 2.37** | **39.62 ± 1.38** | **43.86 ± 2.48** |
>
>
> Finally, we incorporate graph datasets with 3D conformation [1]: BBBP for blood-brain barrier permeability prediction, BACE for β-secretase inhibition activity prediction, and HIV for antiviral activity prediction. Notably, we employ specialized models capable of processing three-dimensional data. The results are presented in the table below. Our model achieves excellent performance on these datasets.
>
> | Macro-F1      | HIV   | BBBP | BACE  |
> |:--------------:|:--------------:|:--------------:|:--------------:|
> | SchNet       | 49.13 ± 42.16      | 43.44 ± 0.94       | 57.09 ± 4.18       |
> | +G²GNN        | 52.46 ± 5.29       | 58.94 ± 5.64       | 61.68 ± 1.30       |
> | +TopoImb      | 50.19 ± 3.11       | 46.82 ± 2.22       | 54.30 ± 6.72       |
> | +ImbGNN       | 54.67 ± 4.63       | 62.20 ± 4.61       | 63.26 ± 3.11       |
> | **+DBP(Ours)**  | **58.91 ± 4.90** | **73.06 ± 1.19**| **67.13 ± 2.83** |
>
> | Macro-F1      | HIV  | BBBP | BACE|
> |:--------------:|:--------------:|:--------------:|:--------------:|
> | DimeNet       | 49.68 ± 0.66      | 41.72 ± 2.97      | 53.13 ± 7.01      |
> | +G²GNN         | 51.28 ± 3.33      | 64.28 ± 5.21      | 58.92 ± 5.41      |
> | +Topolmb       | 51.07 ± 2.28      | 48.96 ± 10.18     | 54.83 ± 6.19      |
> | +ImbGNN        | 53.75 ± 1.29      | 66.39 ± 2.53      | 62.22 ± 3.71      |
> | **+DBP(Ours)**  | **56.54 ± 2.10**      | **73.45 ± 4.19**| **70.80 ± 3.59**|
>
> | Macro-F1   | HIV | BBBP  | BACE |
> |:--------------:|:--------------:|:--------------:|:--------------:|
> | SphereNet       | 50.68 ± 1.50      | 48.62 ± 8.48      | 45.25 ± 5.27     |
> | +G²GNN         | 54.96 ± 4.99      | 62.65 ± 2.85      | 52.25 ± 1.03      |
> | +Topolmb       | 52.40 ± 2.31      | 53.21 ± 6.72     | 47.81 ± 6.71      |
> | +ImbGNN        | 56.79 ± 2.62      | 65.92 ± 2.30      | 56.62 ± 2.43      |
> | **+DBP(Ours)** | **60.29 ± 3.18**      | **76.34 ± 2.16**      | **62.30 ± 3.99** |
>
> [1] Wu Z, Ramsundar B, Feinberg E N, et al. MoleculeNet: a benchmark for molecular machine learning[J]. Chemical science, 2018, 9(2): 513-530.
>
> [2] Schütt K, Kindermans P J, Sauceda Felix H E, et al. Schnet: A continuous-filter convolutional neural network for modeling quantum interactions[J]. Advances in neural information processing systems, 2017, 30.
>
> [3] Gasteiger J, Groß J, Günnemann S. Directional message passing for molecular graphs[J]. arXiv preprint arXiv:2003.03123, 2020.
>
> [4]Liu Y, Wang L, Liu M, et al. Spherical message passing for 3d molecular graphs[C]//International Conference on Learning Representations (ICLR). 2022.

---

> ### Author Response · Authors · 2025-11-21
> **Part 3 of Rebuttal (3/3)**
>
> > **Q1**. What does "Basis" mean in Table 1?
>
> We apologize for the confusion. “Basis” refers to the key technique or model applied in the method. For example, “G²GNN + remove edge” indicates that the remove edge technique is integrated with their proposed G²GNN model, while “up-sampling + GIN” means that we apply the up-sampling technique to GIN.
>
> > **Q2**. Optimization Loss
>
> We apologize for any confusion. **The cross-entropy classification loss and the Prototype-balancing Optimization loss in Equation 11 are two independent losses without any weighted fusion process**. Specifically, the cross-entropy classification loss is used to constrain the neural network parameters, while the Prototype-balancing Optimization is dedicated to optimizing the activation regulation coefficient to ensure balanced perception of prototypical features.
>
> In the revised manuscript, we have added an algorithmic pseudocode in $\underline{\text{Appendix E}}$ to clarify this process, and we provide implementation details in the anonymized code repository.
>
> > **Q3**. Gaps.
>
> Sorry for any confusion. These two types of architectures are typically designed to address only a single aspect of imbalance, either class imbalance or topological imbalance. Specifically, class imbalance methods require models to “enhance underrepresented classes,” a label-centric objective that neglects variations in graph scale. In contrast, topological imbalance methods encourage models to “enhance small-scale graph instances,” a size-focused objective that ignores class distribution. For example, TopoImb employs a training modulator to adjust the learning process for small-scale graphs. Consequently, neither architecture handles complex imbalance scenarios effectively.
>
> Empirically, the topological imbalance model TopoImb performs unsatisfactorily on class-imbalanced datasets, underperforming its backbone model GIN (as shown in Table 1). Similarly, the class-imbalance learning model DataDec performs worse than conventional GNNs in experiments with higher degrees of class imbalance (Table 1 and Table 14) or in multi-class classification tasks (Table 1). **In summary, current models are unable to effectively handle diverse complex graph imbalance scenarios within a single unified framework.**
>
> > **Q4**. Imbalance learning baselines such as upsampling have only been evaluated on GNNs; can their effectiveness be extended or improved when applied to graph Transformers?
>
> Thank you very much for your suggestion. We have integrated the up-sampling and re-weighting methods with representative graph transformer models. The experimental results (Macro-F1) are presented below.
>
> | Up-sampling | PROTEINS | D&D | REDDIT-B |
> |:--------------:|:--------------:|:--------------:|:--------------:|
> | GraphGPS    | 66.71 ± 3.41 | 40.76 ± 2.29 | 52.68 ± 1.83 |
> | Expormer    | 65.28 ± 1.93 | 38.52 ± 6.93 | 60.16 ± 4.10 |
> | Graph-Mamba | 66.98 ± 5.86 | 35.20 ± 1.52 | 58.26 ± 6.42 |
> |UniImb|**70.44± 4.72**|**46.63± 3.42**|**76.24± 4.09**|
>
> | re-weighting | PROTEINS | D&D | REDDIT-B |
> |:--------------:|:--------------:|:--------------:|:--------------:|
> | GraphGPS    | 63.52 ± 2.19 | 41.29 ± 2.01 | 50.37 ± 4.47 |
> | Expormer    | 64.92 ± 4.62 | 37.31 ± 4.24 | 61.62 ± 3.06 |
> | Graph-Mamba | 61.67 ± 8.74 | 36.87 ± 7.16 | 63.10 ± 2.82 |
> |UniImb|**70.44± 4.72**|**46.63± 3.42**|**76.24± 4.09**|
>
> Minor errors:
>
> Thank you very much for pointing these out. We will correct these minor errors in the revision.

---

> > ### Author Response · Authors · 2025-11-26
> > **Looking forward to further discussion**
> >
> > Dear Reviewer 5aWh,
> >
> > We sincerely appreciate the thoughtful feedback you provided on our submission. As the discussion period is now halfway through, we would be very grateful if you could take a moment to engage in the discussion. Following your review, we have prepared a point-by-point addressing the weaknesses and questions you raised:
> >
> > - **[Part 1 of the Response  (W1 & W2)]:** We have conducted combined hyperparameter experiments and added ablation and scalability experiments to validate the core contribution of DBP.
> > - **[Part 2 of the Response  (W3)]:** We have open-sourced the AirGraph dataset and introduced biological multi-class datasets along with 3D graph datasets to demonstrate the method’s effectiveness in addressing real-world imbalance challenges.
> > - **[Part 3 of the Response  (Q1 & Q2 & Q3 & Q4)]:** We have addressed some confusions and clarified descriptions, added experiments combining Graph Transformers with imbalance strategies, and corrected minor issues in the revised version.
> >
> > Thank you again for your valuable suggestions.
> >
> > Warm regards,
> >
> > Authors

---

### Official Review · Reviewer_1cXq · 2025-11-01

**Soundness:** 4
**Presentation:** 4
**Contribution:** 3
**Rating:** 6
**Confidence:** 4

**Summary:**

The paper presents a method for improvements in imbalance learning on graphs. Specifically, the problem is the graph classification setting that assumes the data can have both class and topological imbalance. The method contains multiple components, but the key innovations are the learned augmentations and dynamic prototypes. Across a large set of datasets the authors show significant improvements over prior work in this domain.

**Strengths:**

S1. The motivation for not applying graph augmentations/perturbations uniformly makes sense, which leads to their proposed personalized strategy to learn them based on the graph characteristics.

S2. The dynamic balanced prototype strategy is interesting, having solid theoretical support, and appears to provide great empirical gains.

S3. The level of detail in conducing such comprehensive experimental results is very appreciated and provides the community with rich results to further understand the present method beyond a typical work.

**Weaknesses:**

W1. Generally the problem of imbalance classification on graphs is interesting, but at this point in time there have been numerous works in this direction. Thus, although there is empirical performance gains, the problem itself is well established and the methodology is somewhat lacking key novelty by combining so many techniques in an attempt to obtain the highest performing method (as compared to just focusing on the core innovations of the work).

For further less critical weaknesses, please see the comments and questions below.

**Questions:**

1. Was there any issue with running the TopoImb baseline? The results there look unreasonably low.

2. Is it possible to compare against more recent methods, e.g., [1] or others that have shown performance gains over ImbGNN?

3. Although focused on graph-level tasks, providing a bit more related work on other imbalance problems could be beneficial (e.g., representative works in node, edge, and potentially subgraph), but the discussion in relation to the vision domain seems less relevant.

4. Can you elaborate more on specifically why a contribution is unifying class and topological imbalance? Note that the prior work ImbGNN also discusses this innovation when building the connection between them.

5. As compared to cherry picking the best result to make claims like "UniImb delivers up to 41.62% relative improvement on ..." providing averages across all datasets would be a preferred and less biased approach for the reader to consume the summarized information (note that already the average improvement is quite significant and need not just report the "up to" results here to show the empirical success/benefits of the proposed method).

6. The work "DPGNN: Dual-Perception Graph Neural Network for Representation Learning" is cited as providing a prototype based approach, but does not appear to present such method.

7. Most of the datasets are binary classification and have overlapping domains. Are there datasets with more classes that could be explored (potentially diverse domains)? How would this prototype-based framework handle many classes? You may consider looking here [2].

Based on the response to the comments and questions above, I am open to revising my score.

[1] SamGoG: A Sampling-Based Graph-of-Graphs Framework for Imbalanced Graph Classification
[2] Graph Prototypical Networks for Few-shot Learning on Attributed Networks

---

> ### Author Response · Authors · 2025-11-21
> **Part 1 of Rebuttal (1/3)**
>
> **We sincerely appreciate your valuable feedback and the time you have dedicated to reviewing our manuscript. Your comments have been immensely helpful in refining our work, and we have provided detailed responses to all the points raised in our reply below.**
>
> > **W1**. Generally the problem of imbalance classification on graphs is interesting, but at this point in time there have been numerous works in this direction. Thus, although there is empirical performance gains, the problem itself is well established and the methodology is somewhat lacking key novelty by combining so many techniques in an attempt to obtain the highest performing method (as compared to just focusing on the core innovations of the work).
>
> We appreciate your guidance and agree that a strong paper should focus on its core contribution rather than integrating many techniques for performance gains. Your comments have prompted reflection; we will adjust our research mindset and implement your suggestions in future work.
>
> Our primary contribution is an adaptive method to maintain balanced prototype representations, coupled with a prototype-balancing regularizer to improve performance under class imbalance. We also provide a principled theoretical justification based on the information bottleneck.
>
> To address any remaining concerns, we validate the independent effectiveness of the dynamic balanced prototype strategy through ablation studies, comparative evaluations, and scalability experiments under extremely class-imbalance scenarios, all conducted without integrating additional techniques.
>
> - **Ablation & Comparison Experiments**
>
> Experimental results indicate that the GIN variant retaining only the dynamic balanced prototype component achieves performance close to that of the full model, suggesting that the principal gains originate from DBP. In addition, the + DBP (only) variant consistently outperforms existing baselines, further validating the effectiveness of DBP.
>
> | Macro-F1 | REDDIT-B | IMDB-MULTI |
> |:--------------:|:--------------:|:--------------:|
> | GIN | 33.19 ± 14.26| 13.25 ± 6.19|
> | G²GNN  |68.39 ± 2.97|20.67 ± 9.88|
> |ImbGNN|68.36 ± 8.10 |17.52 ± 8.98|
> | GIN + DBP (only) | 73.26  ± 3.30| 30.27 ± 2.28|
> | GIN + Full | 76.24 ± 4.09| 33.45 ± 7.83 |
>
> - **Scalability Experiment**
>
> We incorporated the standalone dynamic balanced prototype module into several GNNs. As shown in the below table, these experiments demonstrate that DBP scales effectively and substantially improves these GNNs’ performance on imbalanced datasets.
>
> | Macro-F1 | REDDIT-B | IMDB-MULTI |
> |:--------------:|:--------------:|:--------------:|
> | GIN | 33.19 ± 14.26 | 13.25 ± 6.19 |
> | GIN + DBP (only) | **73.26 ± 3.30** | **30.27 ± 2.28** |
> | GCN | 29.19 ± 1.38 | 10.75 ± 4.10 |
> | GCN + DBP (only) | **71.49 ± 1.36** | **29.27 ± 3.17** |
> | GraphSage | 27.23 ± 2.21 | 12.26 ± 2.72 |
> | GraphSage + DBP | **64.29 ± 4.92** | **30.69 ± 1.03** |
>
> > **Q1**. Reasons for TopoImb's poor performance in Table 1.
>
> TopoImb focuses on topological imbalance by increasing the weights of small-scale graphs during training to improve representation learning for small graphs. However, TopoImb does not take class distribution into account and lacks targeted support for minority classes. Its weighting mechanism may unintentionally favor small graphs that belong to majority classes, producing a “rich-get-richer” effect. Consequently, TopoImb performs poorly on class-imbalance experiments (see Table 1) and in some cases even underperforms classic GNNs, and it attains better results on certain metrics in topological-imbalance experiments (see Table 2).
>
> > **Q2**. Is it possible to compare against more recent methods, e.g., [1] or others that have shown performance gains over ImbGNN?
>
> Thank you very much for your suggestion. Since SamGoG is a contemporaneous work, its preprint did not release the code. To address your question, we re-run UniImb according to the experimental setup described in their paper and compared our results with the results reported by SamGoG. As shown in the table below, our model achieves superior classification performance campared with SamGoG.
>
> **Under high class imbalance scenarios**
> | Macro-F1| D&D | REDDIT-B |
> |---------------|----------|----------|
> | SamGoG| 69.20 ± 1.92 | 78.36 ± 1.58 |
> | **Ours** | **73.94 ± 1.45** |**82.18 ± 2.73** |
>
> **Under high topological imbalance scenarios**
> | Macro-F1| D&D | REDDIT-B |
> |--------------------|----------|----------|
> | SamGoG  | 68.08 ± 1.53 | 71.92 ± 2.13 |
> | **Ours** | **74.49 ± 1.13** | **77.14 ± 10.05** |
>
> **More models**. In fact, the development of imbalanced graph learning is still at an exploratory stage. Compared to ImbGNN developed in 2024, as of 2025 no new method has appeared that can provide strong evidence that its performance surpasses ImbGNN. The UniImb we propose fills an important gap in the current research.
>
> [1] SamGoG: A Sampling-Based Graph-of-Graphs Framework for Imbalanced Graph Classification

---

> ### Author Response · Authors · 2025-11-21
> **Part 2 of Rebuttal (2/3)**
>
> > **Q3**. Related work on other imbalance problems could be beneficial (e.g., representative works in node, edge, and potentially subgraph)
>
> We sincerely appreciate your suggestion. In Appendix B.2 of the original submission, we have provided an overview of related work on node-level imbalance on graph. **In response to your suggestions, we have made the following four revisions in the revised manuscript**:
>
> - We have removed the section titled “Imbalanced Learning in the Vision Domain.”
>
> - We have refined the discussion of “Node-level Imbalance Learning on Graphs.”
>
> - We have added a new subsection on “Edge-level Imbalance Learning on Graphs.”
>
> - Regarding “subgraph imbalance”, during the rebuttal we searched for relevant literature, but regrettably there is no formal definition of this concept in the existing work. Therefore, we summarize current research on “subgraph extraction” in graph learning and explicitly highlight that “subgraph imbalance” constitutes a promising direction for future research.
>
> You can refer to the details in $\underline{\text{Appendix B}}$ of the revised version.
>
> > **Q4**. Can you elaborate more on specifically why a contribution is unifying class and topological imbalance? Note that the prior work ImbGNN also discusses this innovation when building the connection between them.
>
> We apologize for any confusion. **Our intention is not to claim the “first identification” of class imbalance or topological imbalance phenomena, but rather to emphasize that the proposed UniImb can achieve consistent performance in complex scenarios within a unified framework**. In contrast,  our experiments show that ImbGNN is not always optimal in complex graph imbalance scenarios. In some cases, its performance even falls short of the standard GIN (see Table 1). This suggests that **ImbGNN does not provide a truly unified solution for both types of imbalance**. Below, we provide a more detailed comparison and analysis to support this conclusion.
>
> **Experimental Evidence**. As shown in Table 1 (class imbalance) and Table 2 (topological imbalance), ImbGNN demonstrates limited advantages over other baselines, achieving the second-best performance on 6/24. Its performance further deteriorates on multi-class datasets and under low imbalance degree conditions. For instance, on NCI1, ImbGNN attains a Macro-F1 score of 61.01 ± 2.55, compared to 72.99 ± 1.23 achieved by GIN (see Table 15).
>
> **Analysis**. ImbGNN enhances the representation of small-scale graphs by connecting them with large-scale graphs that share similar topological structures, thereby mitigating topological imbalance. However, the method relies on the assumption that "graphs with similar topologies share the same class labels" and claims to address class imbalance. This assumption exhibits clear limitations, particularly in multi-class datasets. Moreover, augmenting data by concatenating multiple graph instances can introduce information redundancy, increasing the risk of model overfitting.
>
> > **Q5**. Inappropriate presentation issues: "Up to"
>
> Thank you for your suggestion. In the revised version, we removed the statement “up to 41.62%” and instead reported the average performance.
>
> >  **Q6**. The work "DPGNN: Dual-Perception Graph Neural Network for Representation Learning" is cited as providing a prototype based approach, but does not appear to present such method.
>
> We sincerely apologize for the incorrect citation. DPGNN is from the work by Wang et al. [1].
>
> Our approach differs from DPGNN in three key aspects:
>
> - Task goal. DPGNN focuses on node classification tasks, where each node is assigned a fine-grained label. In contrast, our work addresses graph classification tasks, where an entire graph is assigned a single label.
>
> - Definition. DPGNN uses the cluster centroids of graph instances as prototypes. In contrast, our prototypes are learnable and can adaptively capture representative features from the imbalanced data.
>
> - Innovation. We make a substantive contribution to prototype usage : design an interaction mechanism between prototypes and graph instances that proceeds in two stages: extraction followed by enhancement. This interaction is governed by a prototype load balance constraint to ensure that all graph instances are treated equitably.
>
> Reference:
>
> [1] Wang, Yu, Charu Aggarwal, and Tyler Derr. "Distance-wise prototypical graph neural network for imbalanced node classification." Proceedings of the 17th International Workshop on Mining and Learning with Graphs (MLG). 2022.

---

> ### Author Response · Authors · 2025-11-21
> **Part 3 of Rebuttal (3/3)**
>
> > **Q7**. Most of the datasets are binary classification and have overlapping domains. Are there datasets with more classes that could be explored (potentially diverse domains)? How would this prototype-based framework handle many classes? You may consider looking here [2].
>
> Thank you for your suggestion. We have added **three multi-class datasets** and **three graph datasets with 3D conformation** to increase the diversity of domains covered by the experiments.
>
> **Multi-class Datasets.** We have added three additional datasets [1]: ENZYMES with six classes from bioinformatics, Letter-high with fifteen classes from the visual domain, and Letter-low with fifteen classes from the visual domain. The experimental results under extremely class-imbalance scenarios are reported in the table and have been included in $\underline{\text{Appendix I.2}}$  of the revised manuscript.
>
> | Macro-F1 | ENZYMES | Letter-high | Letter-low |
> |:----------:|:----------:|:----------:|:----------:|
> | GIN      | 10.15 ± 4.36 | 25.62 ± 2.77 | 23.08 ± 3.83 |
> | G²GNN    | 17.94 ± 1.35 | 29.40 ± 5.39 | 31.28 ± 2.47 |
> | ImbGNN   | 14.76 ± 2.06 | 31.96 ± 3.47 | 27.29 ± 6.54 |
> | **UniImb**   | **41.38 ± 2.37** | **39.62 ± 1.38** | **43.86 ± 2.48** |
>
> **Datasets with 3D conformation.** We expanded our evaluation with three 3D molecular graph datasets [2]: BBBP for blood brain barrier permeability, BACE for β secretase inhibition, and HIV for antiviral activity. To capture atomic coordinates, interatomic distances and bond angles, we employ geometry aware graph neural networks as base architectures, including SchNet [3], DimeNet [4] and SphereNet [5], and apply imbalance mitigation methods on these models. And we use only the core Dynamic Balanced Prototype module (DBP) of our model.
>
> Macro F1 results in the table show that our DBP continues to lead on these graph datasets with 3D conformation, confirming its applicability to domains with complex structural dependencies. And these experimental results have been added to the revised version in $\underline{\text{Section 5.7}}$.
>
> | Macro-F1      | HIV   | BBBP | BACE  |
> |:--------------:|:--------------:|:--------------:|:--------------:|
> | SchNet       | 49.13 ± 42.16      | 43.44 ± 0.94       | 57.09 ± 4.18       |
> | +G²GNN        | 52.46 ± 5.29       | 58.94 ± 5.64       | 61.68 ± 1.30       |
> | +TopoImb      | 50.19 ± 3.11       | 46.82 ± 2.22       | 54.30 ± 6.72       |
> | +ImbGNN       | 54.67 ± 4.63       | 62.20 ± 4.61       | 63.26 ± 3.11       |
> | **+DBP(Ours)**  | **58.91 ± 4.90**       | **73.06 ± 1.19**       | **67.13 ± 2.83**       |
>
> | Macro-F1      | HIV  | BBBP | BACE|
> |:--------------:|:--------------:|:--------------:|:--------------:|
> | DimeNet       | 49.68 ± 0.66      | 41.72 ± 2.97      | 53.13 ± 7.01      |
> | +G²GNN         | 51.28 ± 3.33      | 64.28 ± 5.21      | 58.92 ± 5.41      |
> | +Topolmb       | 51.07 ± 2.28      | 48.96 ± 10.18     | 54.83 ± 6.19      |
> | +ImbGNN        | 53.75 ± 1.29      | 66.39 ± 2.53      | 62.22 ± 3.71      |
> | **+DBP(Ours)**  | **56.54 ± 2.10**      | **73.45 ± 4.19**      | **70.80 ± 3.59**     |
>
> | Macro-F1   | HIV | BBBP  | BACE |
> |:--------------:|:--------------:|:--------------:|:--------------:|
> | SphereNet       | 50.68 ± 1.50      | 48.62 ± 8.48      | 45.25 ± 5.27     |
> | +G²GNN         | 54.96 ± 4.99      | 62.65 ± 2.85      | 52.25 ± 1.03      |
> | +Topolmb       | 52.40 ± 2.31      | 53.21 ± 6.72     | 47.81 ± 6.71      |
> | +ImbGNN        | 56.79 ± 2.62      | 65.92 ± 2.30      | 56.62 ± 2.43      |
> | **+DBP(Ours)** | **60.29 ± 3.18**      | **76.34 ± 2.16**      | **62.30 ± 3.99** |
>
> Reference:
>
> [1] https://chrsmrrs.github.io/datasets/docs/datasets/
>
> [2] Wu Z, Ramsundar B, Feinberg E N, et al. MoleculeNet: a benchmark for molecular machine learning[J]. Chemical science, 2018, 9(2): 513-530.
>
> [3] Schütt K, Kindermans P J, Sauceda Felix H E, et al. Schnet: A continuous-filter convolutional neural network for modeling quantum interactions[J]. Advances in neural information processing systems, 2017, 30.
>
> [4] Gasteiger J, Groß J, Günnemann S. Directional message passing for molecular graphs[J]. arXiv preprint arXiv:2003.03123, 2020.
>
> [5]Liu Y, Wang L, Liu M, et al. Spherical message passing for 3d molecular graphs[C]//International Conference on Learning Representations (ICLR). 2022.

---

> ### Author Response · Authors · 2025-11-26
> **Looking forward to further discussion**
>
> Dear Reviewer 1cXq,
>
> We sincerely appreciate the thoughtful feedback you provided on our submission. As the discussion period is now halfway through, we would be very grateful if you could take a moment to engage in the discussion. Following your review, we have prepared a point-by-point addressing the weaknesses and questions you raised:
>
> - **[Part 1 of the Response  (W1 & Q1 & Q2)]:** We have conducted extensive ablation and scalability experiments to validate the core contributions of DBP, explained the poor performance of TopoImb, and compared our method with the more relevant baseline, SamGoG.
> - **[Part 2 of the Response  (Q3 & Q4 & Q5 & Q6)]:** We have added more related work, clarified the differences from ImbGNN, and corrected errors in performance descriptions and citations in the revised version.
>
> - **[Part 3 of the Response  (Q7)]:** We have introduced three multi-class datasets and three 3D graph datasets to enhance the domain diversity of our experiments.
>
> Thank you again for your valuable suggestions.
>
> Warm regards,
>
> Authors

---

### Official Review · Reviewer_LB4s · 2025-11-01

**Soundness:** 3
**Presentation:** 3
**Contribution:** 2
**Rating:** 6
**Confidence:** 3

**Summary:**

This paper addresses a comprehensive graph classification scenario involving both class imbalance and topological imbalance, and proposes a method named UniImb. Specifically, UniImb integrates multiple carefully designed techniques and introduces a prototype learning strategy to capture representative features, while a balancing constraint ensures its effectiveness in such imbalanced settings. Extensive experiments demonstrate that the proposed model achives better results.

**Strengths:**

1. The proposed method to extracting prototype features is skillfully designed to address the challenge of graph imbalance.
2. The experimental evaluation is convincing, demonstrating consistently performance gains over existing methods.
3. Mathematical notation is clearly defined and used consistently throughout the manuscript, which is overall well-structured and clearly written.

**Weaknesses:**

1. Adding an algorithm pseudocode box can more clearly show the technical details.
2. Why can't existing models comprehensively handle both types of imbalance? Although the authors provide an explanation in the introduction, a more intuitive clarification would be helpful.
3. The primary experimental results (Tables 1 and 2) evaluate these two problems in isolation. While Section 5.4 does introduce an "intertwined" scenario. This makes it difficult to assess the framework's synergistic advantages. The current evidence in the manuscript demonstrates that Unilmb performs well on two separate tasks, rather than a truly unified solution whose main strength lies in handling their complex co-occurrence.
4. The paper introduces a Personalized Graph Perturbation Strategy, claiming it customizes augmentation for small graphs or minority-class graphs . However, the paper fails to explain or validate why a perturbation strategy based on average degree would be suitable for addressing class imbalance.
3. The AirGraph dataset, which exhibits strong practical application potential, is newly introduced; thus, a more detailed exposition of this experiment is warranted in the main body.

**Questions:**

Refer to the above box.

---

> ### Author Response · Authors · 2025-11-21
> **Rebuttal to Reviewer LB4s**
>
> **We are grateful for the insightful feedback you have provided and the time you have invested in the review of our manuscript. Your suggestions are of paramount importance for the improvement of our work. In the following, we will diligently address each of the points you have raised.**
>
> > **W1**. Adding an algorithm pseudocode box can more clearly show the technical details.
>
> Thank you for your suggestion. We have added the algorithm pseudocode to improve readability. Due to Markdown’s limitations in rendering complex text, please refer to $\underline{\text{Appendix E}}$ in the revised version for the full content.
>
> > **W2**. Gaps.
>
> Sorry for any confusion. These two types of architectures are typically designed to address only a single aspect of imbalance, either class imbalance or topological imbalance. Specifically, class imbalance methods require models to “enhance underrepresented classes,” a label-centric objective that neglects variations in graph scale. In contrast, topological imbalance methods encourage models to “enhance small-scale graph instances,” a size-focused objective that ignores class distribution. For example, TopoImb employs a training modulator to adjust the learning process for small-scale graphs. Consequently, neither architecture handles complex imbalance scenarios effectively.
>
> Empirically, the topological imbalance model TopoImb performs unsatisfactorily on class-imbalanced datasets, underperforming its backbone model GIN (as shown in Table 1). Similarly, the class-imbalance learning model DataDec performs worse than conventional GNNs in experiments with higher degrees of class imbalance (Table 1 and Table 14) or in multi-class classification tasks (Table 1). **In summary, current models are unable to effectively handle diverse complex graph imbalance scenarios within a single unified framework.**
>
>
> > **W3**. "Intertwined" scenario
>
> Sorry for the confusion. Tables 1, 2, and Figure 3 respectively demonstrate the competitive performance of UniImb under three different settings: class imbalance, topological imbalance, and the combination of both. Together, these experiments consistently verify the generalization capability of UniImb across diverse imbalance types and varying imbalance degrees in complex scenarios. *It is worth noting that each experiment involves distinct data preprocessing procedures; therefore, their results are not directly comparable in a quantitative sense.*
>
> In addition, we introduce the AirGraph dataset, which records air pollution levels collected from monitoring stations across mainland China. This dataset **naturally exhibits intertwined class and topological imbalance**: class imbalance arises because pollution events are relatively rare, while topological imbalance results from uneven sensor coverage across provinces due to economic disparities. For your convenience, we reproduce the experimental results in the table below. This dataset further verifies the effectiveness of UniImb in real-world scenarios.
>
> | Model | Macro-F1 | Micro-F1 |
> |:---:|:---:|:---:|
> | GIN | 22.30 ± 4.77 | 52.48 ± 3.21 |
> | G²GNN | 24.59 ± 5.08 | 57.93 ± 4.29 |
> | Topolmb | 26.51 ± 3.42 | 58.74 ± 3.96 |
> | ImbGNN | 25.15 ± 2.26 | 58.20 ± 1.79 |
> | SOLT-GNN | 28.74 ± 5.20 | 61.06 ± 4.62 |
> | **Unilmb** |**39.67 ± 1.82**|**72.49 ± 2.03**|
>
> > **W4**. Ablation Study of Personalized Graph Perturbation Strategy
>
> Sorry for the confusion. The personalized graph perturbation strategy includes a feature perturbation mechanism that applies customized masks to node features and an edge masking mechanism that personalizes edge masking, enabling learnable data augmentation and improving the model’s generalization on tail-class graphs. **In the submitted Appendix I.5**, we perform comprehensive ablation studies of the personalized perturbation strategy and introduce the degree-oriented optional augmentation (DoOA) [1] perturbation as a comparison; we reproduce the results (Macro-F1) you asked about below, which show it is highly effective for addressing class imbalance.
>
> | Model | PROTEINS | DD | NCI1 |
> | :---: |:---: | :---: |:---:|
> | w/o Pertu | 63.9 | 44.8 | 61.0 |
> | Mask node | 64.2 | 43.3 | 61.1 |
> | Drop Edge | 66.4 | 44.5 | 59.0 |
> | DoOA | 68.6 | 45.8 | 64.3 |
> | **Pertu** | **70.4** | **46.6** | **68.3** |
>
> [1] Xu W, Wang P, Zhao Z, et al. When imbalance meets imbalance: Structure-driven learning for imbalanced graph classification[C]//Proceedings of the ACM Web Conference 2024. 2024: 905-913.
>
> > **W5**. AirGraph
>
> Thank you for your interest in AirGraph. In the revised manuscript we provide a comprehensive description of the dataset, detailing its attributes, experimental protocol, and associated analyses, and we will release AirGraph as an open resource to support future research. *During the rebuttal phase we also included multi-class benchmarks and three dimensional conformation graph datasets to broaden evaluation and more thoroughly assess the model’s effectiveness.*

---

> > ### Comment · Reviewer_LB4s · 2025-11-25
> > **Response to Rebuttal**
> >
> > Regarding W2: The claim that "current models are unable to effectively handle diverse complex graph imbalance" is inaccurate. Reference 1 ("When imbalance meets imbalance...") explicitly addresses the "intertwined" scenario. Consequently, this work serves as a critical baseline that should be compared to validate the paper's contribution.
> >
> > Regarding W3: The isolated evaluation in Tables 1 and 2 is still inconvincible. If these datasets lack "intertwined" scenarios, the paper's motivation is questionable. If they do, the authors should not separate the tasks; instead, they should analyze the method's performance on co-occurring imbalances.
> > For instance, even in the "class imbalance" experiments (Table 1), does the model successfully address the generalization issues for head and tail graphs (topological imbalance)? Evaluating these aspects separately fails to demonstrate the advantages of a unified solution.

---

> ### Author Response · Authors · 2025-11-26
> **Thanks for your feedback**
>
> **Thank you very much for your prompt reply.**
>
> > **Regarding W2: The claim that "current models are unable to effectively handle diverse complex graph imbalance" is inaccurate. Reference 1 ("When imbalance meets imbalance...") explicitly addresses the "intertwined" scenario. Consequently, this work serves as a critical baseline that should be compared to validate the paper's contribution.**
>
> Sorry for the confusion. **ImbGNN from Reference 1 has been adopted as a key baseline in our study.** We have evaluated our method against it on 19 diverse datasets, which include commonly used benchmarks, multi-class datasets, and 3D structural graphs. Our approach consistently outperforms ImbGNN across all of them. Please note that ImbGNN does not offer a truly unified solution for complex graph imbalance scenarios. In contrast, our model demonstrates robust effectiveness, thereby filling this critical gap.
>
> **Experimental Evidence.** As shown in Table 1 and Table 2, ImbGNN demonstrates limited advantages over other baselines, achieving the second-best performance on 6/24. Its performance further deteriorates on multi-class datasets and under low imbalance degree conditions. For instance, on NCI1, ImbGNN attains a Macro-F1 score of 61.01 ± 2.55, compared to 72.99 ± 1.23 achieved by GIN (see Table 15).
>
> **Analysis.** ImbGNN enhances the representation of small-scale graphs by connecting them with large-scale graphs that share similar topological structures, thereby mitigating topological imbalance. However, the method relies on the assumption that "graphs with similar topologies share the same class labels" and claims to address class imbalance. This assumption exhibits clear limitations, particularly in multi-class datasets. Moreover, augmenting data by concatenating multiple graph instances can introduce information redundancy, increasing the risk of model overfitting.
>
> > **Regarding W3: The isolated evaluation in Tables 1 and 2 is still inconvincible. If these datasets lack "intertwined" scenarios, the paper's motivation is questionable. If they do, the authors should not separate the tasks; instead, they should analyze the method's performance on co-occurring imbalances. For instance, even in the "class imbalance" experiments (Table 1), does the model successfully address the generalization issues for head and tail graphs (topological imbalance)? Evaluating these aspects separately fails to demonstrate the advantages of a unified solution.**
>
> Sorry for the confusion. We have evaluated the effectiveness of our model “intertwined” scenarios (Figure 2 and Figure 3). Furthermore, we define the superiority of our unified approach as the ability of our model to perform robustly across various types of graph imbalance scenarios, including class imbalance and topological imbalance with varying degrees, as well as intertwined imbalance.
>
> Based on your suggestions, we selected the PROTEINS and REDDIT-B class-imbalance datasets from Table 1, both of which exhibit topological imbalance, with class imbalance ratios of approximately 4 and 6, respectively, for the subsequent experiments. We report the accuracy results for large-scale and small-scale graphs on these two datasets. The results demonstrate that our model can effectively represent graph instances of varying scales and alleviate the challenges posed by topological imbalance.
>
> | Model  | PROTEINS |  | REDDIT-B |  |
> |--------|:----------:|----------|:----------:|----------|
> |        | Small-scale | Large-scale | Small-scale | Large-scale |
> | GIN    | 24.78 ± 5.27 | 54.48 ± 3.75 | 31.59 ± 11.34 | 57.28 ± 8.65 |
> | G2GNN  | 64.73 ± 3.10 | 76.29 ± 2.72 | 83.57 ± 4.63 | 87.53 ± 5.61 |
> | ImbGNN | 62.96 ± 4.67 | 77.43 ± 1.65 | 84.19 ± 2.10 | 87.75 ± 3.22 |
> | Unilmb | **67.18 ± 1.09** | **79.87 ± 3.46** | **87.35 ± 1.98** | **89.64 ± 3.46** |

---

> > ### Comment · Reviewer_LB4s · 2025-11-27
> > **Suggestion on aligning primary experiments with the "intertwined" motivation**
> >
> > Thank you for the detailed response.
> >
> > The current presentation in Tables 1 and 2 still appears less relevant to the paper's core motivation of a "unified framework" for "intertwined" imbalances.
> > I suggest reorganizing the main experimental results to explicitly showcase performance on intertwined scenarios across standard datasets. Rather than treating "intertwined" scenarios as a separate subsection or relying solely on the new AirGraph dataset, utilizing the existing benchmarks in Tables 1 and 2 would provide much stronger evidence for the unified solution.

---

> > > ### Author Response · Authors · 2025-11-27
> > > **Thanks for your suggestion!**
> > >
> > > Reviewer LB4s，
> > >
> > > **Thank you very much for your feedback.**
> > >
> > > We sincerely appreciate your constructive feedback. As the rebuttal period concludes, we are committed to further refining our evaluation of the intertwined imbalance scenarios and will incorporate these improvements into the revised manuscript as appropriate. In fact, **the central objective of our work is to enhance robustness in complex graph imbalance scenarios**—a scope that extends beyond intertwined cases to encompass both class imbalance and topological imbalance across multiple degrees. The broad effectiveness of our approach is convincingly demonstrated through extensive evaluations across 19 diverse datasets.

---

### Official Review · Reviewer_qbQQ · 2025-11-04

**Soundness:** 3
**Presentation:** 4
**Contribution:** 2
**Rating:** 4
**Confidence:** 5

**Summary:**

The paper introduces UniImb, a unified framework tackling both class and topological imbalance in graph classification. The core novelty lies in the Dynamic Balanced Prototype (DBP) module that learns balanced prototype representations to reduce bias toward majority graphs. UniImb enhances graph representations and achieves up to 41.6% and 20.6% gains on class- and topology-imbalanced tasks, respectively.

**Strengths:**

(1) The experimental results are comprehensive.

(2) The imbalance graph classification is a very important problem, and the imbalance phenomenon is very widely encountered across many real-world applications, such as drug discovery.

(3) The paper also introduces another air-pollution graph classification dataset, which might be of interest to the following research.

**Weaknesses:**

(1) The main weakness is the novelty of the study. Although imbalance graph classification is indeed a very important problem, the more realistic setting, as the field advances, should focus on graphs aligned more with reality, such as graphs with 3D structures.

(2) The proposed method stacks many different techniques together, which makes the core contribution hardly justified. Although the ablation study has demonstrated that the most critical part is DBP, it might be more interesting to intuitively demonstrate how DBP can benefit the imbalance graph classification. Specifically, what new signals have been learned by introducing DBP techniques?

**Questions:**

(1) Since the main contribution lies in DBP, it might be more effective to focus solely on this technique rather than stacking it with multiple others, such as TopoEncoding. Emphasizing DBP’s independent efficacy would strengthen the justification of its contribution—particularly if experimental evidence can directly show that DBP alone mitigates class imbalance.

(2) It would also be valuable to provide an intuitive and visual illustration of how DBP alleviates both quantitative and topological imbalance. For instance, does the learned DBP reflect specific topological patterns that are desirable for minority-class test graphs but absent in the minority-class training data? Such visualization could make the mechanism and impact of DBP more tangible and convincing.

(3) My main concern, as noted in the weakness section, lies in the real-world practicality of the study. While the analyzed graphs are important, they do not capture realistic structural properties, such as the 3D conformations of proteins and molecules, which are crucial in real-world applications. This limitation raises concerns about the method’s applicability to domains where complex structural dependencies and class imbalance are pervasive.

---

> ### Author Response · Authors · 2025-11-21
> **Part 1 of Rebuttal (1/2)**
>
> **We are grateful for your insightful comments and the time you invested in reviewing our manuscript. They have been invaluable in helping us improve our work, and we have addressed all of your points in our responses below.**
>
> > **W1**. The main weakness is the novelty of the study. Although imbalance graph classification is indeed a very important problem, the more realistic setting, as the field advances, should focus on graphs aligned more with reality, such as graphs with 3D structures.
>
> Our contribution is the proposal of a unified framework that comprehensively addresses complex scenarios involving both class imbalance and topological imbalance. The core of the model is the dynamic balanced prototype strategy, combined with a prototype-balancing optimization. Additionally, we employ the information bottleneck principle to strengthen the theoretical justification of the method.
>
> In our experiments, we evaluated the model using 13 datasets, most of which are derived from real-world applications. Notably, we also released a **large-scale air quality graph dataset**, which records pollution level data collected from monitoring stations across mainland China.
>
> To further address your concerns, we have incorporated **three graph datasets with 3D conformation [1]: BBBP for blood-brain barrier permeability prediction, BACE for β-secretase inhibition prediction, and HIV for antiviral activity prediction.** To accommodate 3D graph structures and explicitly model atomic three-dimensional coordinates, interatomic distances, and bond angles, we introduce GNN models based on 3D geometric information, including SchNet [2], DimeNet [3], and SphereNet [4], which serve as backbones for graph imbalance learning methods. Following the established class imbalance data handling protocol, the datasets are split into training, validation, and test sets with an 80%/10%/10% ratio. In our model, we retained only the core dynamic balanced prototype module to directly demonstrate the effectiveness of our core contribution.
>
> The experimental results in the table (Macro‑F1) show that our method still achieves leading performance on these graph datasets with 3D conformations, demonstrating its effectiveness on data with complex structural dependencies.
>
> | Macro-F1      | HIV   | BBBP | BACE  |
> |:--------------:|:--------------:|:--------------:|:--------------:|
> | SchNet       | 49.13 ± 42.16      | 43.44 ± 0.94       | 57.09 ± 4.18       |
> | +G²GNN        | 52.46 ± 5.29       | 58.94 ± 5.64       | 61.68 ± 1.30       |
> | +TopoImb      | 50.19 ± 3.11       | 46.82 ± 2.22       | 54.30 ± 6.72       |
> | +ImbGNN       | 54.67 ± 4.63       | 62.20 ± 4.61       | 63.26 ± 3.11       |
> | **+DBP(Ours)**  | **58.91 ± 4.90**       | **73.06 ± 1.19**       | **67.13 ± 2.83**       |
>
> | Macro-F1      | HIV  | BBBP | BACE|
> |:--------------:|:--------------:|:--------------:|:--------------:|
> | DimeNet       | 49.68 ± 0.66      | 41.72 ± 2.97      | 53.13 ± 7.01      |
> | +G²GNN         | 51.28 ± 3.33      | 64.28 ± 5.21      | 58.92 ± 5.41      |
> | +Topolmb       | 51.07 ± 2.28      | 48.96 ± 10.18     | 54.83 ± 6.19      |
> | +ImbGNN        | 53.75 ± 1.29      | 66.39 ± 2.53      | 62.22 ± 3.71      |
> | **+DBP(Ours)**  | **56.54 ± 2.10**      | **73.45 ± 4.19**      | **70.80 ± 3.59**     |
>
> | Macro-F1   | HIV | BBBP  | BACE |
> |:--------------:|:--------------:|:--------------:|:--------------:|
> | SphereNet       | 50.68 ± 1.50      | 48.62 ± 8.48      | 45.25 ± 5.27     |
> | +G²GNN         | 54.96 ± 4.99      | 62.65 ± 2.85      | 52.25 ± 1.03      |
> | +Topolmb       | 52.40 ± 2.31      | 53.21 ± 6.72     | 47.81 ± 6.71      |
> | +ImbGNN        | 56.79 ± 2.62      | 65.92 ± 2.30      | 56.62 ± 2.43      |
> | **+DBP(Ours)** | **60.29 ± 3.18**      | **76.34 ± 2.16**      | **62.30 ± 3.99** |
>
> These results have been integrated into Section 5.7 of the revised version.
>
> Reference:
>
> [1] Wu Z, Ramsundar B, Feinberg E N, et al. MoleculeNet: a benchmark for molecular machine learning[J]. Chemical science, 2018, 9(2): 513-530.
>
> [2] Schütt K, Kindermans P J, Sauceda Felix H E, et al. Schnet: A continuous-filter convolutional neural network for modeling quantum interactions[J]. Advances in neural information processing systems, 2017, 30.
>
> [3] Gasteiger J, Groß J, Günnemann S. Directional message passing for molecular graphs[J]. arXiv preprint arXiv:2003.03123, 2020.
>
> [4] Liu Y, Wang L, Liu M, et al. Spherical message passing for 3d molecular graphs[C]//International Conference on Learning Representations (ICLR). 2022.

---

> ### Author Response · Authors · 2025-11-21
> **Part 2 of Rebuttal (2/2)**
>
> > **W2.** The proposed method stacks many different techniques together, which makes the core contribution hardly justified. Although the ablation study has demonstrated that the most critical part is DBP, it might be more interesting to intuitively demonstrate how DBP can benefit the imbalance graph classification. Specifically, what new signals have been learned by introducing DBP techniques?
>
> Thank you very much for your suggestion. To address your concern, we take the extremely class‑imbalanced COLLAB dataset as an example. We first extracted the attention matrix between prototypes and graph instances from the prototype‑perception process. Using this matrix, we selected an anchor prototype and retrieved the three minority‑class graph instances with the highest affinity to that prototype (shown in the revised manuscript as Fig. 7(a)–(c)). Further analysis of these examples revealed that they all share the same local substructure (see Fig. 7(d)), a substructure that is relatively important within each graph instance because its constituent nodes have high degrees. *This finding indicates that the prototype captures discriminative features of the graph instances, thereby improving minority‑class classification performance.*
>
> As the rebuttal system’s Markdown does not support inline figure display, we have added this discussion to Experiment Section 5.9 of the revised manuscript.
>
> >**Q1**. Since the main contribution lies in DBP, it might be more effective to focus solely on this technique rather than stacking it with multiple others, such as TopoEncoding. Emphasizing DBP’s independent efficacy would strengthen the justification of its contribution—particularly if experimental evidence can directly show that DBP alone mitigates class imbalance.
>
> Thank you for your valuable suggestion. To address your concerns, we made **two efforts**. First, we evaluated the contribution of DBP by integrating it individually with the GIN backbone under extremely class-imbalance scenarios. Second, we combined the standalone DBP module with multiple GNN backbones to assess its general effectiveness across diverse architectures.
>
> **(1) Ablation & Comparison Experiments**
>
> Experimental results show that the variant retaining only the DBP component achieves near-equivalent performance to the full model, indicating that the primary gains originate from this core element. In isolation, the DBP component consistently surpasses existing baselines, further confirming its central role and effectiveness.
>
> | Macro-F1 | REDDIT-B | IMDB-MULTI |
> |:--------------:|:--------------:|:--------------:|
> | GIN | 33.19 ± 14.26| 13.25 ± 6.19|
> | G²GNN  |68.39 ± 2.97|20.67 ± 9.88|
> |ImbGNN|68.36 ± 8.10 |17.52 ± 8.98|
> | GIN + DBP (only) | 73.26 ± 3.30| 30.27 ± 2.28|
> | GIN + Full | 76.24 ± 4.09| 33.45 ± 7.83 |
>
> **(2) Scalability Experiment**
>
> We integrated the standalone DBP module with multiple GNNs, and the experimental results are presented in the table below. The results demonstrate that DBP can significantly improve the imbalanced learning performance of these GNNs.
>
> | Macro-F1 | REDDIT-B | IMDB-MULTI |
> |:--------------:|:--------------:|:--------------:|
> | GIN | 33.19 ± 14.26 | 13.25 ± 6.19 |
> | GIN + DBP (only)| **73.26 ± 3.30** | **30.27 ± 2.28** |
> | GCN | 29.19 ± 1.38 | 10.75 ± 4.10 |
> | GCN + DBP (only) | **71.49 ± 1.36** | **29.27 ± 3.17** |
> | GraphSage | 27.23 ± 2.21 | 12.26 ± 2.72 |
> | GraphSage + DBP | **64.29 ± 4.92** | **30.69 ± 1.03** |
>
> > **Q2**. It would also be valuable to provide an intuitive and visual illustration of how DBP alleviates both quantitative and topological imbalance. For instance, does the learned DBP reflect specific topological patterns that are desirable for minority-class test graphs but absent in the minority-class training data? Such visualization could make the mechanism and impact of DBP more tangible and convincing.
>
> Please refer to W2.
>
> > **Q3**. My main concern, as noted in the weakness section, lies in the real-world practicality of the study. While the analyzed graphs are important, they do not capture realistic structural properties, such as the 3D conformations of proteins and molecules, which are crucial in real-world applications. This limitation raises concerns about the method’s applicability to domains where complex structural dependencies and class imbalance are pervasive.
>
> Please refer to W1. As of now, our model has been comprehensively evaluated against `23` baseline methods. The results demonstrate its consistent superiority across `19` diverse datasets, including widely-used benchmarks in the graph learning, multiclass datasets, our novel AirGraph dataset, and datasets with 3D conformation .

---

> ### Author Response · Authors · 2025-11-26
> **Looking forward to further discussion**
>
> Dear Reviewer qbQQ,
>
> We sincerely appreciate the thoughtful feedback you provided on our submission. As the discussion period is now halfway through, we would be very grateful if you could take a moment to engage in the discussion. Following your review, we have prepared point-by-point responses addressing the weaknesses and questions you raised:
>
> - **[Part 1 of the Response  (W1)]:**
> We have clarified the misunderstanding regarding the novelty of our submission and introduced three graph datasets with 3D conformations to validate the effectiveness of our method in addressing real-world graph imbalance challenges involving complex structural dependencies.
> - **[Part 2 of the Response  (W2 & Q1 & Q2 & Q3)]:**
> We have conducted additional ablation and scalability analyses to further clarify the contribution of our DBP, and used visualizations to intuitively highlight the key topological patterns it captures.
>
> Thank you again for your valuable suggestions.
>
> Warm regards,
> Authors

---

> > ### Comment · Reviewer_qbQQ · 2025-11-26
> > **Thank you for your response**
> >
> > Thank you for your response.
> >
> > (1) It is great to see additional experiments demonstrating the effectiveness of the proposed imbalance mitigation problem also on 3D graph classification scenarios, which will be more realistic.
> >
> > (2) For W2, although from the Figure 7 visualization, we can see the retrieved top-3 graphs share some common substructure, they do not necessarily indicate that the trained model can capture this substructure and leverage this substructure to conduct graph classification. In addition, we also need to demonstrate that this substructure is not very common in the majority of classes. Would it be possible to extend this analysis a little bit more? Thank you so much!

---

> > > ### Author Response · Authors · 2025-11-29
> > >
> > > Thank you for acknowledging our response. Based on your suggestion, we have further expanded our content analysis.
> > >
> > > First, we adopted the Shapley value analysis to quantify the actual contribution of the key substructure to the model’s predictions. The Shapley value is a commonly used method for evaluating the independent contribution of subcomponents. Specifically, as shown in Figure 7 of the manuscript, we masked the identified key substructure and compared the variant's predictions with the original predictions. Through this approach, we found that the prediction contribution from the remaining graph structure was only 6.84%, while the key substructure contributed as much as 93.16%. This result clearly demonstrates that the model has indeed learned and relied on this discriminative substructure to accomplish the graph classification task.
> > >
> > > Regarding your second question, we further analyzed the distribution of this key substructure across different classes: it appeared only 11 times in the majority class, but 348 times in the minority class. Such a significant difference in distribution strongly indicates that this substructure is relatively rare in the majority class, whereas it is highly representative in the minority class.

---

### Author Response · Authors · 2025-12-02
**Summary**

Dear ACs and Reviewers,

We sincerely thank all reviewers for taking valuable time to provide profound and constructive comments on our work.

For the convenience of the ACs, we will summarize each reviewer’s comments along with our responses.

---

**Key Strengths Highlighted by Reviewers**

- **Motivation & Problem Clarity:** Clear and well-motivated problem definition, acknowledging the importance of the graph imbalance problem in real-world applications (Reviewer qbQQ, 5aWh).

- **Methodological and Theoretical Novelty:** An interesting, theoretically grounded dynamic balanced prototype design coupled with an innovative, plug-and-play UniImb framework (Reviewer LB4s, 1cXq, 5aWh).

- **Open-Source Large-Scale Imbalanced Graph Dataset:** Introduces a new air-pollution graph classification dataset AirGraph for future research (Reviewer qbQQ, 5aWh).

- **Strong Experimental Performance with Extensive Experiments:** Demonstrated effectiveness and robustness across various datasets and settings (Reviewer qbQQ, LB4s, 1cXq, 5aWh).

---
**Concerns and Reply of Each Reviewer**

`Reviewer qbQQ (Score:4)`

- Graph datasets with 3D conformations (W1, Q3)

A: we introduced three graph datasets with 3D conformations to show the effectiveness of our method.

- Evaluation of the effectiveness of the standalone core DBP (W2, Q1)

A: We further highlighted our novelty and conducted additional ablation and scalability analyses to further clarify the contribution of our core module, DBP.


- Visual analyses of DBP (Q2)

A: We used visualization methods to demonstrate the key topological patterns that can be directly captured by DBP. In addition, we employed Shapley value and substructure frequency analysis to further reinforce the role of DBP.

---
`Reviewer LB4s (Score:6)`

- Refinement of certain paper details (W1, W5)

A: We added the algorithm pseudocode (W1) and added a description of the AirGraph dataset in the main text (W5).

- Explanation of the current model’s gaps (W2)

A: We provide a detailed explanation, supported by experiments, of the limitations of existing models in handling complex graph imbalanced scenarios.

- Experiment in "Intertwined" scenario (W3)

A: We conducted comprehensive experiments for this scenario in Figure 2 and Figure 3 in the original manuscript.

In the discussion, **we further clarified that the strength of our model lies in its comprehensive handling of various degrees of class imbalance and topological imbalance, as well as their intertwined scenarios, rather than focusing solely on a single intertwined imbalance issue.**

- Validity of the graph perturbation strategy (W4)

A: We provide a comprehensive ablation study of this strategy in Appendix I.5 of the original manuscript, demonstrating its positive effect on improving minority class learning.

---
`Reviewer 1cXq (Score:6)`

- Evaluation of the effectiveness of the standalone core DBP (W1)

A: We conducted extensive ablation and scalability experiments to validate the core contributions of Dynamic Balanced Prototype (DBP).

- Explanations and comparison of existing model (Q1, Q2, Q4)

A: We explained the reasons for the poor performance of TopoImb (Q1) and compared our method with SamGoG (Q2). Additionally, we demonstrated the inefficiency of ImbGNN in complex graph imbalance scenarios (Q4).

- More discussions of related work (Q3)

A: In Appendix B of the revised submission, we provided more related work (e.g., representative works in node, edge, and potentially subgraph).

- Presentation errors (Q5 & Q6)

A: We revised the performance descriptions (Q5) and corrected errors in the citations (Q6) based on the reviewers’ comments.

- More experiments on multi-class datasets (Q7)

A: We introduced three multi-class datasets and three 3D graph datasets to enhance the domain diversity of our experiments.

---

`Reviewer 5aWh (Score:6)`

- Additional hyperparameter combination experiments and performance evaluation of Graph Transformers equipped with imbalance strategies (W1 & Q4)

A: We conducted combined hyperparameter experiments (W1) and added experiments combining Graph Transformers with imbalance strategies (Q4).

- Experiments on biological and industrial datasets (W3)

A: We open-sourced the AirGraph dataset and introduced biological multi-class datasets along with 3D graph datasets to demonstrate the method's effectiveness in addressing real-world imbalance challenges.

- Issues regarding Model Details (W2)

A: We added ablation and scalability experiments to validate the core contribution of Dynamic Balanced Prototype(DBP).

- Explanations of certain descriptions and minor errors in the paper (Q1, Q2 & Q3)

A: We addressed some confusions (Q1, Q2) and clarified descriptions (Q3) and corrected minor issues.

---
We sincerely thank all reviewers and the ACs for their time, thoughtful evaluations, and constructive feedback throughout the discussion phase.

Best regards,

All Authors

---

### Meta-Review · Area_Chair_oQF7 · 2026-01-06

**Summary:**

Reviewers acknowledged the paper's clear motivation of unifying class and topological imbalance, the novel and theoretically grounded core module, strong and extensive experiments, and a valuable new dataset. Primary concerns centered on perceived novelty, insufficient experimental focus on the intertwined scenario, and validation on more complex, real-world data.

**Reviewer Concerns:**

Addressed concerns:
The rebuttal comprehensively addressed key concerns by adding: experiments on 3D molecular graphs and multi-class datasets; extensive ablation studies isolating DBP's effectiveness; additional analyses; and comparisons with recent baselines.

Remaining concerns:
The main experiments slightly misalign with the core unified for intertwined scenarios narrative. However, the provided analyses on naturally imbalanced data and scale breakdowns mitigate this as a critical flaw.

**Reviewer Scores:**

Reviewer 1cXq (6 → Likely 6): All questions were thoroughly answered, warranting a positive score.

Reviewer 5aWh (6 → Likely 6 or 8): All weaknesses and queries were resolved with substantial new experiments.

Reviewer LB4s (6 → Likely 6): Appreciated detailed responses and new analyses.

Reviewer qbQQ (4 → Likely 6): Major concerns were directly addressed with new experiments, leading to a clear positive shift.

---

### Decision · Program_Chairs · 2026-01-26

Accept (Oral)